# FLOWBENCH: A ROBUSTNESS BENCHMARK FOR OPTICAL FLOW ESTIMATION

## ABSTRACT

Optical flow estimation is a crucial computer vision task often applied to safety-critical real-world scenarios like autonomous driving and medical imaging. While optical flow estimation accuracy has greatly benefited from the emergence of deep learning, learning-based methods are also known for their lack of generalization and reliability. However, reliability is paramount when optical flow methods are employed in the real world, where safety is essential. Furthermore, a deeper understanding of the robustness and reliability of learning-based optical flow estimation methods is still lacking, hindering the research community from building methods safe for real-world deployment. Thus we propose FLOWBENCH, a robustness benchmark and evaluation tool for learning-based optical flow methods. FLOWBENCH facilitates streamlined research into the reliability of optical flow methods by benchmarking their robustness to adversarial attacks and out-of-distribution samples. With FLOWBENCH, we benchmark 91 methods across 3 different datasets under 7 diverse adversarial attacks and 23 established common corruptions, making it the most comprehensive robustness analysis of optical flow methods to date. Across this wide range of methods, we consistently find that methods with state-of-the-art performance on established standard benchmarks lack reliability and generalization ability. Moreover, we find interesting correlations between performance, reliability, and generalization ability of optical flow estimation methods, under various lenses such as design choices used, number of parameters, etc. After acceptance, FLOWBENCH will be open-source and publicly available, including the weights of all tested models.

## 1 INTRODUCTION

NEW

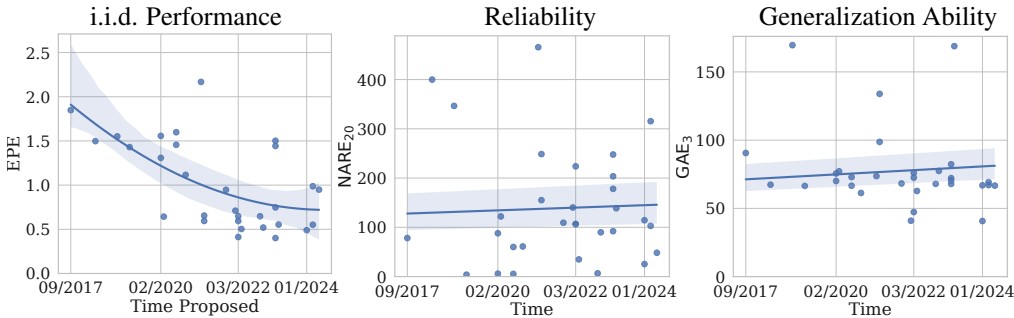

Figure 1: Optical flow estimation methods proposed over time and their reliability and generalization ability. In all three plots, the y-axis represents error, i.e., lower is better. The error of optical flow estimation methods on independent and identically distributed data samples (i.i.d.) has decreased over time, however, their reliability and generalization ability are stagnant if not deteriorating.

The recent growth of Deep Learning (DL) has greatly benefited computer vision, in particular when considering complex tasks such as the estimation of optical flow fields. In optical flow estimation, a method is supposed to estimate the movement of every pixel between at least two consecutive image frames in a subpixel-accurate manner. This task was earlier performed using model-driven

approaches such as Horn & Schunck (1981) and Lucas & Kanade (1981). However, these methods have severe limitations leading to suboptimal estimations and, consequently, to the predominant use of DL to perform the estimations (Dosovitskiy et al., 2015; Ilg et al., 2017; Jahedi et al., 2024b). The performance of learning-based optical flow estimation methods has improved over the years on independent and identically distributed data samples (i.i.d.), leading to lower errors on evaluation as shown by Fig. 1 (left). At the same time, DL-based methods are known to be unreliable (Geirhos et al., 2018; Prasad, 2022), they tend to learn shortcuts rather than meaningful feature representations (Geirhos et al., 2020), and can be easily deteriorated even by small corruptions. This can become a practical threat, as optical flow estimation is highly relevant in safety-critical applications such as autonomous driving (Capito et al., 2020; Wang et al., 2021), robotic surgery (Rosa et al., 2019) and others. Thus, before deploying DL-based optical flow estimation methods, assessing their vulnerability and generalization ability is of paramount importance to gauge their readiness. We observe in Fig. 1 that over the years, despite improvement in the performance of learning-based optical flow estimation methods, their reliability and generalization ability are almost unchanged. Had recent research been focused on these factors, the newly proposed methods could have been more reliable and ready for practical use. Our proposed FLOWBENCH facilitates this study, streamlining it for future research to utilize.

Many works have highlighted the importance of such a study by reducing model vulnerability (Xu et al., 2021b; Croce et al., 2023; Agnihotri et al., 2023; Schrodi et al., 2022; Tran et al., 2022; Grabinski et al., 2022), showing that robustness does follow from high accuracy (Tsipras et al., 2019; Schmidt et al., 2018; Schmalfuss et al., 2022b) or improving generalization (Hendrycks et al., 2020; Hoffmann et al., 2021) for various downstream tasks such as image classification, semantic segmentation, image restoration and others. To facilitate this research, robustness benchmarking tools and benchmarks like Croce et al. (2021); Jung et al. (2023); Tang et al. (2021) have been proposed for image classification models. They look into the adversarial and Out-of-Distribution (OOD) robustness of DL models. However, these works are limited to image classification. A similar benchmarking tool and comprehensive benchmark for optical flow is amiss.

To bridge this gap, we propose FLOWBENCH that facilitates robustness evaluations of optical flow models against adversarial attacks and image corruptions for OOD data and provides a unified evaluation scheme and streamlined code. Using FLOWBENCH, we benchmark 91 model checkpoints over 3 commonly used optical flow estimation datasets. These model checkpoints include SotA optical flow estimation methods and evaluation methods including SotA adversarial attacks and image corruption methods. FLOWBENCH is easy to use and new methods, when proposed, can be easily integrated to benchmark their performance. This will help researchers build better models that are not limited to improved performance on identical and independently distributed (i.i.d.) samples and are less vulnerable to adversarial attacks while generalizing better to image corruptions.

The main contributions of this work are as follows:

- We provide a benchmarking tool FLOWBENCH to evaluate the performance of most DL-based optical flow estimation methods over different datasets and make 91 checkpoints over different datasets publicly available for streamlined benchmarking while enabling the research community to add further checkpoints.

- We benchmark the aforementioned models against SotA and other commonly used adversarial attacks and common corruptions that can be easily queried using FLOWBENCH.

- We perform an in-depth analysis using FLOWBENCH and present interesting findings showing that methods that are SotA on i.i.d. are remarkably less reliable and generalize worse than other non-SotA methods.

- We analyze correlations between performance, reliability, and generalization abilities of optical flow estimation methods, under various lenses such as design choices, and the number of learnable parameters.

- We show that the optimization of white-box adversarial attacks for optical flow estimation can be performed even without the availability of ground truth predictions, furthering the scope of study in their reliability.

## 2 RELATED WORK

FLOWBENCH is the first robustness benchmarking tool and benchmark for optical flow estimation methods that unifies adversarial and OOD robustness, taking inspiration from robustness benchmarks for other vision tasks such as image classification. While several previous works provide benchmarking tools for optical flow estimation, they only facilitate benchmarking of either adversarial or OOD robustness and are less comprehensive than FLOWBENCH. FLOWBENCH leverages the individual strengths of prior benchmarking tools, but casts them into a unified and easy-to-use robustness benchmark. Following, we discuss these related works in detail.

### 2.1 ROBUSTNESS BENCHMARKING FOR IMAGE CLASSIFICATION METHODS

Goodfellow et al. (2015) proposed the Fast Sign Gradient Method (FGSM) attack which gave rise to the domain of adversarial attacks on image classification. Complementing adversarial attacks, Hendrycks & Dietterich (2019) proposed 2D Common Corruptions for image classification tasks on the CIFAR-100 (Krizhevsky et al., 2009) and ImageNet-1k (Russakovsky et al., 2015) datasets and their variants. Since then, most adversarial attacks and OOD Robustness works have focused on image classification tasks, warranting a consolidated benchmarking tool and benchmark for robustness. In the case of image classification, this gap was filled by multiple works such as Robust-Bench (Croce et al., 2021) and RobustArts (Tang et al., 2021). Both works make multiple image classification model checkpoints publicly available, including checkpoints trained for improved robustness. Moreover, RobustBench is a benchmarking tool that facilitates evaluating both adversarial and OOD robustness of image classification models. Other similar benchmarking tools exist, like DeepFool (Moosavi-Dezfooli et al., 2016), Torchattacks (Kim, 2020), and Foolbox (Rauber et al., 2020). Yet, these are merely benchmarking tools and do not provide a comprehensive benchmark - they only facilitate evaluating adversarial robustness but not the OOD robustness of the method. As of now, no benchmarking tool or benchmark exists for optical flow estimation methods' robustness evaluations. Thus, we propose FLOWBENCH which enables benchmarking adversarial and OOD robustness and makes a multitude of model checkpoints available, providing the research community with the much needed tools.

### 2.2 BENCHMARKING OPTICAL FLOW ESTIMATION METHODS

Optical flow estimation has been a problem attempted to be solved for a long time. Over time multiple works have been proposed to streamline research in this direction by providing benchmarking libraries for i.i.d. performance of proposed methods. Such libraries include *mmflow* (Contributors, 2021), *ptlflow* (Morimitsu, 2021), and *Spring* (Mehl et al., 2023). These libraries also provide model checkpoints to facilitate evaluations. *Spring*, also provides a benchmark but the performance evaluations are limited to their proposed Spring dataset. Whereas, both *mmflow* and *ptlflow* do not provide a benchmark but enable benchmarking on multiple optical flow datasets such as FlyingThings3D (Mayer et al., 2016), KITTI2015 (Menze & Geiger, 2015) and MPI Sintel (Butler et al., 2012). However, the evaluation abilities of these benchmarking tools are limited to i.i.d. data. Thus, we built FLOWBENCH, using *ptlflow* and publicly available model checkpoints to extend method evaluations to adversarial and OOD Robustness consolidating research towards reliability and generalization ability of optical flow estimation methods. Additionally, FLOWBENCH is the first to provide a comprehensive benchmark on existing optical flow estimation methods over 3 datasets and multiple adversarial attacks and image corruptions.

### 2.3 ADVERSARIAL ATTACKS

As discussed in Sec. 1, DL models tend to learn shortcuts to map data samples from input to target distribution (Geirhos et al., 2020), leading to the model learning inefficient feature representations. In their work, Goodfellow et al. (2015) showed that this inefficient learning of feature representations can be easily exploited. Goodfellow et al. (2015) added noise to the input data samples which was optimized to increase loss using model information, such that the model was fooled into making incorrect predictions. This demonstrated the vulnerability and unreliability of model predictions as the perturbed input samples still appeared semantically similar to the human eye. They named this attack the **Fast Sign Gradient Method** (FGSM). This attack led to an increased inter-

est by the research community to better optimize the noise inspiring multiple other works such as Basic Iteration method (BIM) (Kurakin et al., 2018), Projected Gradient Descent (PGD) (Kurakin et al., 2017), Auto-PGD (APGD) (Wong et al., 2020) and CosPGD (Agnihotri et al., 2024) which were direct extensions to FGSM, and other attacks such as Perturbation-Constrained Flow Attack (PCFA) (Schmalfuss et al., 2022b) and Adversarial Weather (Schmalfuss et al., 2023), which are indirect extensions of FGSM.

## 3 FLOWBENCH USAGE

In the following, we describe the benchmarking tool, FLOWBENCH. It is built using pltflow (Morimitsu, 2021), and supports 36 unique architectures (new architectures added to ptlflow over time are compatible with FLOWBENCH) and distinct datasets, namely FlyingThings3D (Mayer et al., 2016), KITTI2015 (Menze & Geiger, 2015), MPI Sintel (Butler et al., 2012) (clean and final) and Spring (Mehl et al., 2023) datasets (please refer Appendix C for additional details on the datasets). It enables training and evaluations on all aforementioned datasets including evaluations using SotA adversarial attacks such as CosPGD (Agnihotri et al., 2024) and PCFA (Schmalfuss et al., 2022b), Adversarial weather (Schmalfuss et al., 2023), and other commonly used adversarial attacks like BIM (Kurakin et al., 2018), PGD (Kurakin et al., 2017), FGSM (Goodfellow et al., 2015), under various Lipshitz ($l_p$) norm bounds.

Additionally, it enables evaluations for Out-of-Distribution (OOD) robustness by corrupting the inference samples using 2D Common Corruptions (Hendrycks & Dietterich, 2019) and 3D Common Corruptions (Kar et al., 2022).

We follow the nomenclature set by RobustBench (Croce et al., 2021) and use "threat_model" to define the kind of evaluation to be performed. When "threat_model" is defined to be "None", the evaluation is performed on unperturbed and unaltered images, if the "threat_model" is defined to be an adversarial attack, for example "PGD", "CosPGD" or "PCFA", then FLOWBENCH performs an adversarial attack using the user-defined parameters. We elaborate on this in Appendix E.1. Whereas, if "threat_model" is defined to be "2DCommonCorruptions" or "3DCommonCorruptions", the FLOWBENCH performs evaluations after perturbing the images with 2D Common Corruptions and 3D Common Corruptions respectively. We elaborate on this in Appendix E.2. If the queried evaluation already exists in the benchmark provided by this work, then FLOWBENCH simply retrieves the evaluations, thus saving computation.

FLOWBENCH enables the use of all the attacks mentioned in Sec. 2.3 to help users better study the reliability of their optical flow methods. We choose to specifically include these white-box adversarial attacks as they either serve as the common benchmark for adversarial attacks in classification literature (FGSM, BIM, PGD, APGD) or they are unique attacks proposed specifically for pixel-wise prediction tasks (CosPGD) and optical flow estimation (PCFA and Adversarial Weather). These attacks can either be *Non-targeted* which are designed to simply fool the model into making incorrect predictions, irrespective of what the model eventually predicts, or can be *Targeted*, where the model is fooled to make a certain prediction. Most attacks can be, designed to be either Targeted or Non-targeted, these include, FGSM, BIM, PGD, APGD, CosPGD, and Adversarial Weather. However, by design, some attacks are limited to being only one of the two, for example, PCFA which is a targeted attack.

Following we show the basic commands to use FLOWBENCH. We describe each attack and common corruption supported by FLOWBENCH in detail in Appendix E. Please refer to Appendix G for details on the arguments and function calls.

### 3.1 MODEL ZOO

It is a challenge to find all checkpoints, while training them is a time and compute exhaustive process. Thus we gather available model checkpoints from various sources such as ptlflow (Morimitsu, 2021) and mmflow (Contributors, 2021). The trained checkpoints for all models available in FLOWBENCH can be obtained using the following lines of code:

```
from flowbench.evals import load_model
model = load_model(model_name='RAFT', dataset='KITTI2015')
```

Each model checkpoint can be retrieved with the pair of 'model_name', the name of the model, and 'dataset', the dataset for which the checkpoint was last finetuned. In Appendix F we provide a complete overview of all the 91 available pairs of model checkpoints and datasets.

## 3.2 ADVERSARIAL ATTACKS

FLOWBENCH can be used to evaluate models on the discussed adversarial attacks using the following lines of code (please refer Appendix G.1 for details regarding the arguments):

```python
from flowbench.evals import evaluate
model, results = evaluate(model_name='RAFT', dataset='KITTI2015',
                          threat_model='CosPGD', iterations=20, alpha=0.01,
                          epsilon=8/255, lp_norm='Linf', targeted=True,
                          optim_wrt='ground_truth', retrieve_existing=True)
```

## 3.3 OOD ROBUSTNESS                                                    NEW

FLOWBENCH can be used to evaluate models on the 2D and 3D Common Corruptions using the following lines of code, following is an example for the latter (please refer Appendix G.3 (2D Common Corruptions) and Appendix G.4 (3D Common Corruption) for details regarding the arguments):

```python
from flowbench.evals import evaluate
model, results = evaluate(model_name='RAFT', dataset='KITTI2015',
                          threat_model='3DCommonCorruption',
                          severity=3, retrieve_existing=True)
```

## 4 METRICS FOR ANALYSIS AT SCALE

Analysis of optical flow estimation methods at the same scale as this work, especially under the lens of reliability and generalization ability has not been attempted before. The most commonly (Schrodi et al., 2022; Schmalfuss et al., 2022a; Agnihotri et al., 2024; Dosovitskiy et al., 2015) used metric for evaluating the performance of a method is calculating the mean End-Point-Error (EPE) between the predicted optical flow and the ground truth for all pairs of frames in a given dataset. However, this does not reflect the reliability and generalization ability of the method. Moreover, this work has performed over 4500 experiments in total, and analyzing the EPE from each experiment would not lead to a fruitful finding. Thus, we attempt to simplify this with our proposed metrics, the Reliability Error and Generalization Ability Error.

The objective of any optical flow estimation method is to obtain an EPE of zero or as low as possible. The larger the EPE, the worse the performance of the method. Most works (Dosovitskiy et al., 2015; Teed & Deng, 2020; Ilg et al., 2017; Huang et al., 2022) report the mean EPE value over a dataset as a measure of the method's performance. For reliability and generalization, we look at the maximum possible value of mean EPE across attacks over multiple datasets. That is, we ask the question "What is the worst possible performance of a given method?". An answer to this question tells us about the reliability and generalization ability of a method. In the following, we describe the measures for different scenarios in detail.

## 4.1 GENERALIZATION ABILITY ERROR                                      NEW

Inspired by multiple works (Croce et al., 2021; Hendrycks et al., 2020; Hoffmann et al., 2021) that use OOD Robustness of methods for evaluating the generalization ability of the method, even evaluate over every common corruptions, that is 2D Common Corruptions and 3D Common Corruptions combined. Then, we find the maximum of the mean EPE w.r.t. the ground truth for a given method, across all corruptions at a given severity and report this as Generalization Ability Error denoted by $\text{GAE}_{severity\ level}$. For example, for severity 3, the measure would be denoted by $\text{GAE}_3$. The less the GAE value, the better the generalization ability of the given optical flow estimation method. These corruptions perturb the images to cause distributions and domain shifts, such shifts often confuse the methods into making incorrect predictions.

For calculating GAE, we use all 15 2D Common Corruptions: 'Gaussian Noise', Shot Noise', 'Impulse Noise', 'Defocus Blur', 'Frosted Glass Blur', 'Motion Blur', 'Zoom Blur', 'Snow', 'Frost', 'Fog', 'Brightness', 'Contrast', 'Elastic Transform', 'Pixelate', 'JPEG Compression', and eight 3D Common Corruptions: 'Color Quantization', 'Far Focus', 'Fog 3D', 'ISO Noise', 'Low Light', 'Near Focus', 'XY Motion Blur', and 'Z Motion Blur'. All the common corruptions are at severity 3. Kar et al. (2022) offers more 3D Common Corruptions, however computing them is resource intensive. Thus, given our limited resources and an overlap in the corruptions between 2D Common Corruptions and 3D Common Corruptions, we focus on generating 3D Common Corruptions that might be unique from their 2D counterpart, require fewer sources to generate, and are interesting from an optical flow estimation perspective. In Appendix A we show that these synthetic common corruptions can indeed be used as a proxy for possible corruptions when in the wild in the real world.

## 4.2 RELIABILITY ERROR

NEW

An adversarial attack is a perturbation made on the input images to fool a method into changing its predictions while the input image looks semantically similar to a human observer. Most works that focus on the reliability of optical flow estimation methods perform adversarial attacks, however, these works either focus on targeted attacks or on non-targeted attacks, not both at the same time. The objective of targeted attacks is to optimally perturb the input image such that the method predictions are changed towards a specifically desired target, for example, a target can be a $\vec{0}$ flow i.e. attacking so that the flow prediction at all pixels should become zero. Conversely, non-targeted adversarial attacks do not intend to shift the method's predictions to a specific target, they simply intend to fool the method into making any incorrect predictions. To streamline research into the reliability of these methods, we perform both targeted and non-targeted attacks.

**Non-Targeted Attacks.** For non-targeted attacks, we measure the EPE w.r.t. the ground truth, in this case, the higher the EPE value, the worse the performance of the optical flow estimation method. The notation for this metric is, $\text{NARE}_{\text{attack iterations}}$, where NARE stands for Non-targeted Attack Reliability Error, and the subscript informs the number of attack iterations used for optimizing the attack. For example, when 20 attack iterations were used to optimize the attack then the metric would be $\text{NARE}_{20}$. The higher the NARE value, the worse the reliability of the optical flow method.

**Targeted Attacks.** For targeted attacks, we measure the EPE w.r.t. the target flow, however, to standardize notations, we report the negative EPE in this case, thus, the higher the value, the worse the performance of the optical flow estimation method. The notation for this metric is, $\text{TARE}_{\text{attack iterations}}^{\text{target}}$, where TARE stands for Targeted Attack Reliability Error and the superscript informs about the target used (zero vector or negative of the initial flow prediction) and the subscript informs about the number of attack iterations used for optimizing the attack. For example, when the target is $\vec{0}$ and 20 attack iterations were used to optimize the attack then the metric would be $\text{TARE}_{20}^{\vec{0}}$. The higher the TARE value, the worse is the reliability of the optical flow method.

For calculating TARE and NARE values we used BIM, PGD, and CosPGD attack with step size $\alpha$=0.01, perturbation budget $\epsilon = \frac{8}{255}$ under the $\ell_\infty$-norm bound, as targeted and non-targeted attacks respectively. We use $\ell_\infty$-norm bound as we observe in Appendix H that there is a high correlation between the performance of optical flow estimation methods when attacked using $\ell_\infty$-norm bounded attacks and $\ell_2$-norm bounded attacks. We use 20 attack iterations for calculating TARE and NARE as we observe in *Appendix H*, that at a lower number of iterations, the gap in performance of different optical flow estimation methods is small, thus an in-depth analysis would be difficult, and we do not go beyond 20 attack iterations as computing each attack step for an adversarial attack is very expensive, and as shown by Agnihotri et al. (2024) and Schmalfuss et al. (2022b), 20 iterations are enough to optimize an attack to truly understand the performance of the attacked method.

## 5 ANALYSIS AND INTERESTING FINDINGS

To demonstrate the potential of FLOWBENCH, we use it to perform multiple analyses which provide us with a better understanding of many optical flow estimation methods, including novel findings.

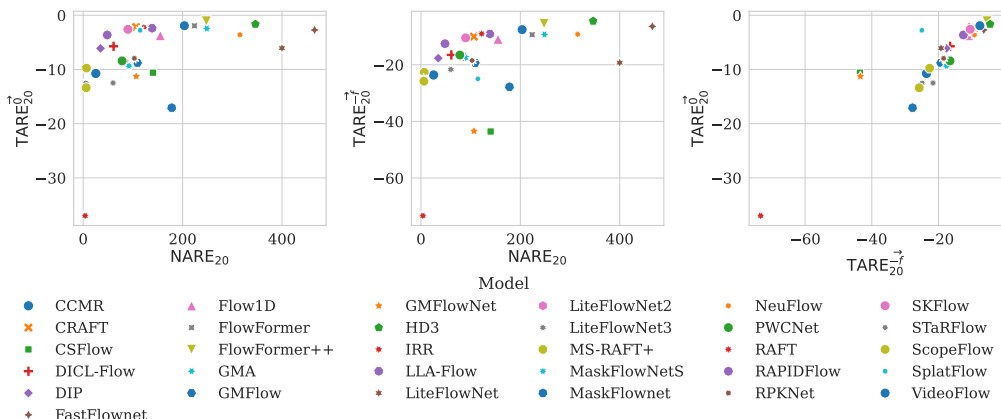

Figure 2: Analysing correlations between Targeted and Non-targeted adversarial attacks. A model is more reliable if it has a low NARM value and a high TARM value.

Following, we discuss the observations made in the comprehensive robustness benchmark created using FLOWBENCH. Please refer to Appendix C for details on the dataset, Appendix D for additional implementation details, and Appendix H for additional results from the benchmarking.

## 5.1 TARGETED V/S NON-TARGETED ADVERSARIAL ATTACKS

We benchmark the performance of all prominent DL-based optical flow estimation methods across three datasets, namely KITTI2015, MPI Sintel (clean), and MPI Sintel (final) against SotA and commonly used adversarial attacks such as BIM, PGD, and CosPGD. Then, we compare the NARE and TARE values (introduced in Sec. 4.2) and find correlations in their performance. These are reliability metrics, higher NARE and TARE values indicates low reliability and vice versa. Please refer to Appendix D for more implementation details. We observe in Fig. 2 that there is a very high correlation between the $\mathrm{TARE}^{\vec{0}}$ and $\mathrm{TARE}^{\overrightarrow{-f}}$ values of every optical flow estimation method. This shows that evaluating either one of the values can serve as a reliable proxy for the other. We use this finding in the later analysis. Additionally, in Fig. 2 we observe that most optical flow estimation methods like ScopeFlow (Bar-Haim & Wolf, 2020), MS-RAFT+ (Jahedi et al., 2024b) and StarFlow (Godet et al., 2021) are relatively more susceptible to targeted attacks than they are to non-targeted attacks. On the other hand, some methods are highly susceptible to both and thus very unreliable, these include SKFlow (Sun et al., 2022), FastFlowNet (Kong et al., 2021), HD3 (Yin et al., 2019) and some SotA methods like FlowFormer (Huang et al., 2022) and FlowFormer++ (Shi et al., 2023b). Interestingly, IRR (Hur & Roth, 2019) stands out as the most reliable optical flow estimation method as it is robust to both targeted and non-targeted adversarial attacks. While ScopeFlow (Bar-Haim & Wolf, 2020), GMFlowNet (Zhao et al., 2022) and MaskFlowNet (Zhao et al., 2020) are less reliable than IRR but more reliable than the other methods.

## 5.2 RELIABILITY V/S GENERALIZATION

Following we analyze if there is a correlation between the reliability and generalization ability of optical flow estimation methods. We observe in Fig. 3, that most methods that have a good performance also generalize better, however, methods like FlowFormer++, while having good i.i.d. performance have a relatively poor generalization ability. As observed in Sec. 5.1, HD3 Yin et al. (2019) stands out as having poor performance and poor generalization ability. Interestingly, as shown by Fig. 3, there is a correlation between the generalization ability ($\mathrm{GAE}_3$ values, introduced in Sec. 4.1, higher GAE value indicates lower generalization ability) and reliability when measured using non-targeted adversarial attacks ($\mathrm{NARE}_{20}$ values). Additionally, most methods identified in Sec. 5.1 to be reliable, for example, CSFLow, MaskFlowNet also have considerable generalization ability compared to the other methods. However, IRR which stood out as the most reliable method has low generalization abilities. It is interesting to note that CCMR (Jahedi et al., 2024a) offers a good trade-off as it has reasonably good performance, reliability, and generalization abilities.

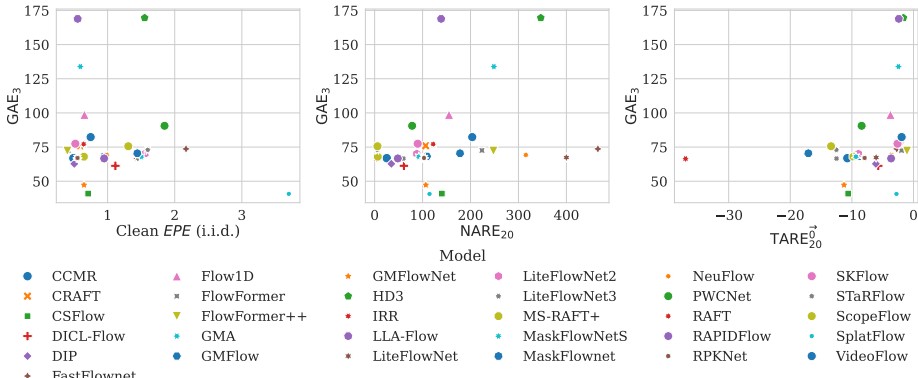

Figure 3: Analysing correlations between reliability and generalization ability of optical flow estimation methods.

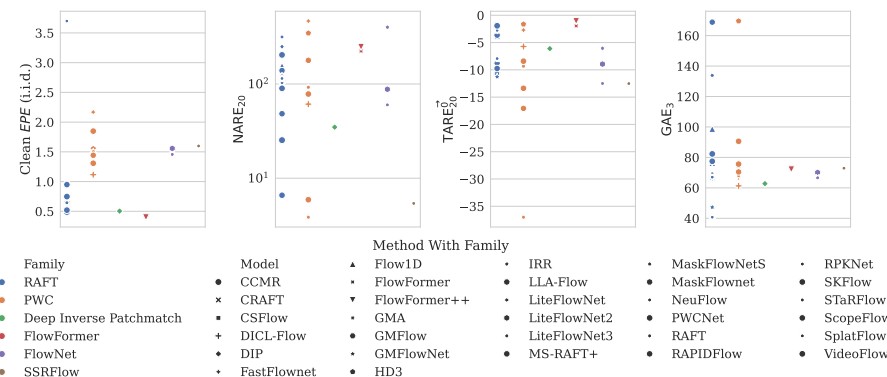

Figure 4: Analyzing correlations between the method family to which the optical flow estimation method belongs and its corresponding performance, reliability, and generalization ability.

## 5.3 ANALYZING METHOD FAMILIES

NEW

Optical flow estimation methods proposed over the years use different training strategies and architecture designs. However, there exist many architectural similarities between the methods, and based on these most methods can be broadly classified into four method families: FlowNet-family, PWC-family, RAFT-family, and FlowFormer-family (please refer to Appendix B for detailed justifications). In Fig. 4 we observe that methods belonging to the FlowFormer-family and RAFT-family and DIP Zheng et al. (2022) have the best i.i.d. performance, however, given their relatively higher $NARE_{20}$ and $TARE_{20}$ values, some exceeding 100, they appear to not be reliable. Here we observe that IRR (Hur & Roth, 2019) stands out as one of the most reliable methods under adversarial attacks. Given that the primary differences between IRR-PWC and other methods from the PWC family are the classical energy minimization-inspired approach and the use of residual networks to propose an iterative residual refinement, it makes an interesting finding.

When considering generalization ability under common corruptions, we observe all methods to have poor performance. Methods such as LLA-Flow Xu et al. (2023b) and HD3 Yin et al. (2019) from the RAFT-family and PWC-family respectively have $GAE_3$ values over 160! Here, SplatFlow Wang et al. (2024) stands out, given that the primary difference between SplatFlow and other RAFT-family methods is the use of splatting for feature matching by SplatFlow, it is an interesting finding.

Additionally, we observe in Fig. 4 that compared to other method families, FlowFormer-family is very susceptible to targeted adversarial attacks. Given that the FlowFormer family comprises only transformer-based architectures for optical flow estimation, this is very interesting, as this contradicts the observations made for transformer-based methods for image classification (Paul & Chen, 2022;

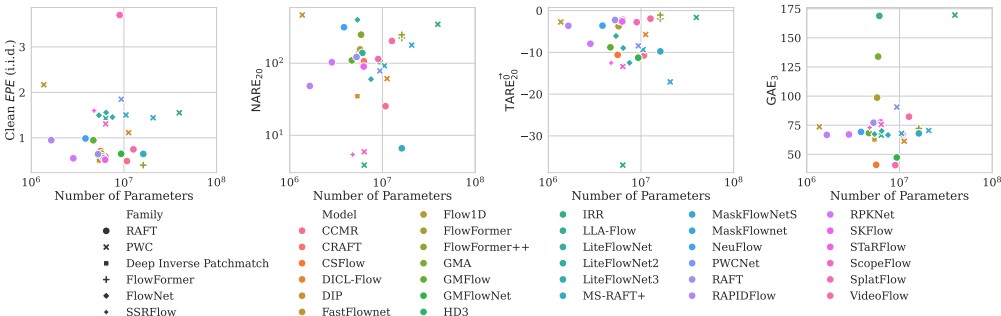

Figure 5: Analysing correlation between the number of learnable parameters in a DL-based optical flow estimation method and its performance, reliability, and generalization ability. Colors show the different optical flow methods while marker styles show the method family to which they belong.

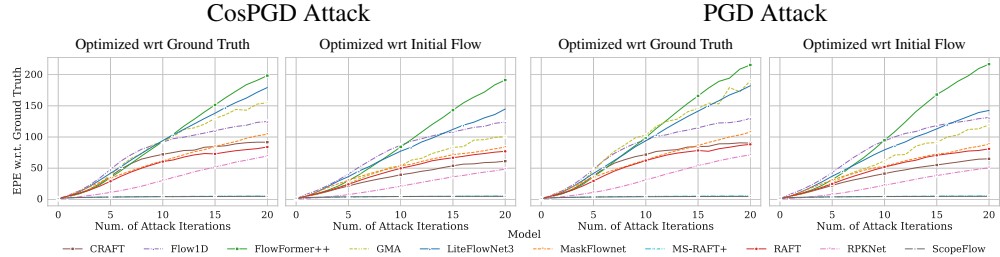

Figure 6: Performance of interesting optical flow estimation methods under different non-targeted adversarial attacks optimized using initial flow predictions on the KITTI2015 dataset.

Hoyer et al., 2022) and semantic segmentation (Xie et al., 2021), where they appear to be more robust. However, this lack of generalization ability can also be attributed to the use of dynamic positional cost queries by both FlowFormer and FlowFormer++, thus, we require more models to be proposed in the FlowFormer family to be certain.

### 5.4 IMPACT OF THE NUMBER OF LEARNABLE PARAMETERS

Many works for classification have shown that Deep Neural Networks with more parameters and less vulnerable to adversarial attacks and generalize better to common corruptions (Liu et al., 2022; Ding et al., 2022; Hoffmann et al., 2021). It would be interesting to see if the same holds true for optical flow estimation methods. Thus, we analyze this in Fig. 5 and observe that while the number of learnable parameters has an impact on the performance of the methods to some extent (other than the exceptions of MaskFlowNet and HD3), the same does not hold for reliability and generalization ability. Methods such as FlowFormer, FlowFormer++ (FlowFormer-family), and VideoFlow (RAFT-family) have relatively more parameters than other methods however they are less reliable and have a poor generalization ability. On the other hand, methods like CSFlow and SplatFlow (both RAFT-family) have significantly fewer parameters but are more reliable and generalize better than the other methods.

### 5.5 OPTIMIZING TARGETED ATTACKS USING INITIAL FLOW PREDICTIONS

Based on the observation in Sec. 5, we identify several interesting methods whose performance warrants additional analysis and discussion. Following, we discuss our observations in detail.

One of the major limitations of white-box adversarial attacks is that they require access to the ground truth to optimize the attack (Agnihotri et al., 2024). However, access to the ground truth is not guaranteed in every scenario. Additionally as discussed by Schmalfuss et al. (2022b), robustness is a measure of the difference in a model's prediction on perturbed input w.r.t. the model's prediction on clean input samples. Thus, the goal of an attack should be to fool the method into changing its initial

predictions (predictions when the method is not attacked), independent of the ground truth. Thus, we attempt to optimize the adversarial attack w.r.t. to the initial flow prediction on the unperturbed input sample before any attacks, as access to this is almost guaranteed. This helps us ascertain if initial flow predictions can be used as a proxy to ground truth while optimizing attacks. Thus, in Eq. (4), Eq. (8), Eq. (9) and there places where applicable $Y = X^{\text{clean}}$ (please refer Appendix E.1). However, this optimization is only possible for attacks that introduce certain randomness in the initial input sample, as shown by Eq. (7). This allows for there to exist a non-zero loss between the predictions on the clean input samples and the perturbed input samples allowing for optimization. We report the evaluations for CosPGD and PGD attack using the KITTI2015 dataset for 10 interesting methods in Fig. 6. We choose the optical flow estimation methods on the basis of their performance in Sec. 5 and their performance on i.i.d. samples. For additional evaluation using more models please refer to Appendix H. We observe in Fig. 6 that there appears a high correlation in the performance of all considered methods under attack when optimized using the ground truth flow and the initial flow prediction, Thus, initial flow predictions from methods do serve as a strong proxy to the ground truth for optimizing attacks. This new finding over a big sample, helps advance study in the reliability of optical flow methods, even when ground truth predictions are not available.

# 6  CONCLUSION

NEW

FLOWBENCH is the first robustness benchmarking tool and a novel benchmark for optical flow estimation methods. It currently supports 91 model checkpoints, over distinct datasets, and all relevant robustness evaluation methods including SotA adversarial attacks and image corruptions. We discuss the unique features of FLOWBENCH in detail and demonstrate that the library is user-friendly. Adding new evaluation methods or optical flow estimation methods to FLOWBENCH is easy and intuitive. In Sec. 5.1, we find that there is a high correlation in the performance of optical flow estimation methods against targeted attacks using different targets, thus saving compute for future works as they need to evaluate only against one target. In Sec. 5.2, we observe the methods known to be SotA on i.i.d. samples are not reliable, and do not generalize well to image corruptions, demonstrating the gap in current research when considering real-world applications. Additionally, we observe here that there is no apparent correlation between generalization abilities and the reliability of optical flow estimation methods. In Sec. 5.3, we show that methods from the FlowFormer family have good i.i.d. performance but are the most unreliable under targeted attacks, also that IRR stands out to have marginally better reliability. generalization abilities. In Sec. 5.4, we show that, unlike image classification, increasing the number of learnable parameters does not help increase the robustness of optical flow estimation methods, however, a couple of RAFT variants have marginally better generalization abilities even with fewer parameters. These observation helps us conclude that based on current works, different approaches might be required to attain reliability under attacks and generalization ability to image corruptions. Lastly, we show that white-box adversarial attacks on optical flow estimation methods can be independent of the availability of ground truth information, and can harness the information in the initial flow predictions to optimize attacks, thus overcoming a huge limitation in the field. Such an in-depth understanding of reliability and generalization abilities to optical flow estimation methods can only be obtained using our proposed FLOWBENCH. We are certain that FLOWBENCH will be immensely helpful in gathering more such interesting findings and its comprehensive and consolidated nature would make things easier for the research community.

**Future Work.**  For optical flow estimation, patch attacks are also interesting and widely studied (Ranjan et al., 2019; Schrodi et al., 2022; Scheurer et al., 2024). We plan to add such patch attacks to FLOWBENCH in future iterations. Schmalfuss et al. (2022b) proposed optimizing adversarial noise jointly for the consecutive image frames and also over the entire evaluation set. Only PCFA supports such optimization regimes in FLOWBENCH, so it would be interesting to extend such optimization to other adversarial attacks as well. Croce et al. (2021) show that the training methods used significantly impact the robustness of image classification methods. The same might be true for optical flow estimation methods, thus robustness evaluations under the lens of different training setups used would make an interesting extension to the analysis in this work. Lastly, traditional non-DL-based optical flow estimation methods might be more robust to adversarial attacks than current DL-based methods. Thus, it would be interesting to study their robustness and hopefully adapt them to increase the reliability of current methods.

## REPRODUCIBILITY STATEMENT

Every experiment in this work is reproducible and is part of an effort toward open-source work. FLOWBENCH will be open source and publicly available, including all evaluation logs and model checkpoint weights. This work intends to help the research community build more reliable and generalizable optical flow estimation methods such that they are ready for deployment in the real world even under safety-critical applications. FLOWBENCH is built upon ptlflow and thus any new model added with ptlflow would most likely be supported by FLOWBENCH as well.

There always exists stochasticity when evaluating adversarial attacks, due to the randomness these attacks exploit, and when evaluating common corruptions due to different seeds and calculation approximations made by different python libraries. Therefore, for transparency and reproducibility, we evaluate different runs on the same seed and different runs of different seeds. We report these evaluations in Appendix K, using Tab. 2 for adversarial attacks and Tab. 3 for common corruptions, and observe that the variance is extremely low and the analysis performed in this work still stands.

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

# FlowBench: A Robustness Benchmark for Optical Flow Estimation

## Paper #1055 Supplementary Material

TABLE OF CONTENT

The supplementary material covers the following information:

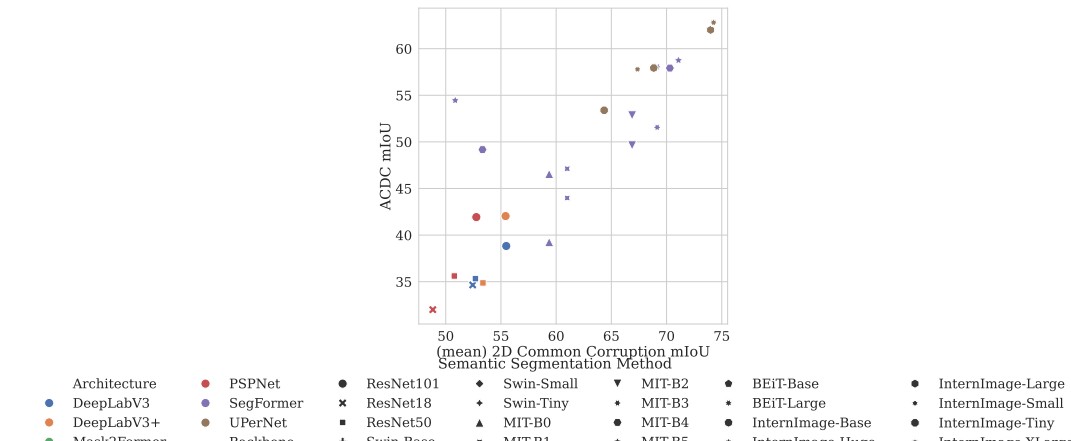

Figure 7: Results from work by Anonymous. Here they find a **very strong positive correlation between mean mIoU over the ACDC evaluation dataset (Sakaridis et al., 2021) and mean mIoU over each 2D Common Corruption (Hendrycks & Dietterich, 2019)** over the Cityscapes dataset (Cordts et al., 2016). All models were trained using the training subset of the Cityscapes dataset. ACDC is the Adverse Conditions Dataset with Correspondences for Semantic Driving Scene Understanding captured in similar scenes are cityscapes but under four different domains: Day/Night, Rain, Snow, and Fog in the wild. ACDC is a community-used baseline for evaluating the performance of semantic segmentation methods on domain shifts observed in the wild.

- – Appendix H.1.6: Evaluations for all models against Adversarial Weather attack, all four conditions: Fog, Rain, Snow, and Sparks, as targeted (both targets $\overrightarrow{0}$ and $\overrightarrow{-f}$) and non-targeted attack.
- – Appendix H.2: Evaluations for all models under 2D Common Corruptions and 3D Common Corruptions at severity 3, for KITTI2015, MPI Sintel (clean) and MPI Sintel (final) datasets.

- Appendix I: We share the initial prototype of the future website.

- Appendix J: We discuss the limitations of FLOWBENCH.

- Appendix K: We discuss the reproducibility of our evaluations and show that the variance in metrics is extremely low and our analysis comfortably holds under these variances.

## A  DO SYNTHETIC CORRUPTIONS REPRESENT THE REAL WORLD?

NEW

In their work Anonymous, they find the correlation between mean mIoU over the ACDC evaluation dataset (Sakaridis et al., 2021) and mean mIoU over each 2D Common Corruption (Hendrycks & Dietterich, 2019) over the Cityscapes dataset (Cordts et al., 2016). We include Figure 7 from their work here for ease of understanding. All models were trained using the training subset of the Cityscapes dataset. ACDC is the Adverse Conditions Dataset with Correspondences for Semantic Driving Scene Understanding captured in similar scenes are cityscapes but under four different domains: Day/Night, Rain, Snow, and Fog in the wild. ACDC is a community-used baseline for evaluating the performance of semantic segmentation methods on domain shifts observed in the wild. They find that there exists a very strong positive correlation between the two. This shows, that **yes, synthetic corruptions can serve as a proxy for the real world**. Unfortunately, a similar "in the wild" captured dataset does not exist for optical flow estimation to evaluate the effect of domain

shifts on the performance of optical flow methods. However, given that for the task of semantic segmentation, we find a very high positive correlation between the performance on real-world corruptions and synthetic corruptions, it is a safe assumption that the same would hold true for optical flow estimation as well. Thus, in this work, we evaluate against synthetic 2D Common Corruptions (Hendrycks & Dietterich, 2019) and synthetic 3D Common Corruptions (Kar et al., 2022).

## B    REASONS FOR CATEGORIZING METHODS TO THEIR RESPECTIVE FAMILIES

NEW

Over the years, various modifications have been proposed for DL-based optical flow estimation methods. These can be based on the training strategy used or new architectures. However, barring DIP Zheng et al. (2022) and StarFlow Godet et al. (2021) that appear to have significantly different architectures, all other optimal flow methods can be categorized into four major families: FlowNet (Dosovitskiy et al., 2015), PWC (Pyramid, Wrapping, and Cost Volume) (Sun et al., 2018), RAFT (Teed & Deng, 2020), and FlowFormer (Huang et al., 2022). Following we discuss the reasoning for the categorization of each method considered.

### B.1    FLOWNET FAMILY

NEW

Dosovitskiy et al. (2015) were the first to propose an end-to-end differentiable DL-based architecture for optical flow estimation, FlowNet. Many further works were inspired by FlowNet, making changes to FlowNet to propose novel optical flow estimation methods. These methods include:

- **FlowNet2.0** (Ilg et al., 2017): They improve upon FlowNet by changes to the schedule of training data usage, using a stacked architecture to include the warping of the second image with intermediate optical flow, and a sub-network to focus on small displacements.
- **LiteFlowNet** (Hui et al., 2018): Compared to FlowNet2.0 they use a more effective flow inference approach at each pyramid level through a lightweight cascaded network. They also use a flow regularization layer to ameliorate the issue of outliers and vague flow boundaries by using a feature-driven local convolution, and they use feature warping instead of image warping. They use the same training schedule as FlowNet2.0 but they train their network stage-wise.
- **LiteFlowNet2** (Hui et al., 2020): They improve the accuracy and latency from LiteFlowNet by making minor architectural changes to LiteFlowNet. They follow the training schedule of FlowNet2.0 to some extent and perform stage-wise training.
- **LiteFlowNet3** (Hui & Loy, 2020): They further improve upon the LiteFlowNet2.0 by amending each cost vector using an adaptive modulation before the flow decoding to alleviate the issue of outliers in the cost volume. Additionally, they replace each potentially inaccurately predicted optical flow with an accurate one from a near position through a warping of the flow field. They follow a special training schedule, first training the Lite-FlowNet2 modules as mentioned in Hui et al. (2020), and then training the entire architecture again with the LiteFlowNet3 modifications to LiteFlowNet2 following the training protocol from FlowNet2.0.

### B.2    PWC FAMILY

NEW

While still using features from different at different scales, warping, and cost volume, Sun et al. (2018) proposed PWC-Net which with its architectural changes, presented a significant shift in architectures from the traditional FlowNet. Sun et al. (2018) describe, "PWC-Net uses the current optical flow estimate to warp the CNN features of the second image. It then uses the warped features and features of the first image to construct a cost volume, which is processed by a CNN to estimate the optical flow." This was faster than FlowNet2.0, easier to train, and significantly outperformed it on established benchmarks like KITTI2015 (Menze & Geiger, 2015) and MPI Sintel (Butler et al., 2012). PWC-Net uses a similar training schedule and protocol as FlowNet2.0. Many other works followed PWC-Net either changing the training strategy or making architectural changes to PWC-Net to further improve on i.i.d. performance. These methods include:

- **FastFlowNet** (Kong et al., 2021): They replace the dual convolution feature pyramid in PWC-Net with the head enhanced pooling pyramid (HEPP) for enhancing the high-resolution pyramid feature and reducing model size, then, they propose center dense dilated correlation layer (MFC) for constructing compact cost volume while keeping the large search radius. followed by shuffle block decoders (SBD) at each pyramid level to regress optical flow with significantly cheaper computation. They follow the same training protocol mentioned by FlowNet2.0.

- **DICL** (Wang et al., 2020): They improve upon PWC-Net by decoupling the connection between 2D displacements and learn the matching costs at each 2D displacement hypothesis independently, i.e., displacement-invariant cost learning. They apply the same 2D convolution-based matching net independently on each 2D displacement hypothesis to learn a 4D cost volume and avoid learning a 5D feature volume, thus saving computing resources. They use the same training protocol as PWC-Net and FlowNet2.0, and use the data augmentations proposed by VCN.

- **HD3** (Yin et al., 2019): They adapt a PWC-Net-like architecture for the decomposition of the discrete probability distribution instead of the feature representations allowing them to learn probabilistic pixel correspondences in both optical flow and stereo matching. They decompose the full match density into multiple scales hierarchically and estimate the local matching distributions at each scale conditioned on the matching and warping at coarser scales. This allows the local distributions to be composed together to form the global match density. They essentially follow the same training protocol as FlowNet2.0 while omitting some hard examples. Additionally, they use ImageNet1k (Russakovsky et al., 2015)-pre-trained weights for their pyramid feature extractor.

- **IRR** (Hur & Roth, 2019): They take inspiration from classical energy minimization approaches, as well as residual networks to propose an iterative residual refinement, they show that their proposed IRR can be combined with both FlowNets and PWC-Net. In our work, we consider their adaptation to PWC-Net as that has better i.i.d. performance. They use the same training procedure as PWC-Net but additionally set out-of-bound pixels (after applying augmentations the same as those in FlowNet2.0) as occluded.

- **MaskFlowNet** (Zhao et al., 2020): Zhao et al. (2020) apart their proposed Occlusion-Aware Feature Matching Module (OFMM) and Asymmetric Occlusion-Aware Feature Matching Module (AsymOFMM) in PWC-Net and consists to two cascaded subnetworks for obtaining dual feature pyramids. Their proposed method helps them overcome the ambiguity caused due to occlusions in images that induce inaccuracies in the flow fields during warping. They use the same training protocol as IRR-PWC-Net. However, first, they train the MaskFlowNetS, then keep its weights frozen while training the entire MaskFlowNet. They use additional data from KITTI2015 and HD1k dataset (Kondermann et al., 2014) for fine-tuning on MPI-Sintel.

- **MaskFlowNetS** (Zhao et al., 2020): Proposed as the first stage of MaskFlowNet, MaskFlowNetS inherits the network architecture from PWC-Net, but replaces the feature matching modules (FMMs) by their proposed AsymOFMMs. They use the same training procedure as IRR-PWC-Net.

- **ScopeFlow** (Bar-Haim & Wolf, 2020): Bar-Haim & Wolf (2020) improve upon IRR-PWC-net by improving the data sampling process while testing the regularization and augmentations used to mitigate the bias induced by the training protocols. They keep some aspects of the training protocols from FlowNet2.0 intact while changing a few like cropping, and regularization at different stages of the multi-phase training.

- **VCN** Yang & Ramanan (2019): They improve upon the 4D cost volume used by variants of the PWC family by proposing volumetric encoder-decoder architectures that efficiently capture large receptive fields, multi-channel cost volumes that capture multi-dimensional notions of pixel similarities, and separable volumetric filtering that significantly reduces computation and parameters while preserving i.i.d. performance. They use a very similar training procedure as FlowNet2.0 and PWC-Net, however, with fewer iterations.

## B.3 RAFT FAMILY

NEW

Teed & Deng (2020) proposed Recurrent All-Pairs Field Transforms (RAFT) to extract per-pixel features to build a multi-scale 4D correlation volume for all pairs of pixels. Here a recurrent unit is used to perform lookups on these correlation volumes. They use additional data and fine-tuning compared to FlowNet2.0. RAFT was a significant architectural change from PWC-Nets and inspired many future works that made modifications to RAFT to further improve i.i.d. performance. These methods include:

- **CCMR** (Jahedi et al., 2024a): They propose adapting RAFT to use attention-based motion grouping concepts for multi-scale optical flow estimation. CCMR first computes global multi-scale context features and then uses them to guide the actual motion grouping. While iterating both steps over all coarse-to-fine scales, Jahedi et al. (2024a) adapt cross-covariance image transformers to allow for an efficient realization while maintaining scale-dependent properties. They use a training procedure similar to MS-RAFT+, after the traditionally followed training procedure of FlowNet2.0, they additionally finetune on a mixed set from KITTI and Viper dataset (Richter et al., 2017).

- **CRAFT** (Sui et al., 2022): CRAFT inherits the flow estimation pipeline of RAFT and revitalizes the correlation volume computation part with two proposed components: the Semantic Smoothing Transformer on the features from the second frame, and a Cross-Frame Attention Layer to compute the correlation volume. Sui et al. (2022) propose that these two components help suppress spurious correlations in the correlation volume. They use the same training procedure as RAFT.

- **CSFlow** (Shi et al., 2022): They propose, "Cross Strip Correlation module (CSC) and Correlation Regression Initialization module (CRI). CSC utilizes a striping operation across the target image and the attended image to encode global context into correlation volumes while maintaining high efficiency. CRI is used to maximally exploit the global context for optical flow initialization". They take inspiration from RAFT and adapt the multi-layer GRU from the stereo estimation task to optical flow. They follow a training procedure very similar to RAFT.

- **Flow1D** (Xu et al., 2021a): They take inspiration from transformers (Bao et al., 2022) and propose a 1D attention operation that is first applied in the vertical direction of the target image, and then a simple 1D correlation in the horizontal direction of the attended image to achieve 2D correspondence modeling effect. The directions of attention and correlation can also be exchanged, resulting in two 3D cost volumes that are concatenated for optical flow regression, where they adopt RAFT's framework to estimate the optical flow iteratively. They follow a very similar training procedure to RAFT, however for harder datasets, they use additional data for fine-tuning.

- **GMA** (Jiang et al., 2021a): They adapt an RAFT architecture to include their proposed global motion aggregation (GMA) module, a transformer-based approach to find long-range dependencies between pixels in the first image, and perform global aggregation on the corresponding motion features. This modified RAFT architecture with a GMA further inspired other architectures and works for optical flow estimation. GMA has a very similar training procedure to RAFT.

- **GMFlow** (Xu et al., 2022): They adapt RAFT to identify correspondences in image pairs by comparing their feature similarities. They use transformer-based modules to enhance extracted features, followed by self-attention modules for feature matching and flow propagation. Their feature extraction and feature upsampling modules are identical to RAFT. They follow a very similar training procedure to RAFT.

- **GMFlowNet** (Zhao et al., 2022): They adopt the iterative update operator of RAFT as the optimization step for their proposed GMFlowNet. They use their proposed patch-based overlapping attention (POLA) instead of multi-headed self-attention of transformer blocks to extract large context features to improve the matching step. They follow a very similar training procedure to RAFT.

- **LCV** (Khairi et al., 2024): They propose a lightweight module for learnable cost volume that adds onto RAFT to improve i.i.d. performance. For training, they initialize their learnable cost volume kernels to be identity and directly load the pre-trained weights from

RAFT, and then they follow a similar training schedule as RAFT but with significantly fewer iterations.

- **LLA-Flow** (Xu et al., 2023b): They propose the local similarity aggregation for 4D cost volume and present lightweight operations to diminish the impact of outliers caused by lack of texture. They apply their module on RAFT to improve i.i.d. performance. They follow a very similar training procedure to RAFT.

- **MS-RAFT+** (Jahedi et al., 2024b): MS-RAFT adapted RAFT for combining hierarchical concepts at multiple scales. MS-RAFT+ builds on top of MS-RAFT by exploiting an additional finer scale for estimating the flow, which is made feasible using the on-demand cost computation proposed by RAFT. They follow a very particular training schedule which is in parts similar to RAFT, however, due to an overhead of a number of learnable parameters, requires more data and training time.

- **MatchFlow** (Dong et al., 2023): They propose a different feature matching extractor (FME) to be used with RAFT and GMA module, this proposed FME is pre-trained on a different dataset, which allows for increased i.i.d. performance due to better feature extraction and matching. After incorporating the pre-trained FME, the resultant MatchFlow is trained very similarly to RAFT.

- **RapidFlow** (Morimitsu et al., 2024a): Inspired by RAFT, Morimitsu et al. (2024a) propose Recurrent Adaptable Pyramids with Iterative Decoding. They propose a recurrent feature encoder that uses a single shared block with efficient 1D layers (NeXt1D) to generate feature pyramids of variable levels. Their decoder is similar to RAFT, with a few changes inspired by SKFLow (Sun et al., 2022). They follow a very similar training procedure to RAFT.

- **RPKNet** (Morimitsu et al., 2024b): They adapt RAFT to use their proposed Partial Kernel Convolution (PKConv) layers and Separable Large kernels (SLK). PKConv is used to produce variable multi-scale features with a single shared block, while SLK is used to capture large context information with low computational cost. They follow a very similar training procedure to RAFT.

- **SCV** (Jiang et al., 2021b): They adapt RAFT to use a sparse correlation volume instead of a dense correlation volume. They follow a very similar training procedure to RAFT.

- **SeparableFlow** (Zhang et al., 2021): They propose a separable cost volume module, a drop-in replacement to RAFT's correlation cost volumes, that uses non-local aggregation layers to exploit global context cues and prior knowledge, to disambiguate motions in poorly constrained ambiguous regions. They follow a training procedure the same as RAFT.

- **SKFlow** (Sun et al., 2022): They propose using Super Kernels that allow for larger receptive fields allowing it to recover occluded motions. Finally, they use the non-local GMA module from GMA for optical flow estimations. They follow a similar training procedure to RAFT.

- **SplatFlow** (Wang et al., 2024): They essentially propose to use splatting for feature matching in architectures like RAFT and GMA. As SplatFlow is proposed to be a multi-frame optical flow estimation method, it requires three frames at a time for training as opposed to the two frames used by RAFT and GMA. We use their modified version with GMA. This requires first loading the pr-trained weights of GMA as proposed by GMA, freezing them, and training the GPU prediction and convex upsampling networks introduced by Splatflow. Then, all parameters are fine-tuned using dataset-specific finetuning procedures very similar to RAFT.

- **VideoFlow** (Shi et al., 2023a): They propose a multi-frame optical flow estimation method and use the same iterative flow refinement module as other methods in the RAFT family, specifically they use the SKBlocks from SKFlow. For feature extractors, they take inspiration from FlowFormer (Huang et al., 2022) and use ImageNet1k pre-trained Twins-SVT (Chu et al., 2021). They use three and five-image frames during training while following training procedures slightly similar to RAFT.

- **NeuFlow** (Zhang et al., 2024): Inspired by GMFlow, they use transformer-based blocks to implement global cross-attention, however, they use Flash Attention (Dao et al., 2022)

for slight speed improvements. They use a very similar upsampling module as GMFlow and RAFT. However, to obtain feature maps with finer details, they directly extract features from the original images using a CNN block, instead of using features for matching at the $\frac{1}{16}^{th}$ and $\frac{1}{8}^{th}$ scale like RAFT and GMFlow. They use a very similar training procedure as RAFT.

### B.4 FLOWFORMER FAMILY

Proposed by Huang et al. (2022), FlowFormer marks a significant shift in the architecture of optical flow estimation methods compared to the RAFT family.

- **FlowFormer** (Huang et al., 2022): It is a transformer-based neural network architecture for optical flow estimation. Huang et al. (2022) describe, FlowFormer tokenizes the 4D cost volume built from an image pair, encodes the cost tokens into a cost memory with alternate group transformer (AGT) layers in a latent space, and decodes the cost memory via a recurrent transformer decoder with dynamic positional cost queries. The two-stage Twins-SVT (Chu et al., 2021) feature extractor is pre-trained on the ImageNet1k dataset. After that, the training procedure of the entire FlowFormer is similar to RAFT's training procedure.

- **FlowFormer++** (Shi et al., 2023b): This is built upon FlowFormer to include Masked Cost Volume Autoencoding (MCVA) to improve the i.i.d. performance of FlowFormer by pre-training the cost-volume encoder with a mask encoding strategy proposed by them. Flow-Former++ requires significantly different pre-training, while the training and fine-tuning procedures are similar to RAFT.

## C DATASET DETAILS

FLOWBENCH supports a total of four distinct optical flow datasets. Following, we describe these datasets in detail.

### C.1 FLYINGTHINGS3D

This is a synthetic dataset proposed by Mayer et al. (2016) largely used for training and evaluation of optical flow estimation methods. This dataset consists of 25000 stereo frames, of everyday objects such as chairs, tables, cars, etc. flying around in 3D trajectories. The idea behind this dataset is to have a large volume of trajectories and random movements rather than focus on a real-world application. In their work, Dosovitskiy et al. (2015) showed models trained on FlyingThings3D can generalize to a certain extent to other datasets.

### C.2 KITTI2015

Proposed by Menze & Geiger (2015), this dataset is focused on the real-world driving scenario. It contains a total of 400 pairs of image frames, split equally for training and testing. The image frames were captured in the wild while driving around on the streets of various cities. The ground-truth labels were obtained by an automated process.

### C.3 MPI SINTEL

Proposed by Butler et al. (2012) and Wulff et al. (2012), this dataset is derived from an open-source animated short film and consists of a total of 1064 synthetic frames for training and 564 synthetic frames for testing, both at a resolution of $1024 \times 436$. The intention of this dataset is to enforce realism while having a dataset at scale. This dataset is provided as two datasets, which are passes with more transformations and effects on the frames that originally have constant albedo over time, these passes are,

- MPI Sintel (clean): This is the clean pass that adds some realism to the images by adding some spectral effects, like illumination, shadows, and smooth shading.

- MPI Sintel (final): This is the final pass that adds more realism by adding effects such as blur due to depth and camera focus, blur due to motion and atmospheric effects such as snow during snow storms, etc.

### C.4    SPRING

Similar to MPI Sintel, Mehl et al. (2023) proposed a new dataset and benchmark for optical flow estimation which is much larger than any other dataset before. It consists of frames from the open-source Blender movie "Spring" and consists of 6000 stereo image pairs from 47 sequences with SotA visual effects at full HD resolution ($1920 \times 1080$ pixels).

## D    IMPLEMENTATION DETAILS OF THE BENCHMARK

Following we provide details regarding the experiments done for creating the benchmark used in the analysis.

**Compute Resources.**    Most experiments were done on a single 40 GB NVIDIA Tesla V100 GPU each, however, MS-RAFT+, FlowFormer, and FlowFormer++ are more compute-intensive, and thus 80GB NVIDIA A100 GPUs or NVIDIA H100 were used for these models, a single GPU for each experiment.

**Datasets Used.**    Performing adversarial attacks and OOD robustness evaluations are very expensive and compute-intensive. Thus, performing evaluation using all model-dataset pairs is not possible given the limited computing resources at our disposal. Thus, for the benchmark, we only use KITTI2015, MPI Sintel (clean), and MPI Sintel (final) as these are the most commonly used datasets for evaluation (Ilg et al., 2017; Huang et al., 2022; Schmalfuss et al., 2022b; Schrodi et al., 2022; Agnihotri et al., 2024).

**Metrics Calculation.**    In Sec. 4 we introduce three new metrics for better understanding our analysis, given the large scale of the benchmark created. For calculating TARE and NARE values we used BIM, PGD, and CosPGD attack with step size $\alpha$=0.01, perturbation budget $\epsilon = \frac{8}{255}$ under the $\ell_\infty$-norm bound, as targeted and non-targeted attacks respectively. We use $\ell_\infty$-norm bound as we observe in Appendix H that there is a high correlation between the performance of optical flow estimation methods when attacked using $\ell_\infty$-norm bounded attacks and $\ell_2$-norm bounded attacks. We use 20 attack iterations for calculating TARE and NARE as we observe in *Appendix H*, that at a lower number of iterations, the gap in performance of different optical flow estimation methods is small, thus an in-depth analysis would be difficult, and we do not go beyond 20 attack iterations as computing each attack step for an adversarial attack is very expensive, and as shown by Agnihotri et al. (2024) and Schmalfuss et al. (2022b), 20 iterations are enough to optimize an attack to truly understand the performance of the attacked method. For calculating GAE, we use all 15 2D Common Corruptions: 'Gaussian Noise', Shot Noise', 'Impulse Noise', 'Defocus Blur', 'Frosted Glass Blur', 'Motion Blur', 'Zoom Blur', 'Snow', 'Frost', 'Fog', 'Brightness', 'Contrast', 'Elastic Transform', 'Pixelate', 'JPEG Compression', and eight 3D Common Corruptions: 'Color Quantization', 'Far Focus', 'Fog 3D', 'ISO Noise', 'Low Light', 'Near Focus', 'XY Motion Blur', and 'Z Motion Blur'. All the common corruptions are at severity 3. Kar et al. (2022) offers more 3D Common Corruptions, however computing them is resource intensive. Thus, given our limited resources and an overlap in the corruptions between 2D Common Corruptions and 3D Common Corruptions, we focus on generating 3D Common Corruptions that might be unique from their 2D counterpart, require fewer sources to generate, and are interesting from an optical flow estimation perspective.

**Calculating the EPE.**  $EPE$ is the Euclidean distance between the two vectors, where one vector is the predicted flow by the optical flow estimation method and the other vector is the ground truth in case of i.i.d. performance evaluations, non-targeted attacks evaluations, and OOD robustness evaluations, while it is the target flow vector, in case of targeted attacks. For each dataset, the $EPE$ value is calculated over all the samples of the evaluation set of the respective dataset and then the mean $EPE$ value is used as the mean-$EPE$ of the respective method over the respective dataset.    NEW

**Other Metrics.** Apart from EPE, FLOWBENCH also enables calculating a lot of other interesting metrics, such as $\ell_0$, $\ell_2$, $\ell_\infty$, distance between the perturbations of each image before and after a threat. Apart from these, in all scenarios, we also capture the outlier error, 1-px error, 3-px error, 5-px error and cosine distance between two vectors. These vectors are the same as that in the case of $EPE$ calculations. We limited the analysis in this work to use $EPE$, since it is the most commonly used metric for evaluation, moreover, most works on optical flow estimation (Agnihotri et al., 2024; Schmalfuss et al., 2022b; Schrodi et al., 2022; Teed & Deng, 2020; Jahedi et al., 2024b) show a very high correlation between performance evaluations using different metrics.

**Models Used.** All available checkpoints, as shown in Tab. 1 for MPI Sintel and KITTI2015 dataset were used for creating the benchmark, except the following four models: Separableflow, SCV, VCN, Unimatch as due to special operations used in these models, they required specific libraries which were creating conflicts with all the others models, and as most of these models are very old and do not have performance close to SotA performance, we did not include them.

**Adversarial Weather** For generating adversarial weather attacks, we followed the implementation proposed by Schmalfuss et al. (2023). However, generating this attack is highly compute-intensive, and thus doing so for all models was not possible. Thus, based on the performance and reliability of all the models, we identified a few (eight) interesting models and only attacked them using the four different attacks curtailed within adversarial weather. This was done to demonstrate the capability of FLOWBENCH to perform this attack. The following are the specifications for the weather attacks:

- Adversarial Weather: **Snow** (random snowflakes)
  - Number of Particles: 3000
  - Number of optimization steps: 750
- Adversarial Weather: **Rain** (rain streaks of length 0.15 with motion blur )
  - Number of Particles: 20
  - Number of optimization steps: 750
- Adversarial Weather: **Fog** (random large less opacity particles)
  - Number of Particles: 60
  - Number of optimization steps: 750
- Adversarial Weather: **Sparks** (random red sparks)
  - Number of Particles: 3000
  - Number of optimization steps: 750

Please note, that these specifications are identical to the optimal ones proposed by Schmalfuss et al. (2023).

# E DESCRIPTION OF FLOWBENCH

Following, we describe the benchmarking tool, FLOWBENCH. It is built using pltflow (Morimitsu, 2021), and supports 36 unique architectures and 4 distinct datasets, namely FlyingThings3D (Mayer et al., 2016), KITTI2015 (Menze & Geiger, 2015), MPI Sintel (Butler et al., 2012) (clean and final) and Spring (Mehl et al., 2023) datasets (please refer Appendix C for additional details on the datasets). It enables training and evaluations on all aforementioned datasets including evaluations using SotA adversarial attacks such as CosPGD (Agnihotri et al., 2024) and PCFA (Schmalfuss et al., 2022b), Adversarial weather (Schmalfuss et al., 2023), and other commonly used adversarial attacks like BIM (Kurakin et al., 2018), PGD (Kurakin et al., 2017), FGSM (Goodfellow et al., 2015), under various lipshitz ($l_p$) norm bounds.

Additionally, it enables evaluations for Out-of-Distribution (OOD) robustness by corrupting the inference samples using 2D Common Corruptions (Hendrycks & Dietterich, 2019) and 3D Common Corruptions (Kar et al., 2022).

We follow the nomenclature set by RobustBench (Croce et al., 2021) and use "threat_model" to define the kind of evaluation to be performed. When "threat_model" is defined to be "None", the evaluation is performed on unperturbed and unaltered images, if the "threat_model" is defined to be an adversarial attack, for example "PGD", "CosPGD" or "PCFA", then FLOWBENCH performs an adversarial attack using the user-defined parameters. We elaborate on this in Appendix E.1. Whereas, if "threat_model" is defined to be "2DCommonCorruptions" or "3DCommonCorruptions", the FLOWBENCH performs evaluations after perturbing the images with 2D Common Corruptions and 3D Common Corruptions respectively. We elaborate on this in Appendix E.2.

If the queried evaluation already exists in the benchmark provided by this work, then FLOWBENCH simply retrieves the evaluations, thus saving computation.

### E.1 ADVERSARIAL ATTACKS

FLOWBENCH enables the use of all the attacks mentioned in Sec. 2.3 to help users better study the reliability of their optical flow methods. We choose to specifically include these white-box adversarial attacks as they either serve as the common benchmark for adversarial attacks in classification literature (FGSM, BIM, PGD, APGD) or they are unique attacks proposed specifically for pixel-wise prediction tasks (CosPGD) and optical flow estimation (PCFA and Adversarial Weather). These attacks can either be *Non-targeted* which are designed to simply fool the model into making incorrect predictions, irrespective of what the model eventually predicts, or can be *Targeted*, where the model is fooled to make a certain prediction. Most attacks can be, designed to be either Targeted or Non-targeted, these include, FGSM, BIM, PGD, APGD, CosPGD and Adversarial Weather. However, by design, some attacks are limited to being only one of the two, for example, PCFA which is a targeted attack. Following, we discuss these attacks in detail and highlight their key differences.

**FGSM.** Assuming a non-targeted attack, given a model $f_\theta$ and an unperturbed input sample $\boldsymbol{X}^{\text{clean}}$ and ground truth label $\boldsymbol{Y}$, FGSM attack adds noise $\delta$ to $\boldsymbol{X}^{\text{clean}}$ as follows,

$$\boldsymbol{X}^{\text{adv}} = \boldsymbol{X}^{\text{clean}} + \alpha \cdot \text{sign} \nabla_{\boldsymbol{X}^{\text{clean}}} L(f_\theta(\boldsymbol{X}^{\text{clean}}), \boldsymbol{Y}), \tag{1}$$

$$\delta = \phi^\epsilon(\boldsymbol{X}^{\text{adv}} - \boldsymbol{X}^{\text{clean}}), \tag{2}$$

$$\boldsymbol{X}^{\text{adv}} = \phi^r(\boldsymbol{X}^{\text{clean}} + \delta). \tag{3}$$

Here, $L(\cdot)$ is the loss function (differentiable at least once) which calculates the loss between the model prediction and ground truth, $\boldsymbol{Y}$. $\alpha$ is a small value of $\epsilon$ that decides the size of the step to be taken in the direction of the gradient of the loss w.r.t. the input image, which leads to the input sample being perturbed such that the loss increases. $\boldsymbol{X}^{\text{adv}}$ is the adversarial sample obtained after perturbing $\boldsymbol{X}^{\text{clean}}$. To make sure that the perturbed sample is semantically indistinguishable from the unperturbed clean sample to the human eye, steps from Eq. (2) and Eq. (3) are performed. Here, function $\phi^\epsilon$ is clipping the $\delta$ in $\epsilon$-ball for $\ell_\infty$-norm bounded attacks or the $\epsilon$-projection in other $l_p$-norm bounded attacks, complying with the $\ell_\infty$-norm or other $l_p$-norm constraints, respectively. While function $\phi^r$ clips the perturbed sample ensuring that it is still within the valid input space. FGSM, as proposed, is a single step attack. For targeted attacks, $\boldsymbol{Y}$ is the target and $\alpha$ is multiplied by -1 so that a step is taken to minimize the loss between the model's prediction and the target prediction.

**BIM.** This is the direct extension of FGSM into an iterative attack method. In FGSM, $\boldsymbol{X}^{\text{clean}}$ was perturbed just once. While in BIM, $\boldsymbol{X}^{\text{clean}}$ is perturbed iteratively for time steps $t \in [0, \boldsymbol{T}]$, such that $t \in \mathbb{Z}^+$, where $\boldsymbol{T}$ are the total number of permissible attack iterations. This changes the steps of the attack from FGSM to the following,

$$\boldsymbol{X}^{\text{adv}_{t+1}} = \boldsymbol{X}^{\text{adv}_t} + \alpha \cdot \text{sign} \nabla_{\boldsymbol{X}^{\text{adv}_t}} L(f_\theta(\boldsymbol{X}^{\text{adv}_t}), \boldsymbol{Y}), \tag{4}$$

$$\delta = \phi^\epsilon(\boldsymbol{X}^{\text{adv}_{t+1}} - \boldsymbol{X}^{\text{clean}}), \tag{5}$$

$$\boldsymbol{X}^{\text{adv}_{t+1}} = \phi^r(\boldsymbol{X}^{\text{clean}} + \delta). \tag{6}$$

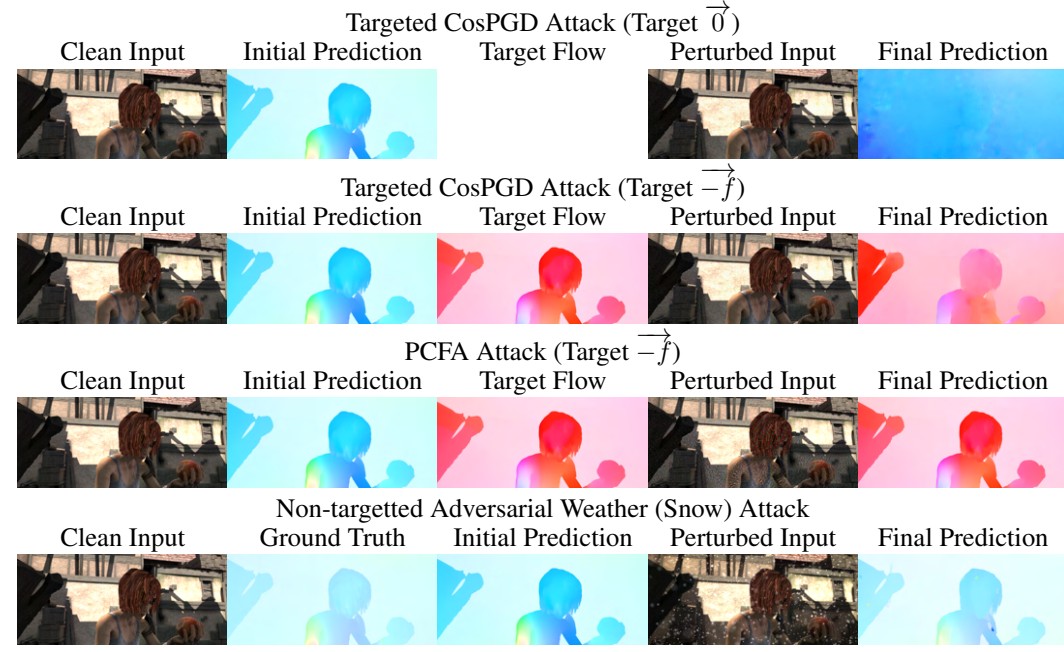

Figure 8: Examples of MPI Sintel images perturbed by the mentioned adversarial attacks and the optical flow predictions using FlowFormer++. These examples are intended to show the versatility of FLOWBENCH.

Here, at $t$=0, $\boldsymbol{X}^{\mathrm{adv}_t}=\boldsymbol{X}^{\mathrm{clean}}$.

**PGD.** Since in BIM, the initial prediction always started from $\boldsymbol{X}^{\mathrm{clean}}$, the attack required a significant amount of steps to optimize the adversarial noise and yet it was not guaranteed that in the permissible $\epsilon$-bound, $\boldsymbol{X}^{\mathrm{adv}_{t+1}}$ was far from $\boldsymbol{X}^{\mathrm{clean}}$. Thus, PGD proposed introducing stochasticity to ensure random starting points for attack optimization. They achieved this by perturbing $\boldsymbol{X}^{\mathrm{clean}}$ with $\mathcal{U}(-\epsilon, \epsilon)$, a uniform distribution in $[-\epsilon, \epsilon]$, before making the first prediction, such that, at $t$=0

$$\boldsymbol{X}^{adv_t} = \phi^r(\boldsymbol{X}^{clean} + \mathcal{U}(-\epsilon, \epsilon)). \tag{7}$$

**APGD.** Auto-PGD is an effective extension to the PGD attack that effectively scales the step size $\alpha$ over attack iterations considering the compute budget and the success rate of the attack.

**CosPGD.** All previously discussed attacks were proposed for the image classification task. Here, the input sample is a 2D image of resolution $\mathrm{H} \times \mathrm{W}$, where H and W are the height and width of the spatial resolution of the sample, respectively. Pixel-wise information is inconsequential for image classification. This led to the pixel-wise loss $\mathcal{L}(\cdot)$ being aggregated to $\mathrm{L}(\cdot)$, as follows,

$$L(f_\theta(\boldsymbol{X}^{\mathrm{adv}_t}), \boldsymbol{Y}) = \frac{1}{\mathrm{H} \times \mathrm{W}} \sum_{i \in \mathrm{H} \times \mathrm{W}} \mathcal{L}(f_\theta(\boldsymbol{X}^{\mathrm{adv}_t})_i, \boldsymbol{Y}_i). \tag{8}$$

This aggregation of $\mathcal{L}(\cdot)$ fails to account for pixel-wise information available in tasks other than image classification, such as pixel-wise prediction tasks like Optical Flow estimation. Thus, in their work Agnihotri et al. (2024) propose an effective extension of the PGD attack that takes pixel-wise information into account by scaling $\mathcal{L}(\cdot)$ by the alignment between the distribution of the predictions and the distributions of $\boldsymbol{Y}$ before aggregating leading to a better-optimized attack, modifying Eq. (4) as follows,

$$\boldsymbol{X}^{\mathrm{adv}_{t+1}} = \boldsymbol{X}^{\mathrm{adv}_t} + \alpha \cdot \mathrm{sign} \nabla_{\boldsymbol{X}^{\mathrm{adv}_t}} \sum_{i \in H \times W} \cos\left(\psi(f_\theta(\boldsymbol{X}^{\mathrm{adv}_t})_i), \Psi(\boldsymbol{Y}_i)\right) \cdot \mathcal{L}\left(f_\theta(\boldsymbol{X}^{\mathrm{adv}_t})_i, \boldsymbol{Y}_i\right).$$

$$\tag{9}$$

Where, functions $\psi$ and $\Psi$ are used to obtain the distribution over the predictions and $Y_i$, respectively, and the function $\cos$ calculates the cosine similarity between the two distributions. CosPGD is the unified SotA adversarial attack for pixel-wise prediction tasks.

**PCFA.** Recently proposed by Schmalfuss et al. (2022b), is the SotA targeted adversarial attack specifically designed for optical flow estimation. It optimizes the input perturbation $\delta = X^{\mathrm{adv}_t} - X^{\mathrm{clean}}$ within a given $l_2$ bound to obtain a given target flow $Y^{\mathrm{targ}}$. Mathematically, PCFA transforms the constrained optimization problem to find the most destructive perturbation under an $l_2$ constraint $\varepsilon_2$ into an unconstrained optimization problem by adding a term that penalizes deviations from the $l_2$ constraint:

$$X^{\mathrm{adv}_{t+1}} = X^{\mathrm{adv}_t} + \underset{\hat{\delta}}{\mathrm{argmin}}(\mathcal{L}(f_\theta(X^{\mathrm{adv}_t}), Y^{\mathrm{targ}}) + \mu \cdot \mathrm{ReLU}(\|\hat{\delta}\|_2^2 - (\varepsilon_2\sqrt{2 \times H \times W})^2)) \quad (10)$$

Here, $\mathcal{L}(\cdot)$ is a generic loss function, like EPE or cosine distance. The penalty scaling parameter $\mu$ influences how severely deviations from the per-pixel $l_2$ bound $\varepsilon_2$ are penalized. The optimization problem $\underset{\hat{\delta}}{\mathrm{argmin}}(\cdot)$ is solved with an L-BFGS optimizer.

**Adversarial Weather.** Unlike the previous attacks which introduced per-pixel modifications, adversarial weather Schmalfuss et al. (2023; 2022a) attacks optical flow methods through optimizing the motion trajectories of rendered weather particles $\mathcal{P}$ like snow flakes, rain drops or fog clouds. The particle trajectories are modelled as positions $P = \{P_1, P_2\}$ in the two frames $I_1, I_2$. Consequently, $X^{\mathrm{adv}}(P)$ is generated by differentiably rendering the particles with their respective 3D positions to the 2D images. The update step optimizes the particle positions to achieve a certain target flow $Y^{\mathrm{targ}}$ while simultaneously limiting the position offset size $\delta_{P^t} = P^{\mathrm{init}} - P^t$:

$$X^{\mathrm{adv}}(P^{t+1}) = X^{\mathrm{adv}}\left(P^t + \alpha \cdot \nabla_{P^t}\left(\mathrm{EPE}(f_\theta(X^{\mathrm{adv}}(P^t)), Y^{\mathrm{targ}}) + \sum_{I \in 1,2} \frac{\beta_I}{|\mathcal{P}|} \sum_{j \in \mathcal{P}} \frac{\|\delta_{P_I^t}^j\|_2^2}{d_I^j}\right)\right). \quad (11)$$

Here, $\beta$ balances the two optimization goals of reaching the target flow and limiting trajectory offsets. The allowed trajectory offsets are further scaled with the particle depth $d$ in the scene, to generate visually pleasing results.

Fig. 8, shows adversarial examples created using the SotA attacks and how they affect the model predictions.

E.2 OUT-OF-DISTRIBUTION ROBUSTNESS

While adversarial attacks help explore vulnerabilities of inefficient feature representations learned by a model, another important aspect of reliability is generalization ability. Especially, generalization to previously unseen samples or samples from significantly shifted distributions compared to the distribution of the samples seen while learning model parameters. As one cannot cover all possible scenarios during model training, a certain degree of generalization ability is expected from models. However, multiple works (Hendrycks & Dietterich, 2019; Kar et al., 2022; Hoffmann et al., 2021) showed that models are surprisingly less robust to distribution shifts, even those that can be caused by commonly occurring phenomena such as weather changes, lighting changes, etc. This makes the study of Out-of-Distribution (OOD) robustness an interesting avenue for research. Thus, to facilitate the study of robustness to such commonly occurring corruptions, FLOWBENCH enables evaluating against prominent image corruption methods. Following, we describe these methods in detail.

**2D Common Corruptions.** Hendrycks & Dietterich (2019) propose introducing distribution shift in the input samples by perturbing images with a total of 15 synthetic corruptions that could occur in the real world. These corruptions include weather phenomena such as fog, and frost, digital corruptions such as jpeg compression, pixelation, and different kinds of blurs like motion, and zoom blur, and noise corruptions such as Gaussian and shot noise amongst others corruption types. Each of these corruptions can perturb the image at 5 different severity levels between 1 and 5. The final performance of the model is the mean of the model's performance on all the corruptions, such that

| Color Quantization | Far Focus | Fog 3D | ISO Noise |
|---|---|---|---|

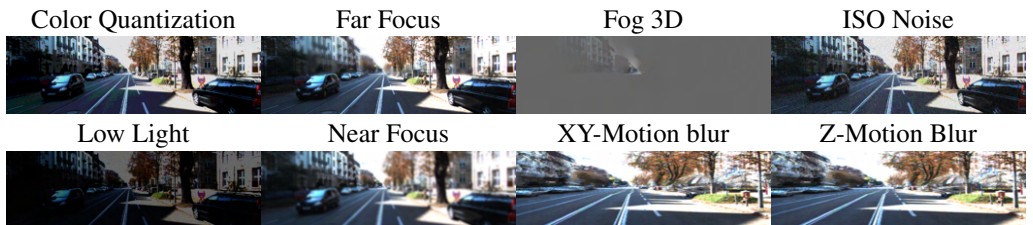

| Low Light | Near Focus | XY-Motion blur | Z-Motion Blur |
|---|---|---|---|

Figure 9: Examples of images from KITTI2015 corrupted using 3D Common Corruptions for evaluation of OOD robustness.

every corruption is used to perturb each image in the evaluation dataset. Since these corruptions are applied to a 2D image, they are collectively termed 2D Common Corruptions.

**3D Common Corruptions.** Since the real world is 3D, Kar et al. (2022) extend 2D Common Corruptions to formulate more realistic-looking corruptions by leveraging depth information (synthetic depth information when real depth is not readily available) and luminescence angles. They name these image corruptions as 3D Common Corruptions. Fig. 9, shows examples of KITTI2015 images corrupted using 3D Common Corruptions.

## F MODEL ZOO

The trained checkpoints for all models available in FLOWBENCH can be obtained using the following lines of code:

```
from flowbench.evals import load_model
model = load_model(model_name='RAFT', dataset='KITTI2015')
```

Each model checkpoint can be retrieved with the pair of 'model_name', the name of the model, and 'dataset', the dataset for which the checkpoint was last fine-tuned. In Table 1, we provide a comprehensive look-up table for all 'model_name' and 'dataset' pairs for which trained checkpoints are available in FlowBench.                                                                    NEW

## G FLOWBENCH USAGE DETAILS

Following we provide a detailed description of the evaluation functions and their arguments provided in FlowBench.

### G.1 ADVERSARIAL ATTACKS

To evaluate a model for a given dataset, on an attack, the following lines of code are required.

```
from flowbench.evals import evaluate
model, results = evaluate(model_name='RAFT', dataset='KITTI2015',
                    threat_model='CosPGD', iterations=20, alpha=0.01,
                    epsilon=8/255, lp_norm='Linf', targeted=True,
                    optim_wrt='ground_truth', retrieve_existing=True)
```

The argument description is as follows:

- 'model_name' is the name of the optical flow estimation method to be used, given as a string.
- 'dataset' is the name of the dataset to be used also given as a string.
- 'threat_model' is the name of the adversarial attack to be used, given as a string.
- 'iterations' are the number of attack iterations, given as an integer.

Table 1: Overview of all available model checkpoints (model X, trained for dataset Y) in FLOWBENCH.

| Model | Dataset | | | Method Family | Time |
|---|---|---|---|---|---|
| | FlyingThings3D (Mayer et al., 2016) | KITTI2015 (Menze & Geiger, 2015) | MPI Sintel (Butler et al., 2012) | | |
| CCMR (Jahedi et al., 2024a) | ✗ | ✓ | ✓ | RAFT | January 2024 |
| CRAFT (Sui et al., 2022) | ✓ | ✓ | ✓ | RAFT | March 2022 |
| CSFlow (Shi et al., 2022) | ✓ | ✓ | ✗ | RAFT | February 2022 |
| DICL (Wang et al., 2020) | ✓ | ✓ | ✓ | PWC | October 2020 |
| DIP (Zheng et al., 2022) | ✓ | ✓ | ✓ | Deep Inverse Patchmatch | April 2022 |
| FastFlowNet (Kong et al., 2021) | ✓ | ✓ | ✓ | PWC | March 2021 |
| Flow1D (Xu et al., 2021a) | ✓ | ✓ | ✓ | RAFT | April 2021 |
| FlowFormer (Huang et al., 2022) | ✓ | ✓ | ✓ | FlowFormer | March 2022 |
| FlowFormer++ (Shi et al., 2023b) | ✓ | ✓ | ✓ | FlowFormer | March 2023 |
| FlowNet2.0 (Ilg et al., 2017) | ✓ | ✗ | ✗ | FlowNet | December 2016 |
| GMA (Jiang et al., 2021a) | ✓ | ✓ | ✓ | RAFT | April 2021 |
| GMFlow (Xu et al., 2022) | ✓ | ✓ | ✓ | RAFT | November 2021 |
| GMFlowNet (Zhao et al., 2022) | ✓ | ✓ | ✓ | RAFT | March 2022 |
| HD3 (Yin et al., 2019) | ✓ | ✓ | ✓ | PWC | December 2018 |
| IRR (Hur & Roth, 2019) | ✓ | ✓ | ✓ | PWC | April 2019 |
| LCV (Khairi et al., 2024) | ✓ | ✗ | ✗ | RAFT | July 2020 |
| LiteFlowNet (Hui et al., 2018) | ✓ | ✓ | ✓ | FlowNet | May 2018 |
| LiteFlowNet2 (Hui et al., 2020) | ✗ | ✓ | ✓ | FlowNet | February 2020 |
| LiteFlowNet3 (Hui & Loy, 2020) | ✗ | ✓ | ✓ | FlowNet | July 2020 |
| LLA-Flow (Xu et al., 2023b) | ✓ | ✓ | ✓ | RAFT | April 2023 |
| MaskFlowNetS (Zhao et al., 2020) | ✓ | ✗ | ✓ | PWC | March 2023 |
| MaskFlowNet (Zhao et al., 2020) | ✗ | ✓ | ✓ | PWC | March 2023 |
| MS-RAFT+ (Jahedi et al., 2024b) | ✓ | ✓ | ✓ | RAFT | October 2022 |
| MatchFlow (Dong et al., 2023) | ✓ | ✓ | ✓ | RAFT | March 2023 |
| NeuFlow (Zhang et al., 2024) | ✗ | ✗ | ✓ | FlowNet | March 2024 |
| PWC-Net (Sun et al., 2018) | ✓ | ✗ | ✓ | PWC | September 2017 |
| RapidFlow (Morimitsu et al., 2024a) | ✓ | ✓ | ✓ | RAFT | May 2024 |
| RAFT (Teed & Deng, 2020) | ✓ | ✓ | ✓ | RAFT | March 2020 |
| RPKNet (Morimitsu et al., 2024b) | ✓ | ✓ | ✓ | RAFT | March 2024 |
| ScopeFlow (Bar-Haim & Wolf, 2020) | ✓ | ✓ | ✓ | PWC | February 2020 |
| SCV (Jiang et al., 2021b) | ✓ | ✓ | ✓ | RAFT | April 2021 |
| SeparableFlow (Zhang et al., 2021) | ✓ | ✓ | ✓ | RAFT | October 2021 |
| SKFlow (Sun et al., 2022) | ✓ | ✓ | ✓ | RAFT | November 2022 |
| SplatFlow (Wang et al., 2024) | ✗ | ✓ | ✗ | RAFT | January, 2024 |
| StarFlow (Godet et al., 2021) | ✓ | ✓ | ✓ | SSRFlow | July 2020 |
| Unimatch (Xu et al., 2023a) | ✓ | ✗ | ✗ | RAFT | November 2022 |
| VCN (Yang & Ramanan, 2019) | ✓ | ✗ | ✗ | PWC | December 2019 |
| VideoFlow (Shi et al., 2023a) | ✓ | ✓ | ✓ | RAFT | March 2023 |

- 'epsilon' is the permissible perturbation budget $\epsilon$ given a floating point (float).

- 'alpha' is the step size of the attack, $\alpha$, given as a floating point (float).

- 'lp_norm' is the Lipschitz continuity norm ($l_p$-norm) to be used for bounding the perturbation, possible options are 'Linf' and 'L2' given as a string.

- 'targeted' is a boolean flag that decides if the attack must be targeted or not. If targeted='True', then by default the target is $\overrightarrow{0}$, passed as target='zero', this can be changed to negative of the initial flow by passing target='negative'.

- 'optim_wrt' decides wrt what attack should be optimized, available choices are 'ground_truth' and 'initial_flow' as string. Please note, this only works well with attacks that utilize Eq. (7).

- 'retrieve_existing' is a boolean flag, which when set to 'True' will retrieve the evaluation from the benchmark if the queried evaluation exists in the benchmark provided by this work, else FLOWBENCH will perform the evaluation. If the 'retrieve_existing' boolean flag is set to 'False' then FLOWBENCH will perform the evaluation even if the queried evaluation exists in the provided benchmark.

## G.2 ADVERSARIAL WEATHER

As an attack, adversarial weather works slightly different compared to other adversarial attacks, thus we additionally mention the commands for using adversarial weather.

```
from flowbench.evals import evaluate
model, results = evaluate(model_name='RAFT', dataset='KITTI2015',
                threat_model='Adversarial_Weather', weather='snow',
                num_particles=10000, targeted=True,
                retrieve_existing=True)
```

The argument description is as follows:

- 'model_name' is the name of the optical flow estimation method to be used, given as a string.
- 'dataset' is the name of the dataset to be used also given as a string.
- 'threat_model' is the name of the adversarial attack to be used, given as a string.
- 'weather' is the name of the weather condition in adversarial weather attack to be used, given as a string, options include 'snow', 'fog', 'rain' and 'sparks'.
- 'num_particles' is the number of particles per frame to be used, given as a integer.
- 'targeted' is a boolean flag that decides if the attack must be targeted or not. If targeted='True', then by default the target is $\overrightarrow{0}$, passed as target='zero', this can be changed to negative of the initial flow by passing target='negative'.
- 'optim_wrt' decides wrt what attack should be optimized, available choices are 'ground_truth' and 'initial_flow' as string. Please note, this only works well with attacks that utilize Eq. (7).
- 'retrieve_existing' is a boolean flag, which when set to 'True' will retrieve the evaluation from the benchmark if the queried evaluation exists in the benchmark provided by this work, else FLOWBENCH will perform the evaluation. If the 'retrieve_existing' boolean flag is set to 'False' then FLOWBENCH will perform the evaluation even if the queried evaluation exists in the provided benchmark.

### G.3  2D COMMON CORRUPTIONS

To evaluate a model for a given dataset, with 2D Common Corruptions, the following lines of code are required.

```
from flowbench.evals import evaluate
model, results = evaluate(model_name='RAFT', dataset='KITTI2015',
                threat_model='2DCommonCorruption',
                severity=3, retrieve_existing=True)
```

The argument description is as follows:

- 'model_name' is the name of the optical flow estimation method to be used, given as a string.
- 'dataset' is the name of the dataset to be used also given as a string.
- 'threat_model' is the name of the common corruption to be used, given as a string.
- 'severity' is the severity of the corruption, given as an integer between 1 and 5 (both inclusive).
- 'retrieve_existing' is a boolean flag, which when set to 'True' will retrieve the evaluation from the benchmark if the queried evaluation exists in the benchmark provided by this work, else FLOWBENCH will perform the evaluation. If the 'retrieve_existing' boolean flag is set to 'False' then FLOWBENCH will perform the evaluation even if the queried evaluation exists in the provided benchmark.

FLOWBENCH supports the following 2D Common Corruption: 'gaussian_noise', shot_noise', 'impulse_noise', 'defocus_blur', 'frosted_glass_blur', 'motion_blur', 'zoom_blur', 'snow', 'frost', 'fog', 'brightness', 'contrast', 'elastic', 'pixelate', 'jpeg'. For the evaluation, FLOWBENCH will evaluate the model on the validation images from the respective dataset corrupted using each of the aforementioned corruptions for the given severity, and then report the mean performance over all of them.

### G.4  3D COMMON CORRUPTIONS

To evaluate a model for a given dataset, with 3D Common Corruptions, the following lines of code are required.

```
from flowbench.evals import evaluate
model, results = evaluate(model_name='RAFT', dataset='KITTI2015',
```

```
                            threat_model='3DCommonCorruption',
                            severity=3, retrieve_existing=True)
```

The argument description is as follows:

- 'model_name' is the name of the optical flow estimation method to be used, given as a string.

- 'dataset' is the name of the dataset to be used also given as a string.

- 'threat_model' is the name of the common corruption to be used, given as a string.

- 'severity' is the severity of the corruption, given as an integer between 1 and 5 (both inclusive).

- 'retrieve_existing' is a boolean flag, which when set to 'True' will retrieve the evaluation from the benchmark if the queried evaluation exists in the benchmark provided by this work, else FLOWBENCH will perform the evaluation. If the 'retrieve_existing' boolean flag is set to 'False' then FLOWBENCH will perform the evaluation even if the queried evaluation exists in the provided benchmark.

FLOWBENCH supports the following 3D Common Corruption: 'color_quant', 'far_focus', 'fog_3d', 'iso_noise', 'low_light', 'near_focus', 'xy_motion_blur', and 'z_motion_blur'. For the evaluation, FLOWBENCH will evaluate the model on the validation images from the respective dataset corrupted using each of the aforementioned corruptions for the given severity, and then report the mean performance over all of them.

## H  ADDITIONAL RESULTS

Following we include additional results from the benchmark made using FLOWBENCH.

### H.1  ADVERSARIAL ATTACKS

Here we report additional results for all adversarial attacks.

#### H.1.1  FGSM ATTACK

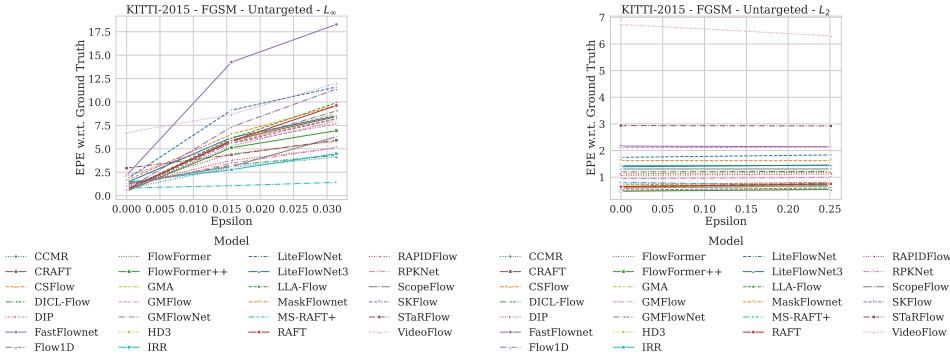

Figure 10: Evaluations for non-targeted FGSM attack under $\ell_\infty$-norm bound using the KITTI2015 dataset. The attack was optimized w.r.t. the ground truth predictions.

Here we report the evaluations using FGSM attack, both as targeted (both targets: $\overrightarrow{0}$ and $\overrightarrow{-f}$) and non-targeted attacks optimized under the $\ell_\infty$-norm bound and the $\ell_2$-norm bound. For $\ell_\infty$-norm bound, perturbation budget $\epsilon = \frac{8}{255}$, while for $\ell_2$-norm bound, perturbation budget $\epsilon = \frac{64}{255}$.

Attack evaluations include Fig. 10, Fig. 11, Fig. 12, Fig. 13, Fig. 14, Fig. 15, Fig. 16, Fig. 17, Fig. 18, Fig. 19, Fig. 20, Fig. 21, Fig. 22, Fig. 23, and Fig. 24.

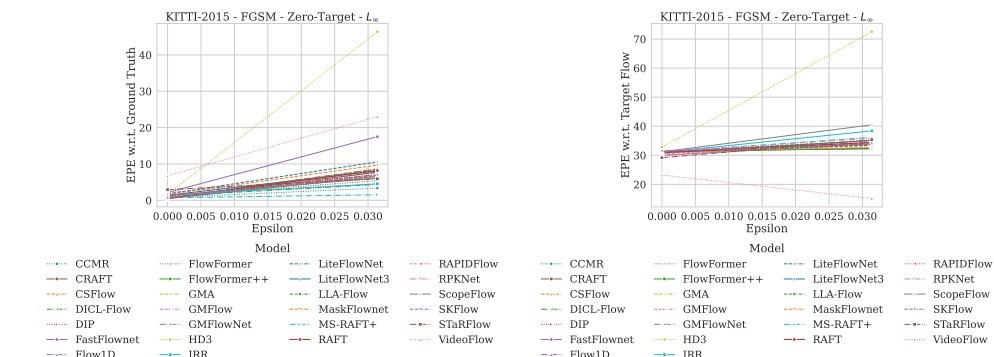

Figure 11: Evaluations for targeted FGSM attack with target $\overrightarrow{0}$ under $\ell_\infty$-norm bound using the KITTI2015 dataset. The attack was optimized w.r.t. the ground truth predictions.

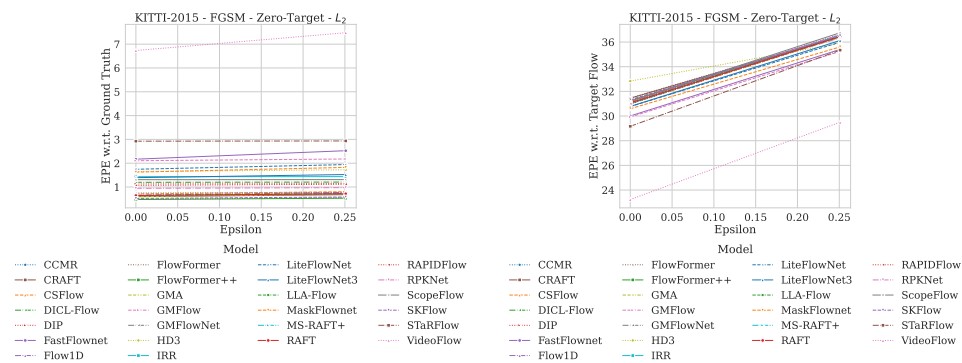

Figure 12: Evaluations for targeted FGSM attack with target $\overrightarrow{0}$ under $\ell_2$-norm bound using the KITTI2015 dataset. The attack was optimized w.r.t. the ground truth predictions.

### H.1.2 BIM ATTACK

Here we report the evaluations using BIM attack, both as targeted (both targets: $\overrightarrow{0}$ and $\overrightarrow{-f}$) and non-targeted attacks optimized under the $\ell_\infty$-norm bound and the $\ell_2$-norm bound over multiple attack iterations. For $\ell_\infty$-norm bound, perturbation budget $\epsilon = \frac{8}{255}$, and step size $\alpha$=0.01, while for $\ell_2$-norm bound, perturbation budget $\epsilon = \frac{64}{255}$ and step size $\alpha$=0.1. Attack evaluations include Fig. 25, Fig. 26, Fig. 27, Fig. 28, Fig. 29, Fig. 30, Fig. 31, Fig. 32, Fig. 33, Fig. 34, Fig. 35, Fig. 36, Fig. 37, Fig. 38, and Fig. 39.

### H.1.3 PGD ATTACK

Here we report the evaluations using PGD attack, both as targeted (both targets: $\overrightarrow{0}$ and $\overrightarrow{-f}$) and non-targeted attacks optimized under the $\ell_\infty$-norm bound and the $\ell_2$-norm bound over multiple attack iterations. For $\ell_\infty$-norm bound, perturbation budget $\epsilon = \frac{8}{255}$, and step size $\alpha$=0.01, while for $\ell_2$-norm bound, perturbation budget $\epsilon = \frac{64}{255}$ and step size $\alpha$=0.1. Attack evaluations include Fig. 40, Fig. 41, Fig. 42, Fig. 43, Fig. 44, Fig. 45, Fig. 46, Fig. 47, Fig. 48, Fig. 49, Fig. 50, Fig. 51, Fig. 52, Fig. 53, and Fig. 54.

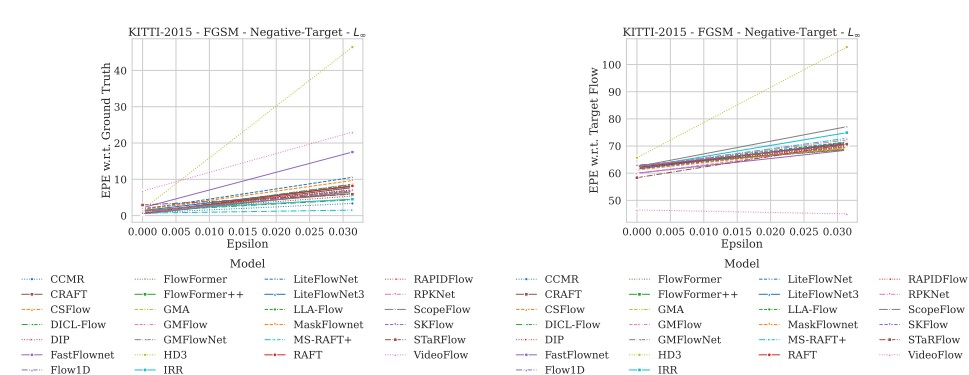

Figure 13: Evaluations for targeted FGSM attack with target $-\vec{f}$ under $\ell_\infty$-norm bound using the KITTI2015 dataset. The attack was optimized w.r.t. the ground truth predictions.

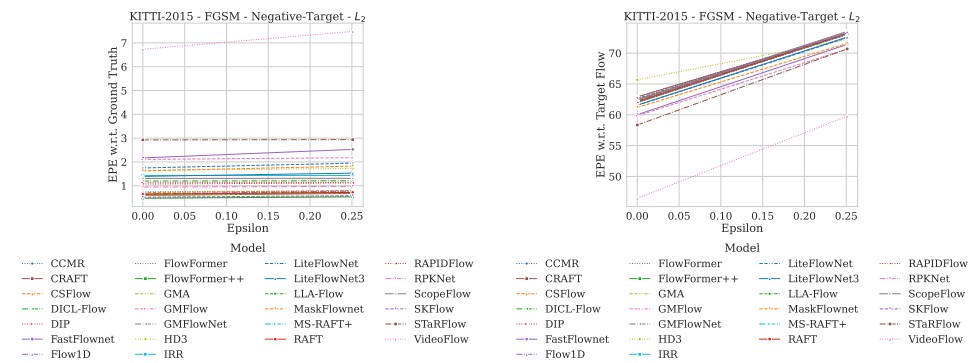

Figure 14: Evaluations for targeted FGSM attack with target $-\vec{f}$ under $\ell_2$-norm bound using the KITTI2015 dataset. The attack was optimized w.r.t. the ground truth predictions.

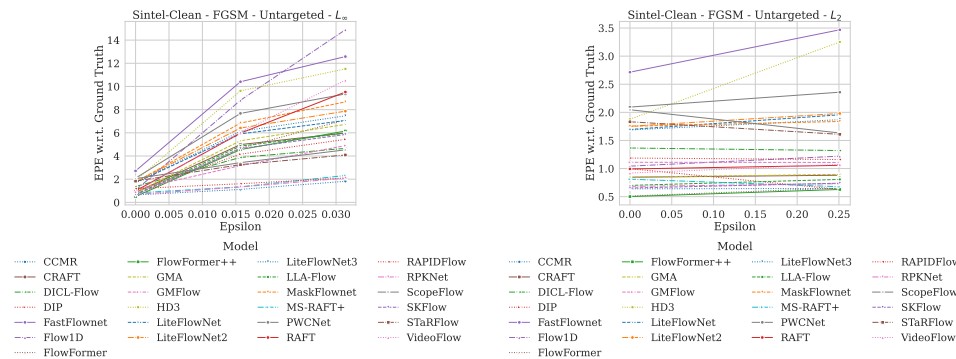

Figure 15: Evaluations for non-targeted FGSM attack under $\ell_\infty$-norm bound using the MPI Sintel (clean) dataset. The attack was optimized w.r.t. the ground truth predictions.

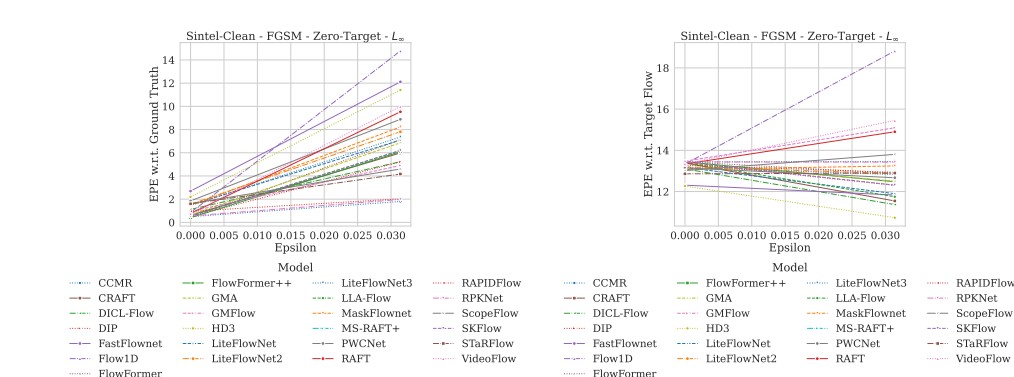

Figure 16: Evaluations for targeted FGSM attack with target $\overrightarrow{0}$ under $\ell_\infty$-norm bound using the MPI Sintel (clean) dataset. The attack was optimized w.r.t. the ground truth predictions.

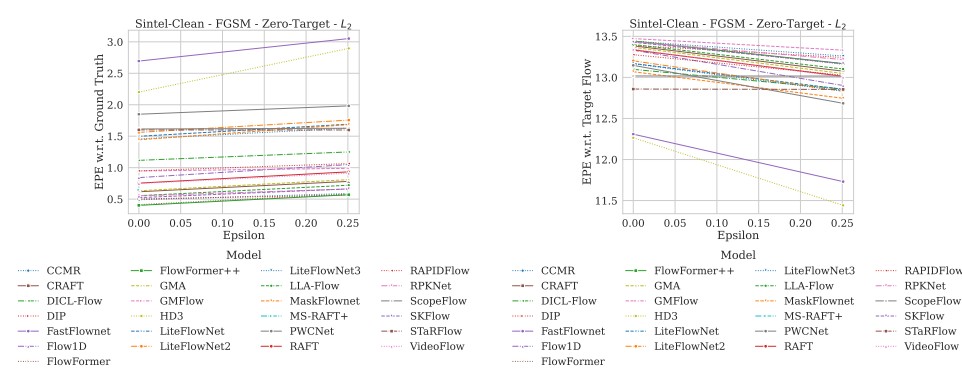

Figure 17: Evaluations for targeted FGSM attack with target $\overrightarrow{0}$ under $\ell_2$-norm bound using the MPI Sintel (clean) dataset. The attack was optimized w.r.t. the ground truth predictions.

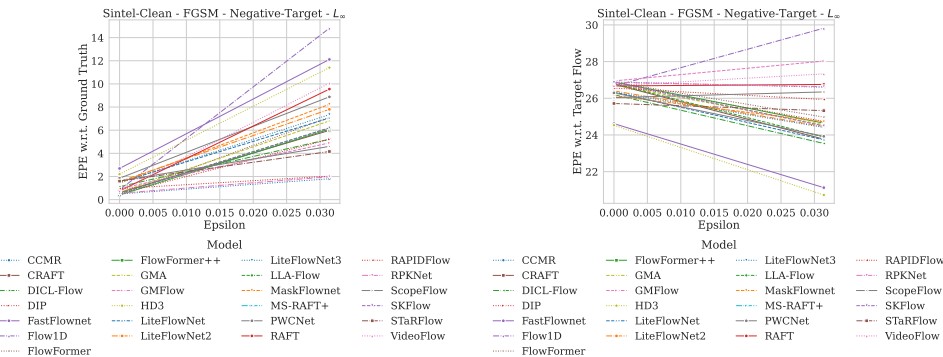

Figure 18: Evaluations for targeted FGSM attack with target $\overrightarrow{-f}$ under $\ell_\infty$-norm bound using the MPI Sintel (clean) dataset. The attack was optimized w.r.t. the ground truth predictions.

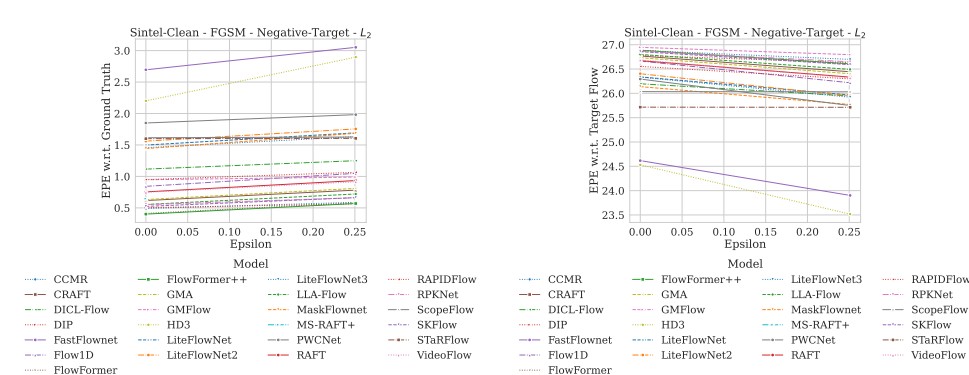

Figure 19: Evaluations for targeted FGSM attack with target $-\vec{f}$ under $\ell_2$-norm bound using the MPI Sintel (clean) dataset. The attack was optimized w.r.t. the ground truth predictions.

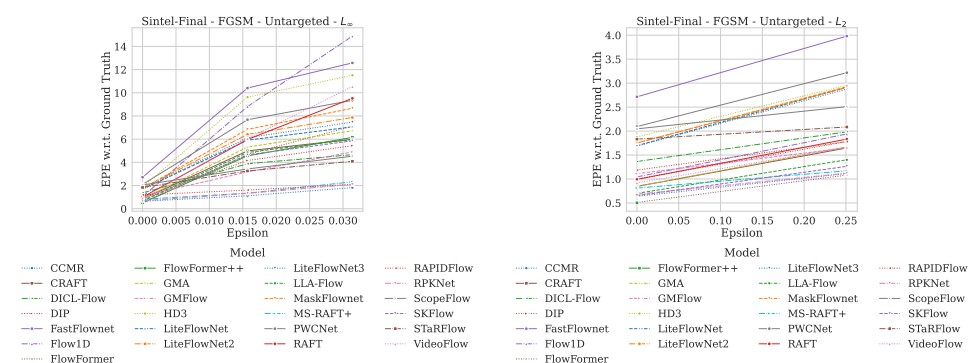

Figure 20: Evaluations for non-targeted FGSM attack under $\ell_\infty$-norm bound using the MPI Sintel (final) dataset. The attack was optimized w.r.t. the ground truth predictions.

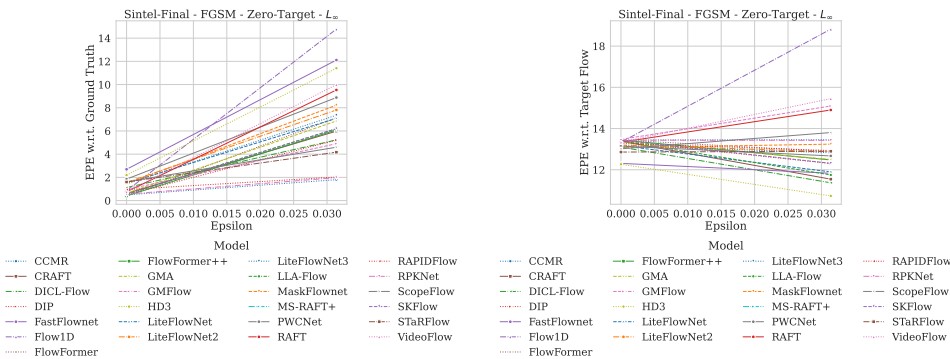

Figure 21: Evaluations for targeted FGSM attack with target $\vec{0}$ under $\ell_\infty$-norm bound using the MPI Sintel (final) dataset. The attack was optimized w.r.t. the ground truth predictions.

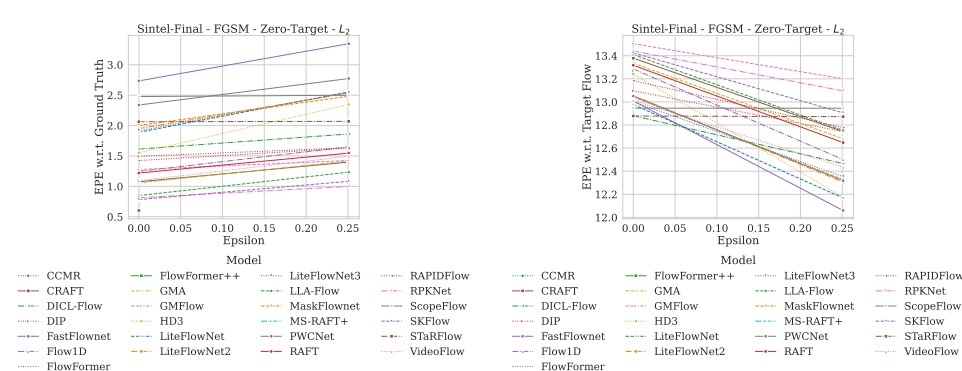

Figure 22: Evaluations for targeted FGSM attack with target $\overrightarrow{0}$ under $\ell_2$-norm bound using the MPI Sintel (final) dataset. The attack was optimized w.r.t. the ground truth predictions.

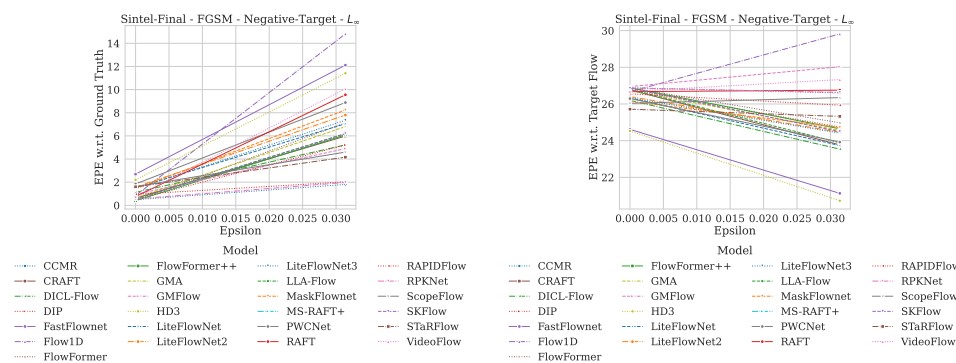

Figure 23: Evaluations for targeted FGSM attack with target $\overrightarrow{-f}$ under $\ell_\infty$-norm bound using the MPI Sintel (final) dataset. The attack was optimized w.r.t. the ground truth predictions.

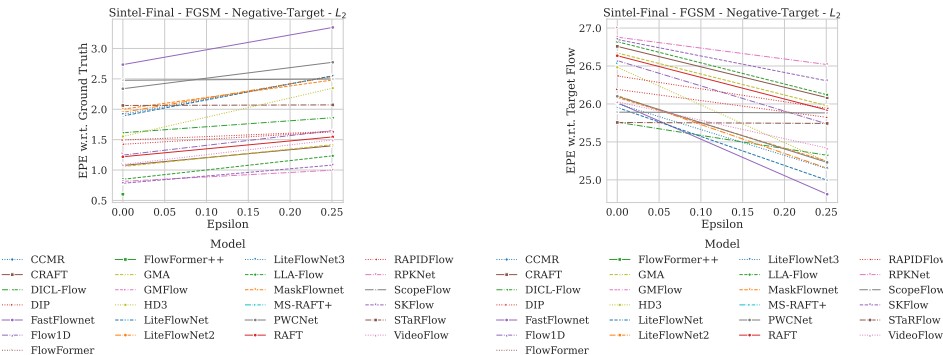

Figure 24: Evaluations for targeted FGSM attack with target $\overrightarrow{-f}$ under $\ell_2$-norm bound using the MPI Sintel (final) dataset. The attack was optimized w.r.t. the ground truth predictions.

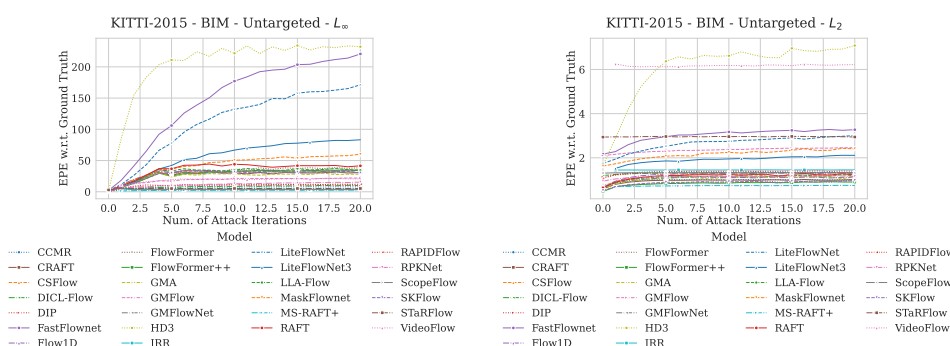

Figure 25: Evaluations for non-targeted BIM attack under $\ell_\infty$-norm bound using the KITTI2015 dataset. The attack was optimized w.r.t. the ground truth predictions.

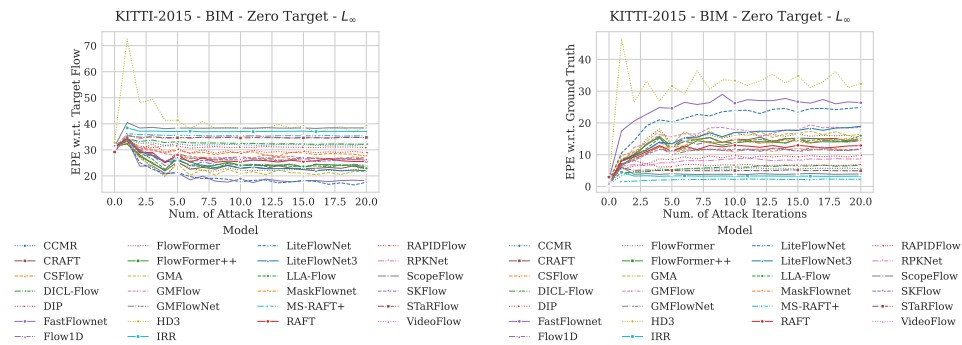

Figure 26: Evaluations for targeted BIM attack with target $\overrightarrow{0}$ under $\ell_\infty$-norm bound using the KITTI2015 dataset. The attack was optimized w.r.t. the ground truth predictions.

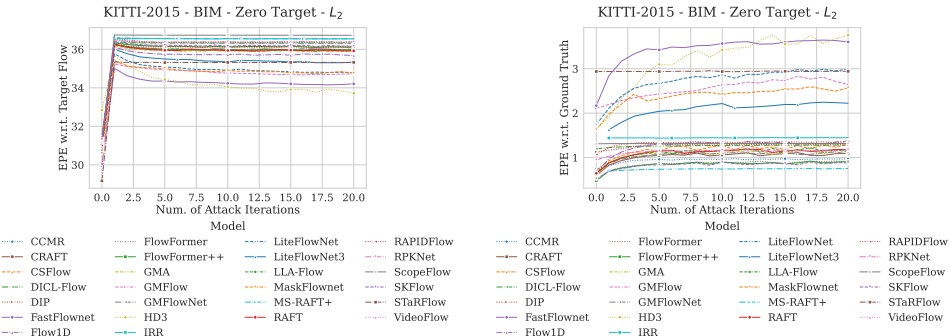

Figure 27: Evaluations for targeted BIM attack with target $\overrightarrow{0}$ under $\ell_2$-norm bound using the KITTI2015 dataset. The attack was optimized w.r.t. the ground truth predictions.

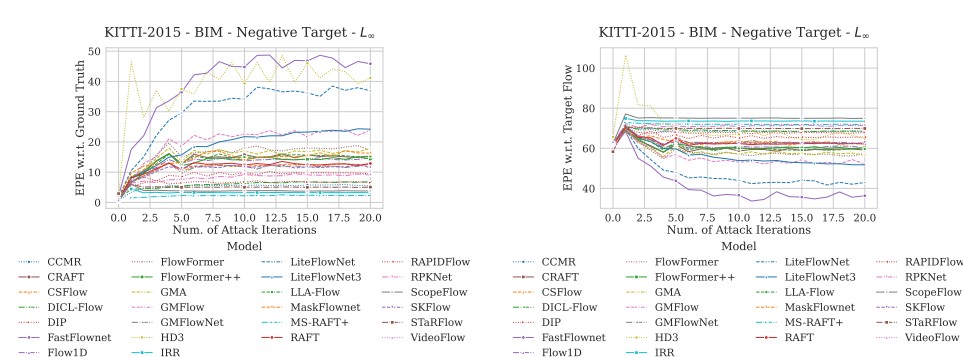

Figure 28: Evaluations for targeted BIM attack with target $-\overrightarrow{f}$ under $\ell_\infty$-norm bound using the KITTI2015 dataset. The attack was optimized w.r.t. the ground truth predictions.

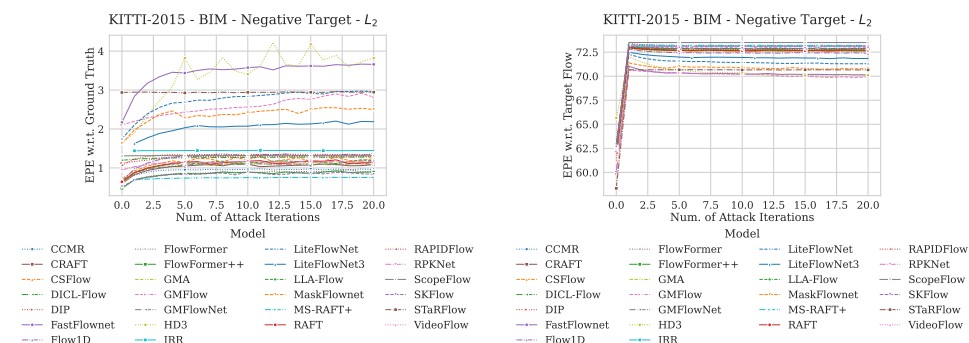

Figure 29: Evaluations for targeted BIM attack with target $-\overrightarrow{f}$ under $\ell_2$-norm bound using the KITTI2015 dataset. The attack was optimized w.r.t. the ground truth predictions.

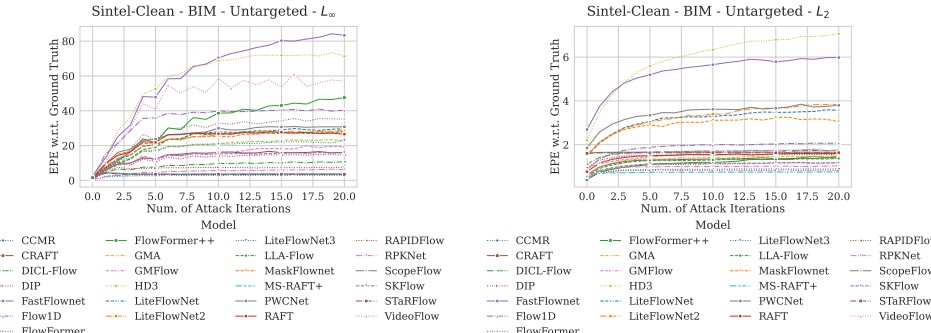

Figure 30: Evaluations for non-targeted BIM attack under $\ell_\infty$-norm bound using the MPI Sintel (clean) dataset. The attack was optimized w.r.t. the ground truth predictions.

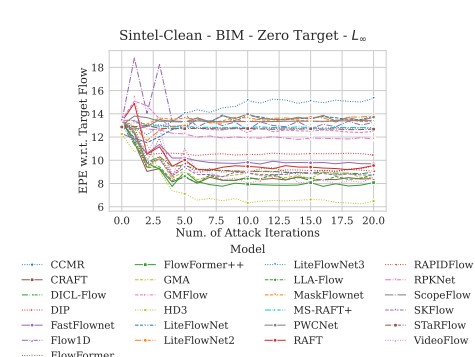

Figure 31: Evaluations for targeted BIM attack with target $\vec{0}$ under $\ell_\infty$-norm bound using the MPI Sintel (clean) dataset. The attack was optimized w.r.t. the ground truth predictions.

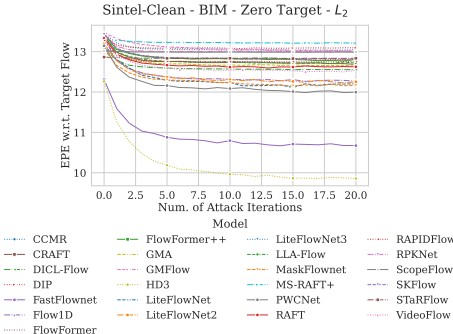

Figure 32: Evaluations for targeted BIM attack with target $\vec{0}$ under $\ell_2$-norm bound using the MPI Sintel (clean) dataset. The attack was optimized w.r.t. the ground truth predictions.

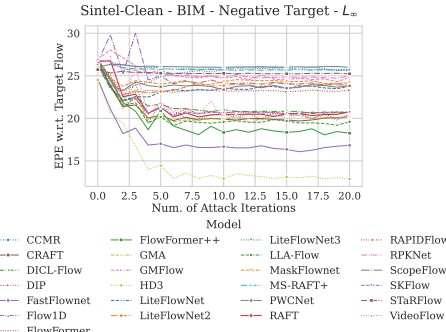

Figure 33: Evaluations for targeted BIM attack with target $-\vec{f}$ under $\ell_\infty$-norm bound using the MPI Sintel (clean) dataset. The attack was optimized w.r.t. the ground truth predictions.

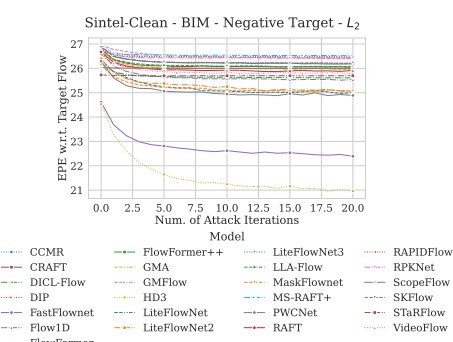

Figure 34: Evaluations for targeted BIM attack with target $-\overrightarrow{f}$ under $\ell_2$-norm bound using the MPI Sintel (clean) dataset. The attack was optimized w.r.t. the ground truth predictions.

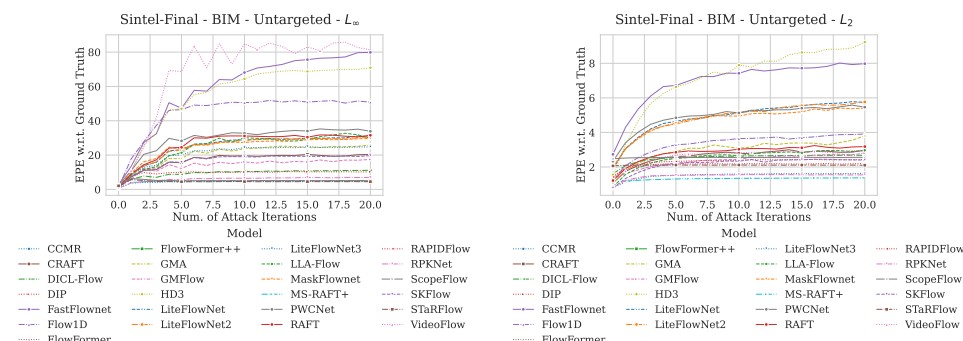

Figure 35: Evaluations for non-targeted BIM attack under $\ell_\infty$-norm bound using the MPI Sintel (final) dataset. The attack was optimized w.r.t. the ground truth predictions.

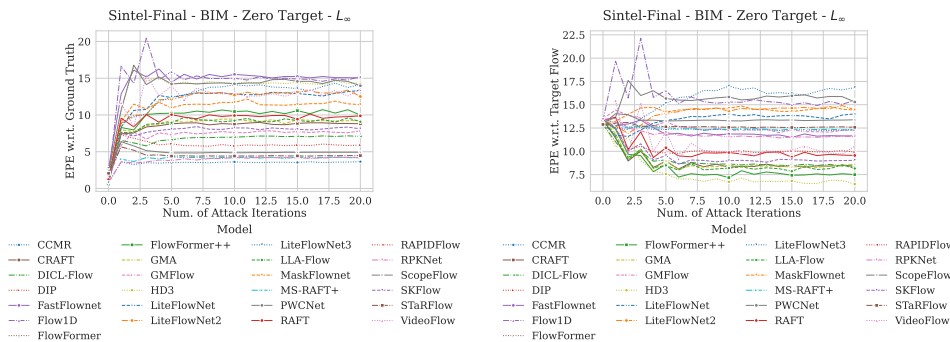

Figure 36: Evaluations for targeted BIM attack with target $\overrightarrow{0}$ under $\ell_\infty$-norm bound using the MPI Sintel (final) dataset. The attack was optimized w.r.t. the ground truth predictions.

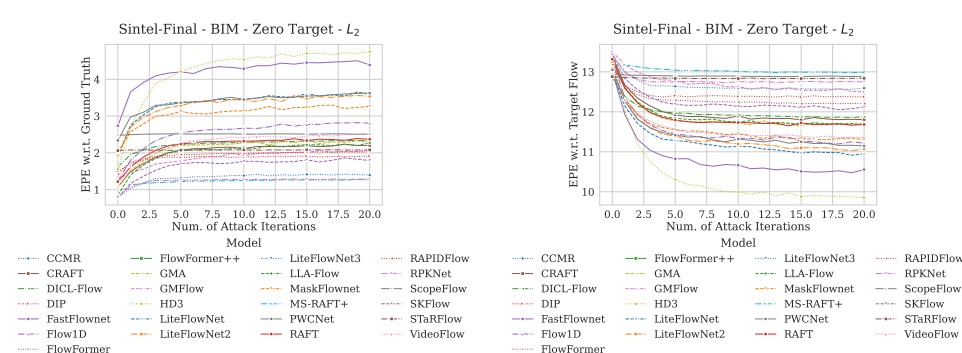

Figure 37: Evaluations for targeted BIM attack with target $\vec{0}$ under $\ell_2$-norm bound using the MPI Sintel (final) dataset. The attack was optimized w.r.t. the ground truth predictions.

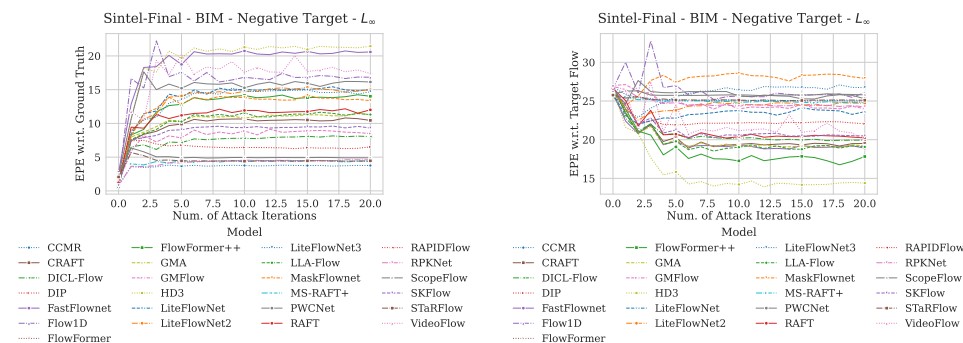

Figure 38: Evaluations for targeted BIM attack with target $-\vec{f}$ under $\ell_\infty$-norm bound using the MPI Sintel (final) dataset. The attack was optimized w.r.t. the ground truth predictions.

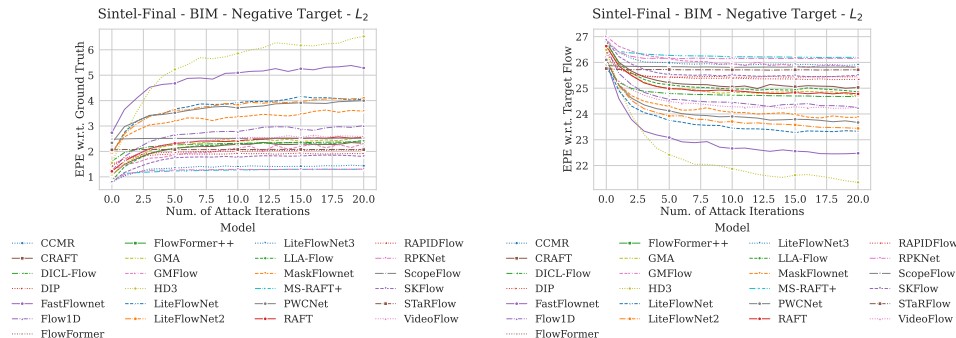

Figure 39: Evaluations for targeted BIM attack with target $-\vec{f}$ under $\ell_2$-norm bound using the MPI Sintel (final) dataset. The attack was optimized w.r.t. the ground truth predictions.

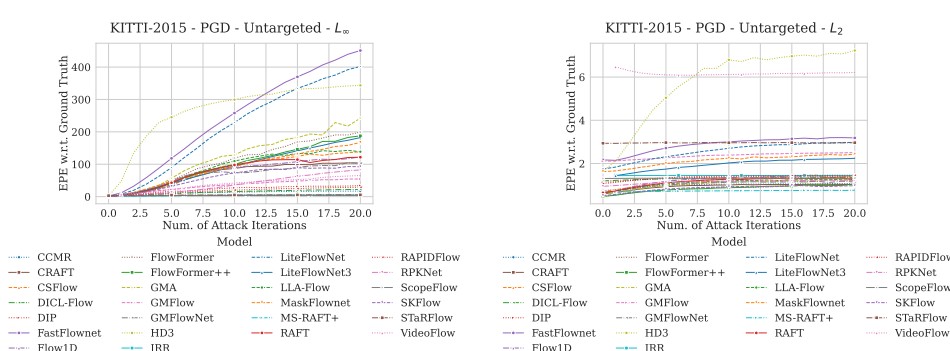

Figure 40: Evaluations for non-targeted PGD attack under $\ell_\infty$-norm bound using the KITTI2015 dataset. The attack was optimized w.r.t. the ground truth predictions.

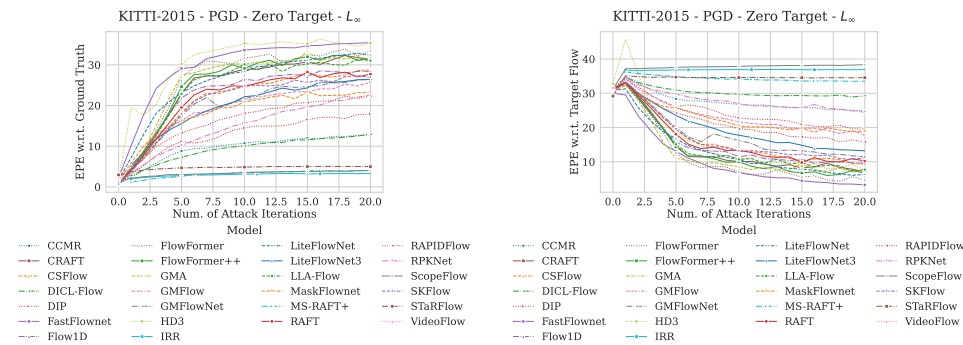

Figure 41: Evaluations for targeted PGD attack with target $\overrightarrow{0}$ under $\ell_\infty$-norm bound using the KITTI2015 dataset. The attack was optimized w.r.t. the ground truth predictions.

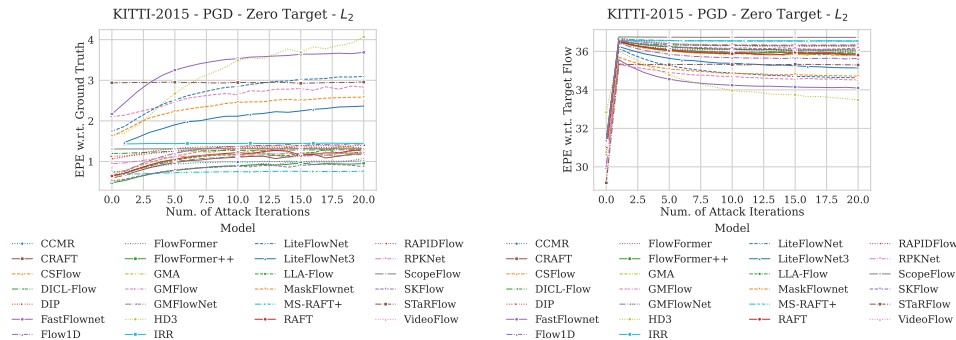

Figure 42: Evaluations for targeted PGD attack with target $\overrightarrow{0}$ under $\ell_2$-norm bound using the KITTI2015 dataset. The attack was optimized w.r.t. the ground truth predictions.

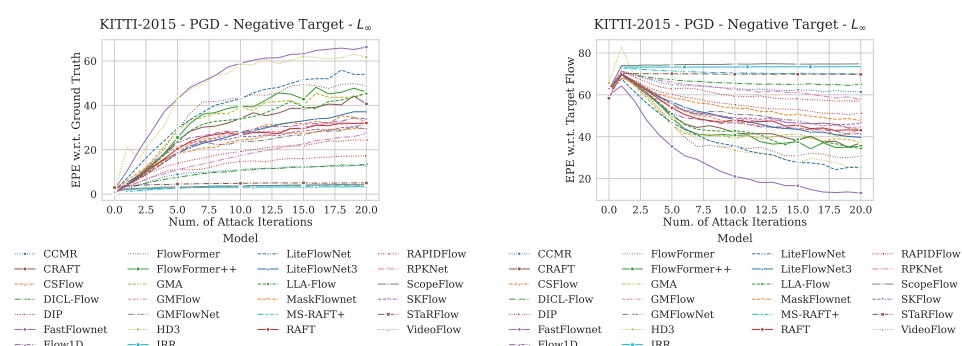

Figure 43: Evaluations for targeted PGD attack with target $-\overrightarrow{f}$ under $\ell_\infty$-norm bound using the KITTI2015 dataset. The attack was optimized w.r.t. the ground truth predictions.

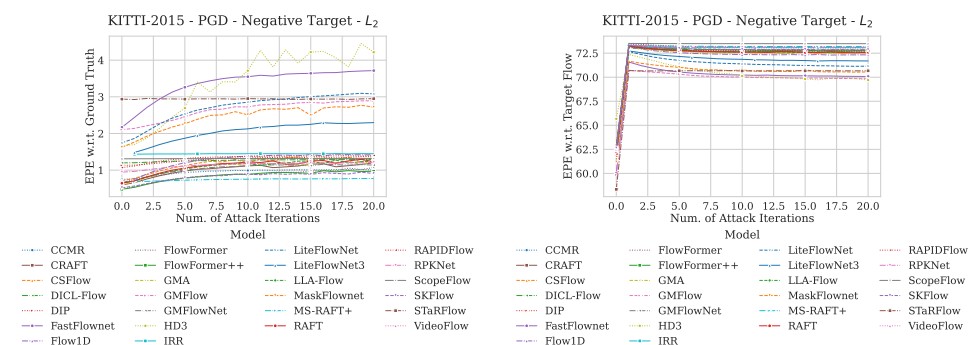

Figure 44: Evaluations for targeted PGD attack with target $-\overrightarrow{f}$ under $\ell_2$-norm bound using the KITTI2015 dataset. The attack was optimized w.r.t. the ground truth predictions.

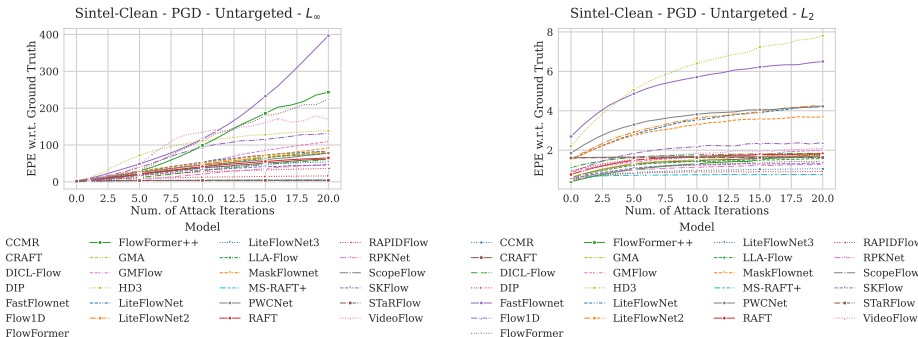

Figure 45: Evaluations for non-targeted PGD attack under $\ell_\infty$-norm bound using the MPI Sintel (clean) dataset. The attack was optimized w.r.t. the ground truth predictions.

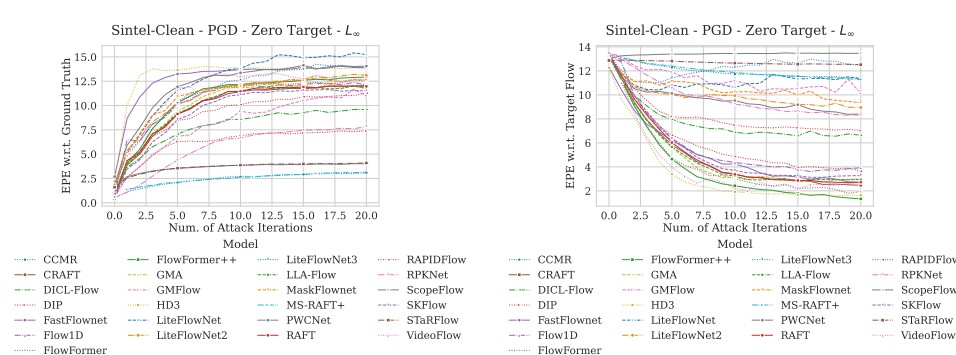

Figure 46: Evaluations for targeted PGD attack with target $\overrightarrow{0}$ under $\ell_\infty$-norm bound using the MPI Sintel (clean) dataset. The attack was optimized w.r.t. the ground truth predictions.

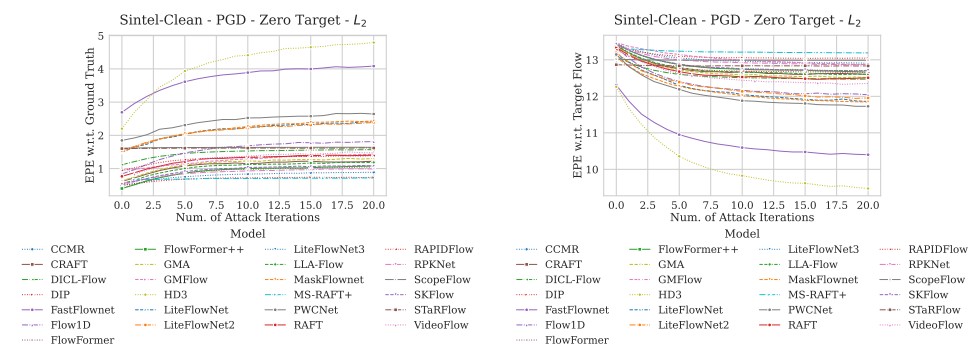

Figure 47: Evaluations for targeted PGD attack with target $\overrightarrow{0}$ under $\ell_2$-norm bound using the MPI Sintel (clean) dataset. The attack was optimized w.r.t. the ground truth predictions.

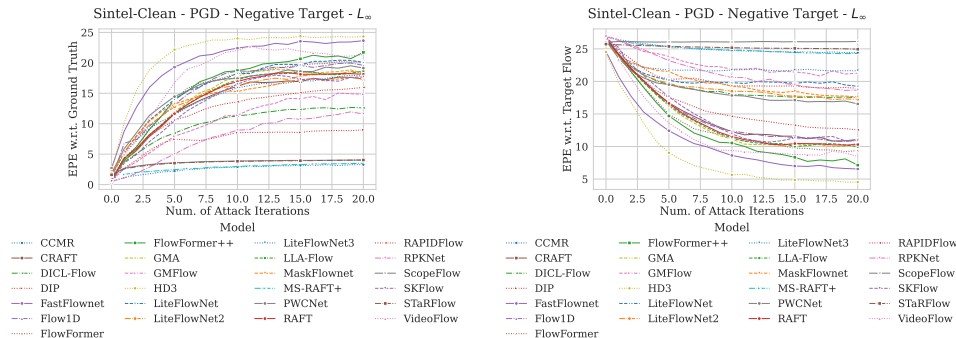

Figure 48: Evaluations for targeted PGD attack with target $-\overrightarrow{f}$ under $\ell_\infty$-norm bound using the MPI Sintel (clean) dataset. The attack was optimized w.r.t. the ground truth predictions.

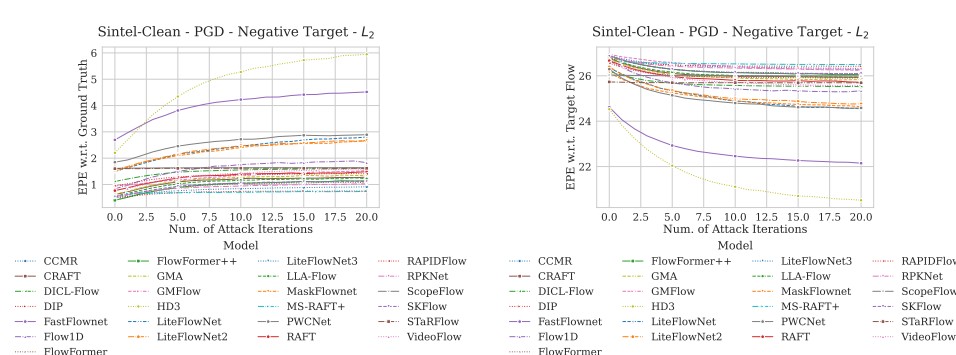

Figure 49: Evaluations for targeted PGD attack with target $-\overrightarrow{f}$ under $\ell_2$-norm bound using the MPI Sintel (clean) dataset. The attack was optimized w.r.t. the ground truth predictions.

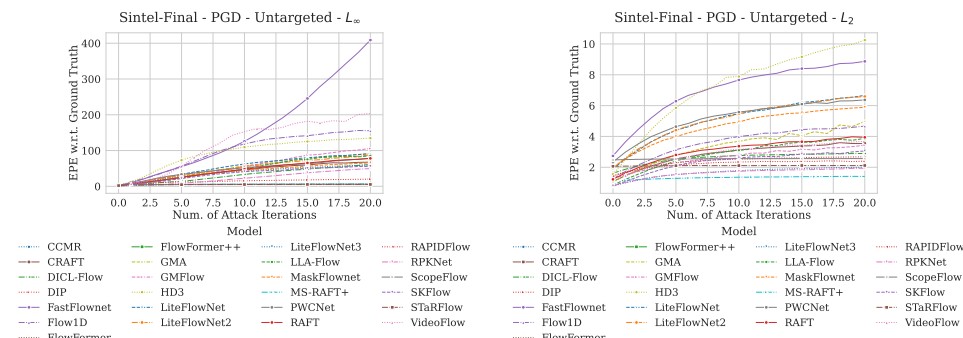

Figure 50: Evaluations for non-targeted PGD attack under $\ell_\infty$-norm bound using the MPI Sintel (final) dataset. The attack was optimized w.r.t. the ground truth predictions.

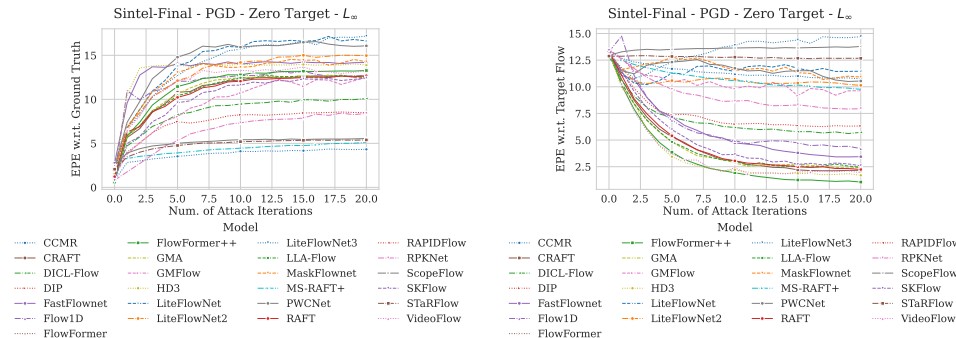

Figure 51: Evaluations for targeted PGD attack with target $\overrightarrow{0}$ under $\ell_\infty$-norm bound using the MPI Sintel (final) dataset. The attack was optimized w.r.t. the ground truth predictions.

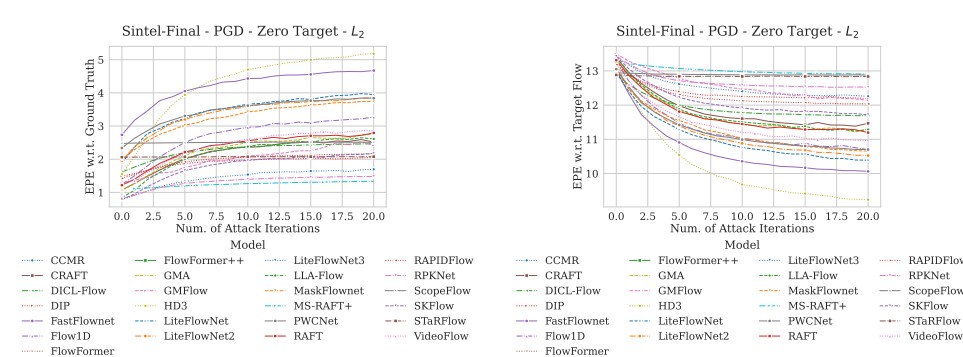

Figure 52: Evaluations for targeted PGD attack with target $\overrightarrow{0}$ under $\ell_2$-norm bound using the MPI Sintel (final) dataset. The attack was optimized w.r.t. the ground truth predictions.

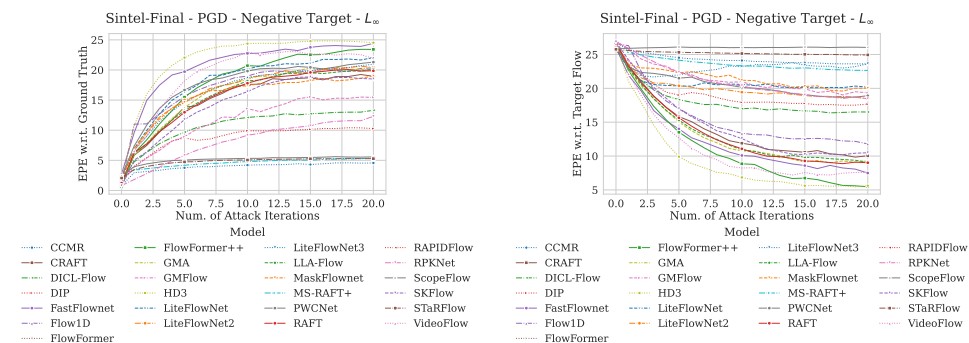

Figure 53: Evaluations for targeted PGD attack with target $-\overrightarrow{f}$ under $\ell_\infty$-norm bound using the MPI Sintel (final) dataset. The attack was optimized w.r.t. the ground truth predictions.

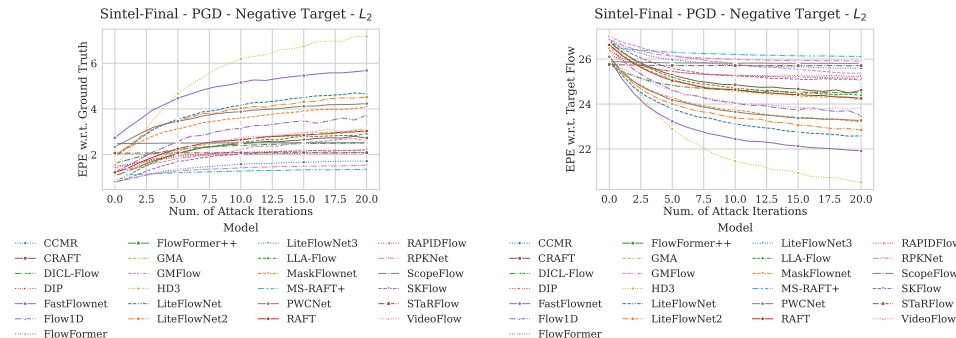

Figure 54: Evaluations for targeted PGD attack with target $-\overrightarrow{f}$ under $\ell_2$-norm bound using the MPI Sintel (final) dataset. The attack was optimized w.r.t. the ground truth predictions.

### H.1.4 CosPGD Attack

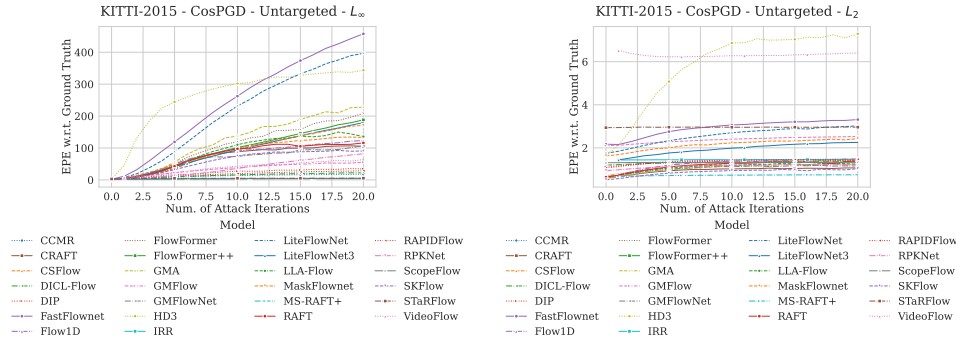

Figure 55: Evaluations for non-targeted CosPGD attack under $\ell_\infty$-norm bound using the KITTI2015 dataset. The attack was optimized w.r.t. the ground truth predictions.

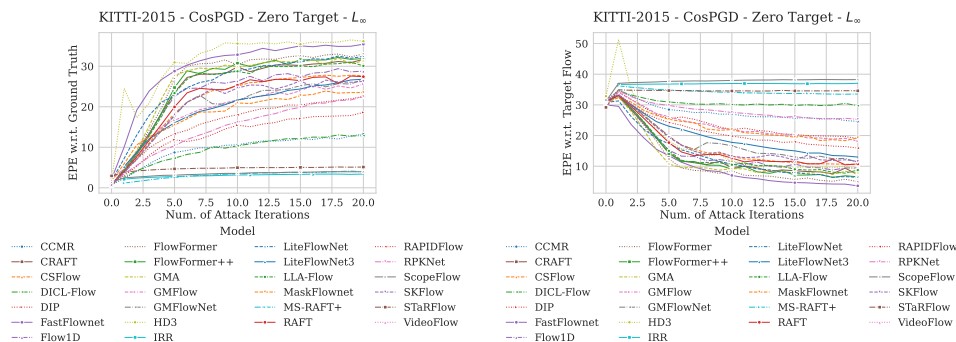

Figure 56: Evaluations for targeted CosPGD attack with target $\overrightarrow{0}$ under $\ell_\infty$-norm bound using the KITTI2015 dataset. The attack was optimized w.r.t. the ground truth predictions.

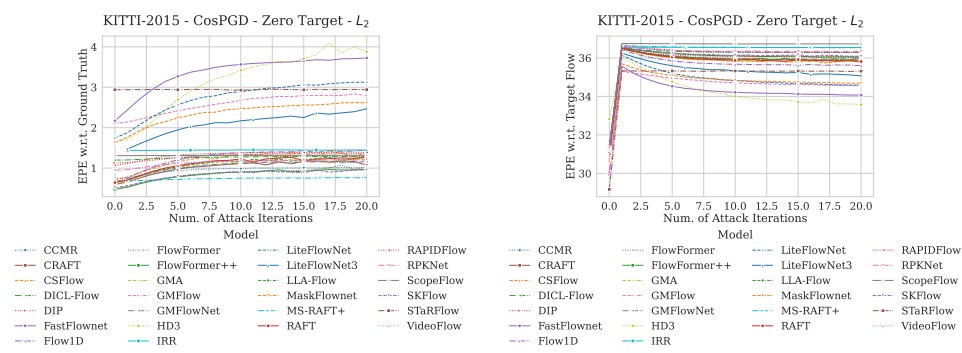

Figure 57: Evaluations for targeted CosPGD attack with target $\overrightarrow{0}$ under $\ell_2$-norm bound using the KITTI2015 dataset. The attack was optimized w.r.t. the ground truth predictions.

Here we report the evaluations using CosPGD attack, both as targeted (both targets: $\overrightarrow{0}$ and $\overrightarrow{-f}$) and non-targeted attacks optimized under the $\ell_\infty$-norm bound and the $\ell_2$-norm bound over multiple attack iterations. For $\ell_\infty$-norm bound, perturbation budget $\epsilon = \frac{8}{255}$, and step size $\alpha$=0.01, while for $\ell_2$-norm bound, perturbation budget $\epsilon = \frac{64}{255}$ and step size $\alpha$=0.1. Attack evaluations include Fig. 55, Fig. 56, Fig. 57, Fig. 58, Fig. 59, Fig. 60, Fig. 61, Fig. 62, Fig. 63, Fig. 64, Fig. 65, Fig. 66, Fig. 67, Fig. 68, and Fig. 69.

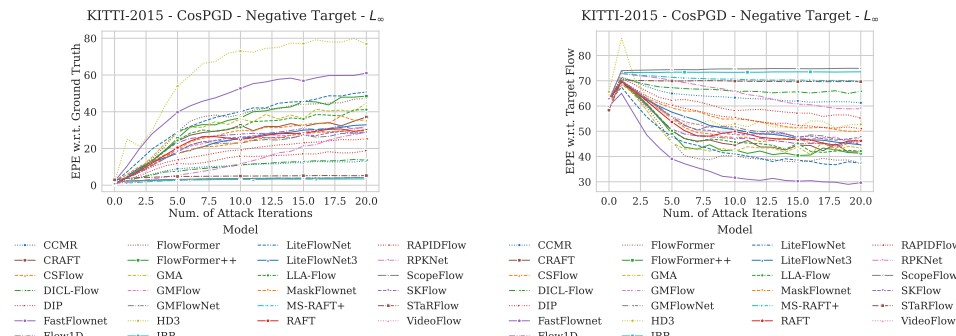

Figure 58: Evaluations for targeted CosPGD attack with target $-\overrightarrow{f}$ under $\ell_\infty$-norm bound using the KITTI2015 dataset. The attack was optimized w.r.t. the ground truth predictions.

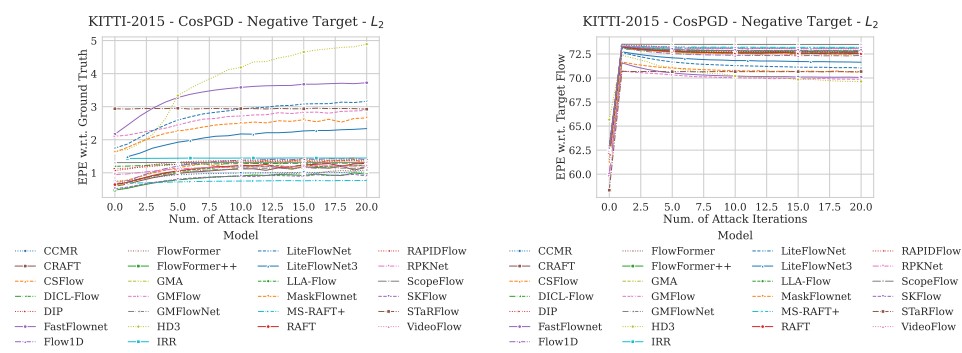

Figure 59: Evaluations for targeted CosPGD attack with target $-\overrightarrow{f}$ under $\ell_2$-norm bound using the KITTI2015 dataset. The attack was optimized w.r.t. the ground truth predictions.

### H.1.5 PCFA ATTACK

Here we report the evaluations using PCFA attack, as targeted (both targets: $\overrightarrow{0}$ and $-\overrightarrow{f}$) optimized under the $\ell_2$-norm bound over multiple attack iterations. Here the perturbation budget $\epsilon = 0.05$ and step size $\alpha = 1e - 7$. Attack evaluations include Fig. 70 and Fig. 71.

### H.1.6 ADVERSARIAL WEATHER ATTACK

Here we report the evaluations using different Adversarial Weather, both as targeted (both targets: $\overrightarrow{0}$ and $-\overrightarrow{f}$) and non-targeted attacks. Attack evaluations include Fig. 72, Fig. 73, Fig. 74 and Fig. 75.

### H.2 COMMON CORRUPTIONS OVERVIEW

Following we provide an overview of the performance over all corruptions. This is reported in Fig. 76.

### H.3 2D COMMON CORRUPTIONS

Here we report evaluations using different 2D common corruptions over all considered datasets. OOD Robustness evaluations with 2D Common Corruptions include Fig. 77, Fig. 78 and Fig. 79.

### H.4 3D COMMON CORRUPTIONS

Here we report evaluations using different considered 3D common corruptions over all considered datasets. OOD Robustness evaluations with 3D Common Corruptions include Fig. 80, Fig. 81 and Fig. 82.

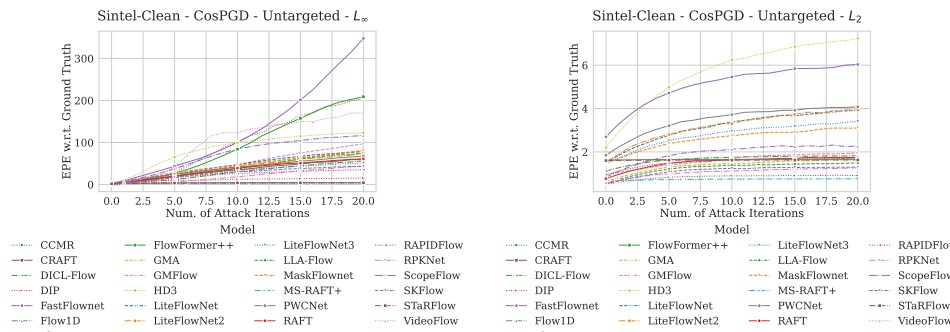

Figure 60: Evaluations for non-targeted CosPGD attack under $\ell_\infty$-norm bound using the MPI Sintel (clean) dataset. The attack was optimized w.r.t. the ground truth predictions.

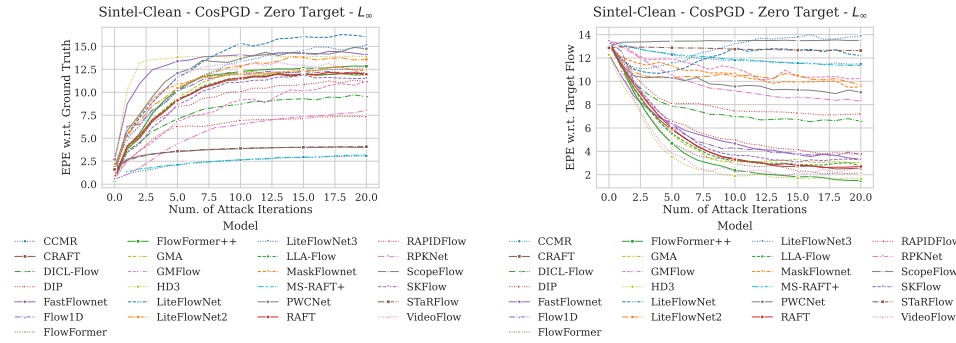

Figure 61: Evaluations for targeted CosPGD attack with target $\overrightarrow{0}$ under $\ell_\infty$-norm bound using the MPI Sintel (clean) dataset. The attack was optimized w.r.t. the ground truth predictions.

## I    INITIAL PROTOTYPE OF THE FUTURE WEBSITE

NEW

In Figure 83 we share a screenshot from our prototype website currently under work, that would help better understand the metrics. In this screenshot, the methods are ranked based on their EPE w.r.t. the ground truth flow under non-targeted CosPGD attack at 20 attack iterations under the $\ell_\infty$-norm bound (lower means the method is more robust) evaluated using the KITTI2015 dataset. We are currently designing it to make the numbers and column headings better visible to the users, and the users can dynamically rank these based on any of the columns. We will host the website after acceptance.

## J    LIMITATIONS

Benchmarking optical flow estimation methods is a compute and labor-intensive endeavor. Thus, best utilizing available resources we use FLOWBENCH to benchmark a limited number of settings. The benchmarking tool itself offers significantly more combinations that can be benchmarked. Nonetheless, the benchmark provided is comprehensive and instills interest to further utilize FLOW-BENCH.

## K    REPRODUCIBILITY OF EVALUATIONS

NEW

There always exists stochasticity when evaluating adversarial attacks, due to the randomness these attacks exploit, and also common corruptions due to variations in seeds and calculation approximations made by various python libraries. Therefore, for transparency and reproducibility, we evaluate different runs on the same seed and different runs of different seeds. We report these evaluations

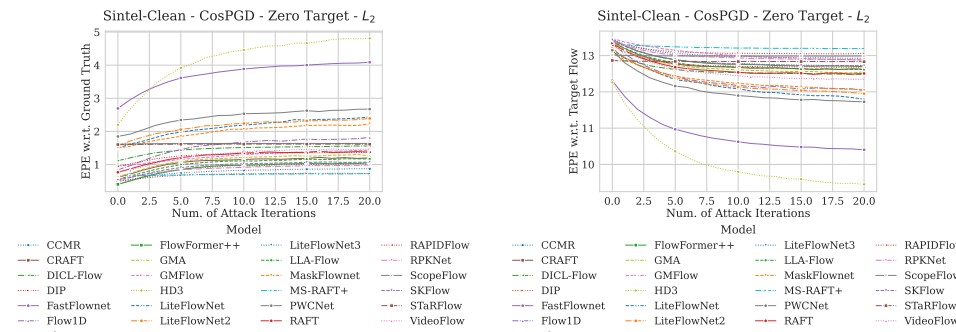

Figure 62: Evaluations for targeted CosPGD attack with target $\overrightarrow{0}$ under $\ell_2$-norm bound using the MPI Sintel (clean) dataset. The attack was optimized w.r.t. the ground truth predictions.

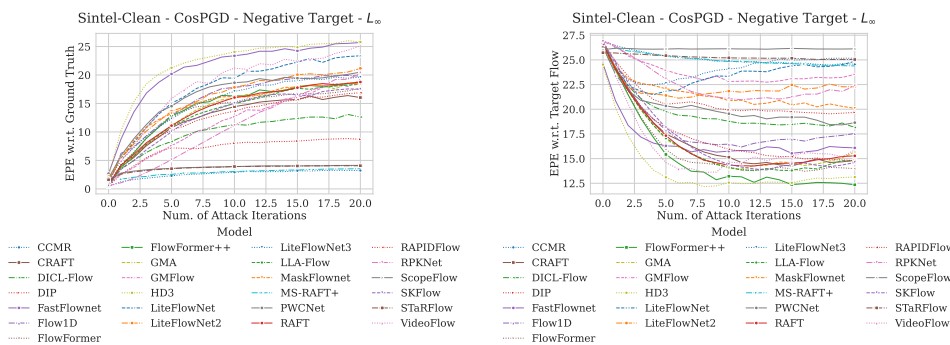

Figure 63: Evaluations for targeted CosPGD attack with target $\overrightarrow{-f}$ under $\ell_\infty$-norm bound using the MPI Sintel (clean) dataset. The attack was optimized w.r.t. the ground truth predictions.

here, using Tab. 2 for adversarial attacks and Tab. 3 for common corruptions, and observe that the variance is extremely low and the analysis performed in this work still stands.

To ensure the reproducibility of our adversarial attack evaluations we repeat experiments in two ways: first, three different runs with the same seed, and second, one run each for three different seeds. We observe very minute variations in results in both cases which can be attributed to calculation approximations made by different libraries such as pytorch (Paszke et al., 2019). Due to the compute-hungry nature of these evaluations, we limit them to using one method: RAFT on the KITTI2015 dataset, and the attack used is CosPGD. We evaluate multiple settings: different $\ell_p$-norm bounds, different attack optimization methods (optimizing w.r.t. ground truth flow and optimizing w.r.t. initial flow prediction.), and for targeted attacks, two different targets. The attack settings are consistent with the paper.

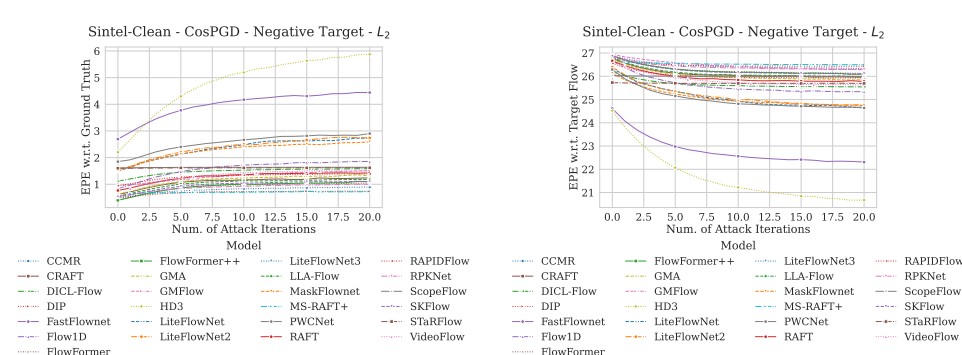

Figure 64: Evaluations for targeted CosPGD attack with target $-\overrightarrow{f}$ under $\ell_2$-norm bound using the MPI Sintel (clean) dataset. The attack was optimized w.r.t. the ground truth predictions.

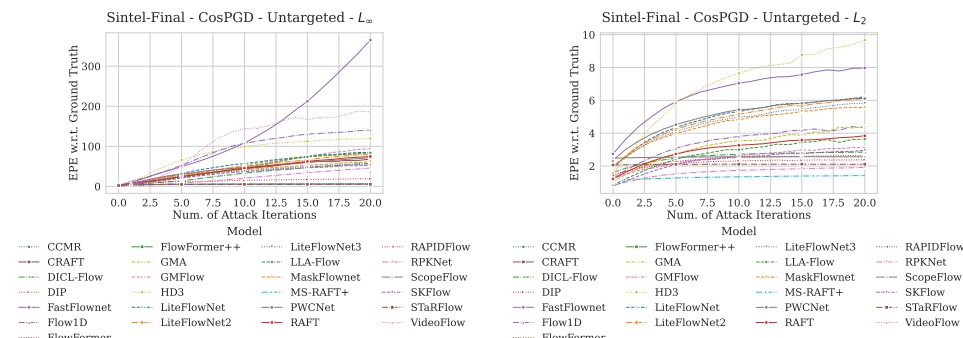

Figure 65: Evaluations for non-targeted CosPGD attack under $\ell_\infty$-norm bound using the MPI Sintel (final) dataset. The attack was optimized w.r.t. the ground truth predictions.

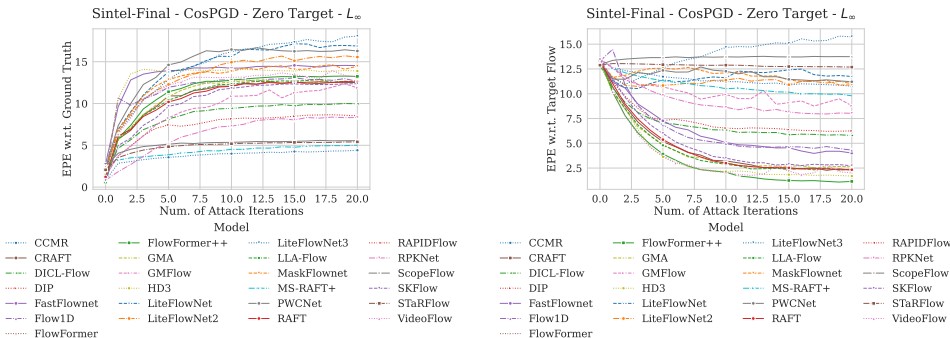

Figure 66: Evaluations for targeted CosPGD attack with target $\overrightarrow{0}$ under $\ell_\infty$-norm bound using the MPI Sintel (final) dataset. The attack was optimized w.r.t. the ground truth predictions.

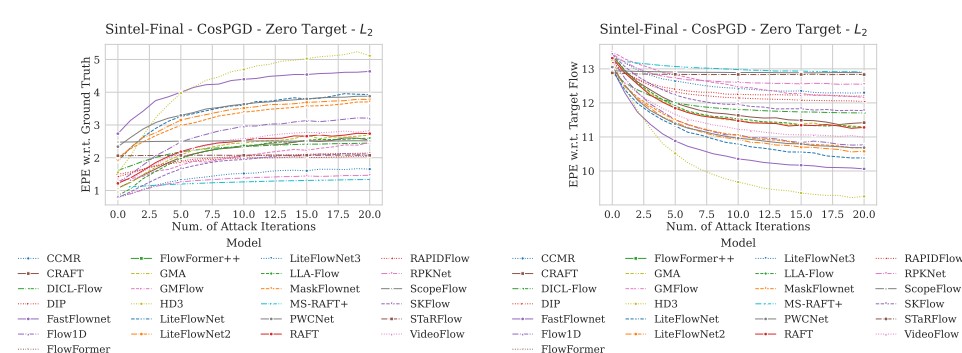

Figure 67: Evaluations for targeted CosPGD attack with target $\overrightarrow{0}$ under $\ell_2$-norm bound using the MPI Sintel (final) dataset. The attack was optimized w.r.t. the ground truth predictions.

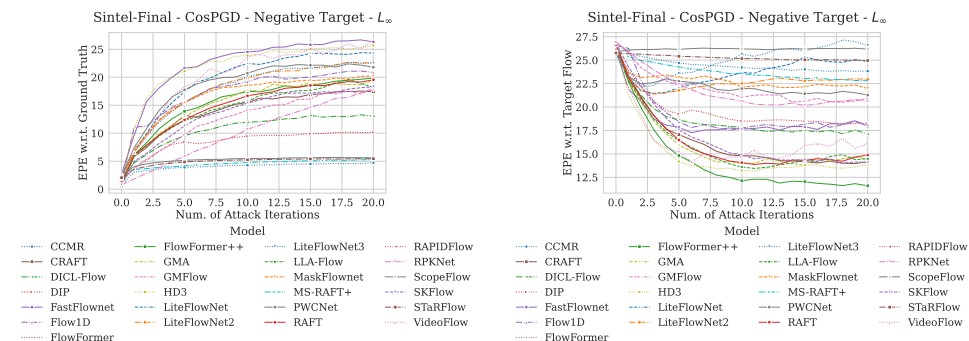

Figure 68: Evaluations for targeted CosPGD attack with target $\overrightarrow{-f}$ under $\ell_\infty$-norm bound using the MPI Sintel (final) dataset. The attack was optimized w.r.t. the ground truth predictions.

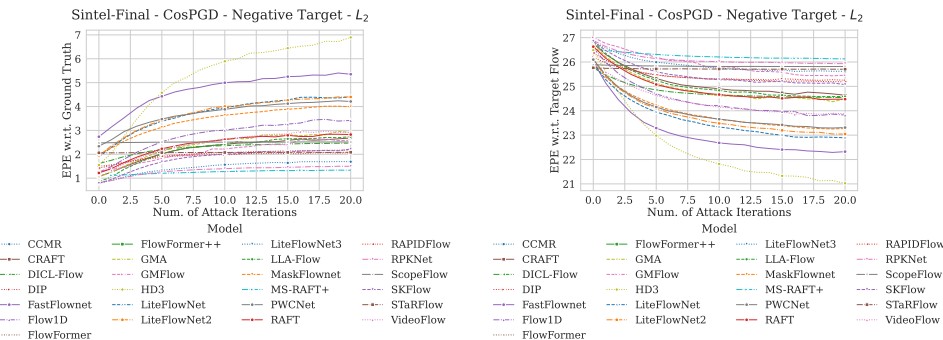

Figure 69: Evaluations for targeted CosPGD attack with target $\overrightarrow{-f}$ under $\ell_2$-norm bound using the MPI Sintel (final) dataset. The attack was optimized w.r.t. the ground truth predictions.

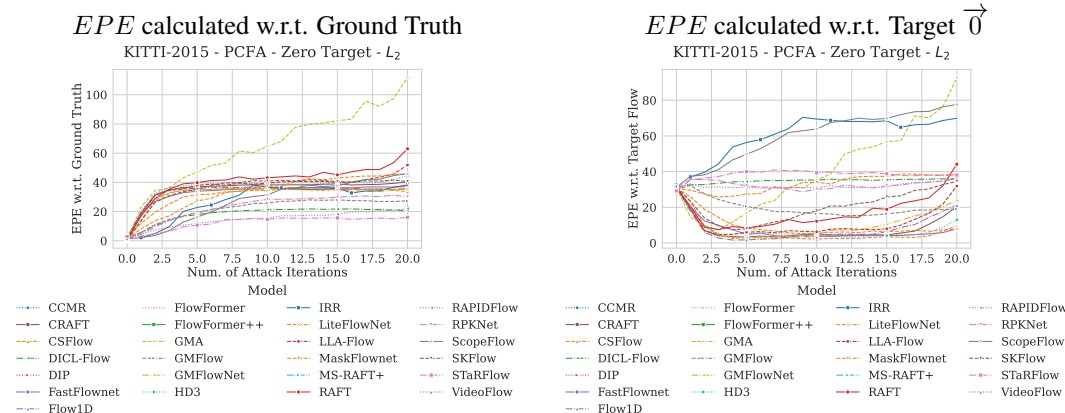

Figure 70: Evaluating all optical flow estimation methods against PCFA attack with target $\vec{0}$ over multiple attack iterations.

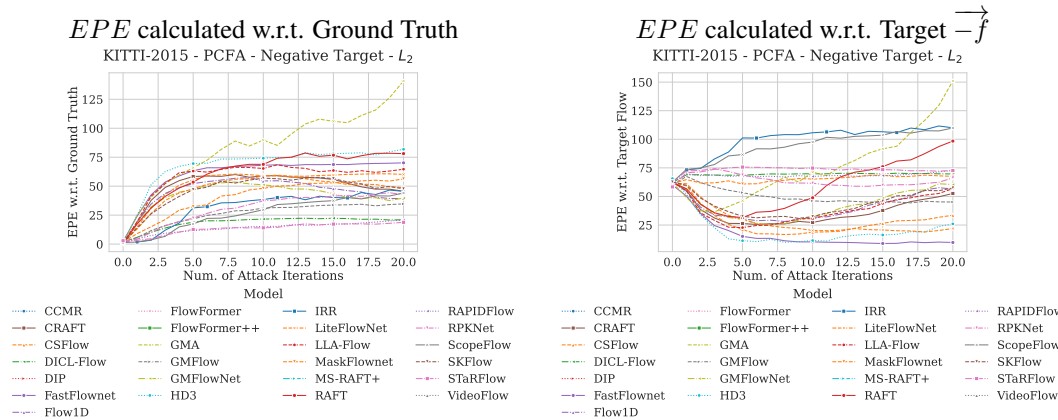

Figure 71: Evaluating all optical flow estimation methods against PCFA attack with target $-\vec{f}$ over multiple attack iterations.

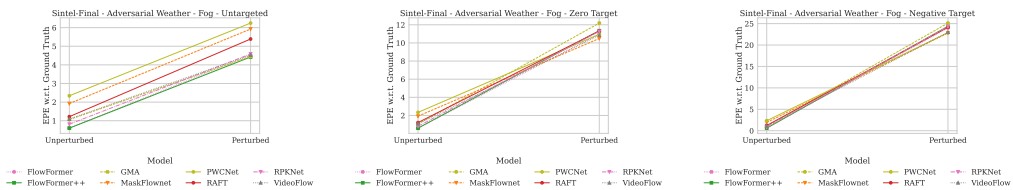

Figure 72: Evaluations for Adversarial Weather attack with Fog optimized as an non-targeted attack (left), and targeted attack with targets $\vec{0}$ (center) and $-\vec{f}$ (right).

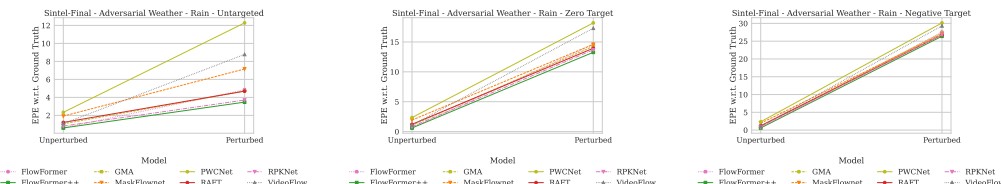

Figure 73: Evaluations for Adversarial Weather attack with Rain optimized as an non-targeted attack (left), and targeted attack with targets $\vec{0}$ (center) and $-\vec{f}$ (right).

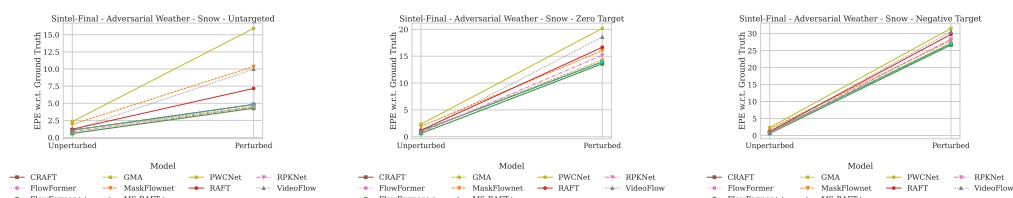

Figure 74: Evaluations for Adversarial Weather attack with Snow optimized as an non-targeted attack (left), and targeted attack with targets $\overrightarrow{0}$ (center) and $\overrightarrow{-f}$ (right).

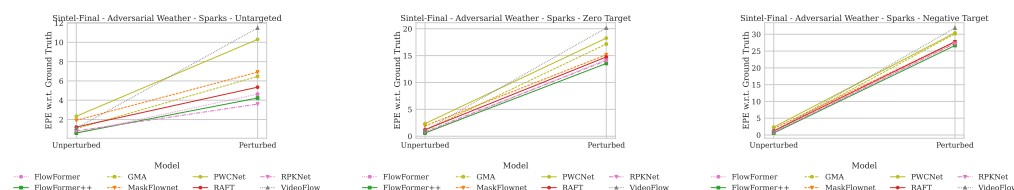

Figure 75: Evaluations for Adversarial Weather attack with Sparks optimized as an non-targeted attack (left), and targeted attack with targets $\overrightarrow{0}$ (center) and $\overrightarrow{-f}$ (right).

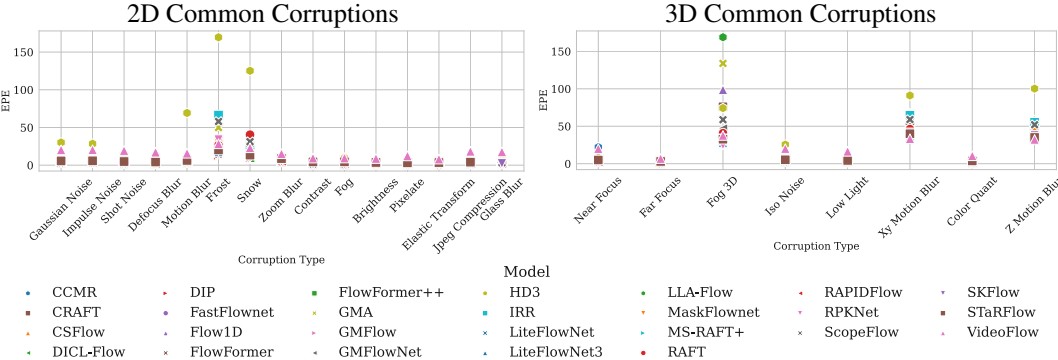

Figure 76: Performance of various optical flow estimation methods after corruptions on the KITTI2015 dataset.

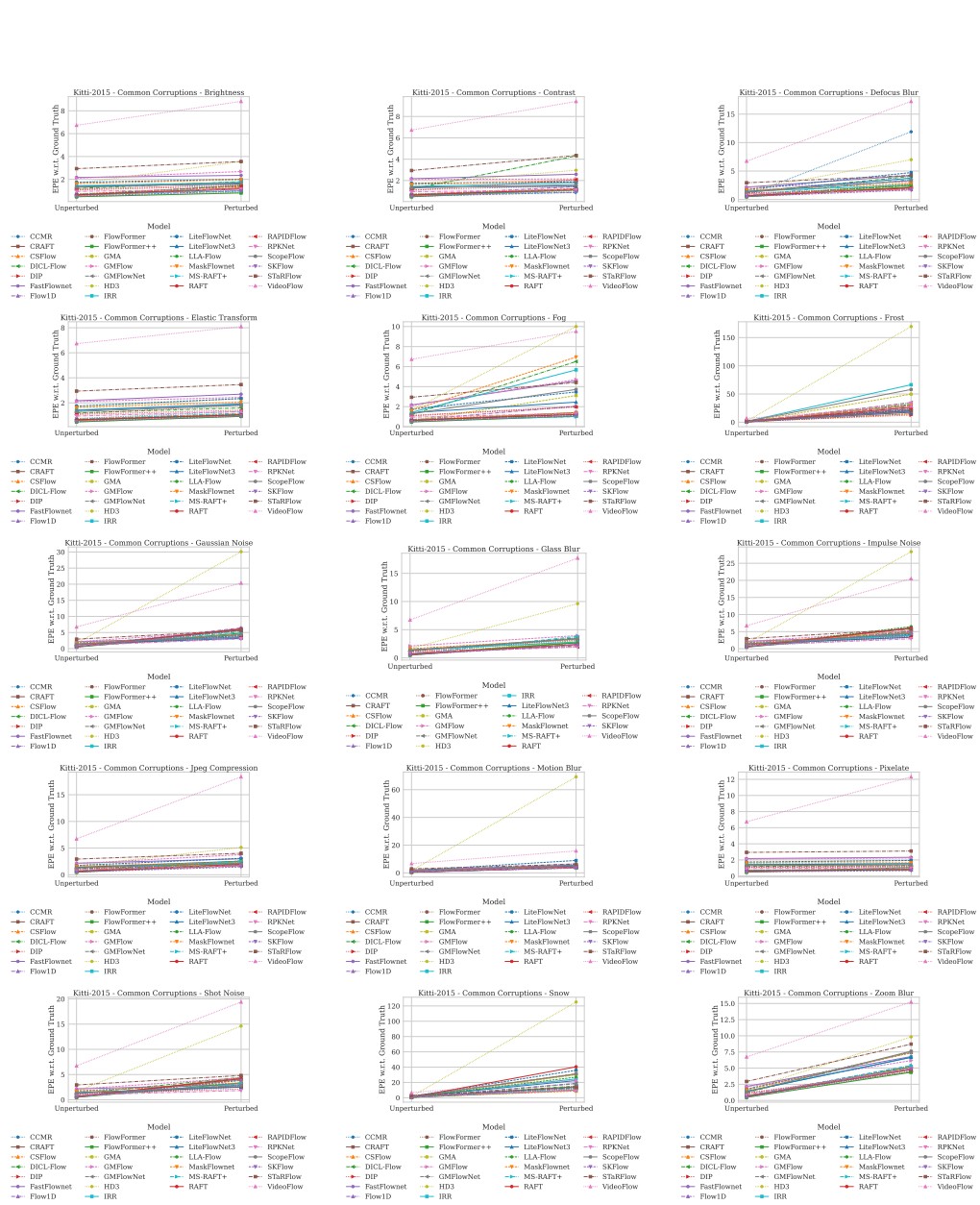

Figure 77: Evaluating optical flow estimation methods against all 2D Common Corruptions on the KITTI2015 dataset.

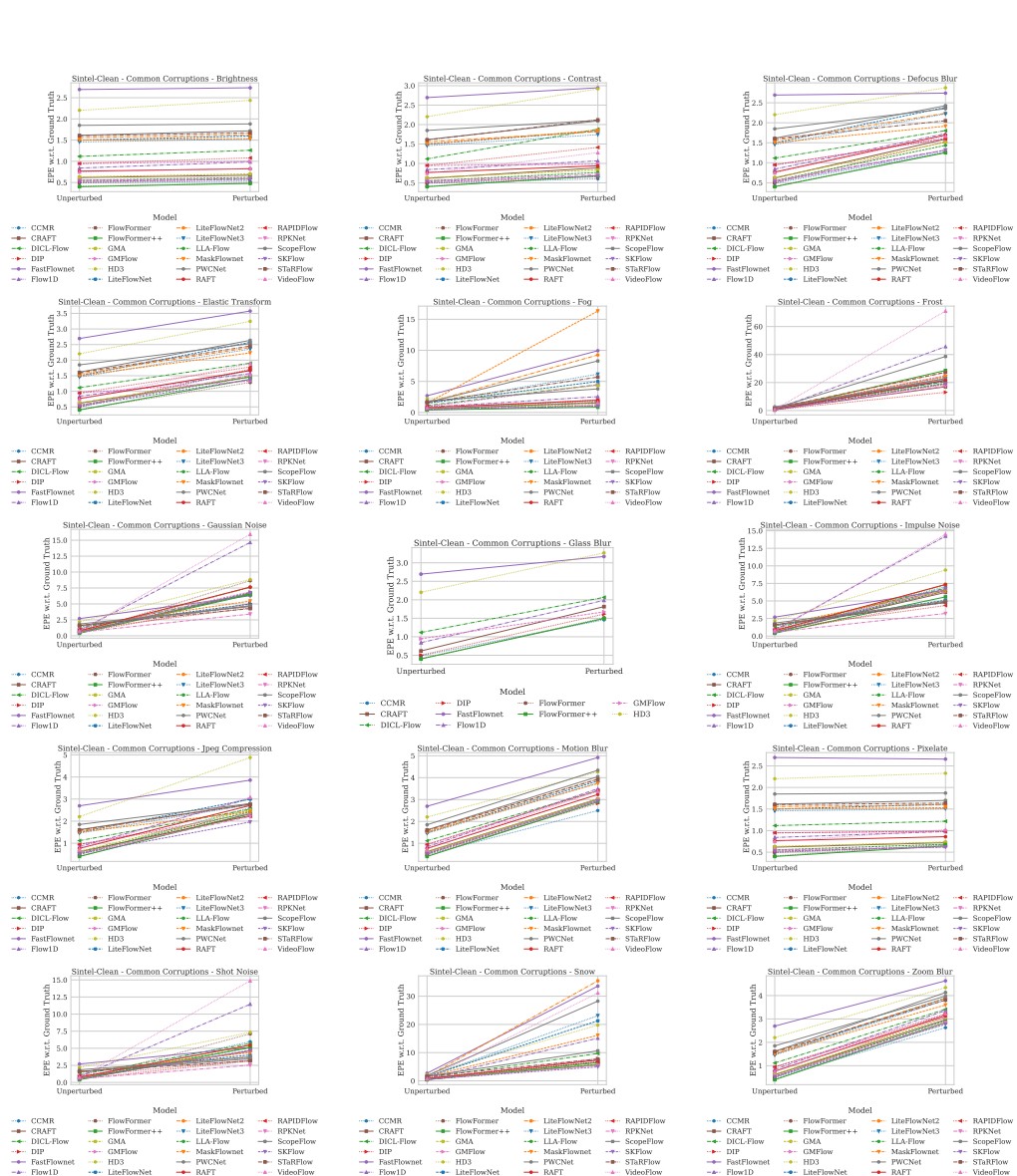

Figure 78: Evaluating optical flow estimation methods against all 2D Common Corruptions on the MPI Sintel (clean) dataset.

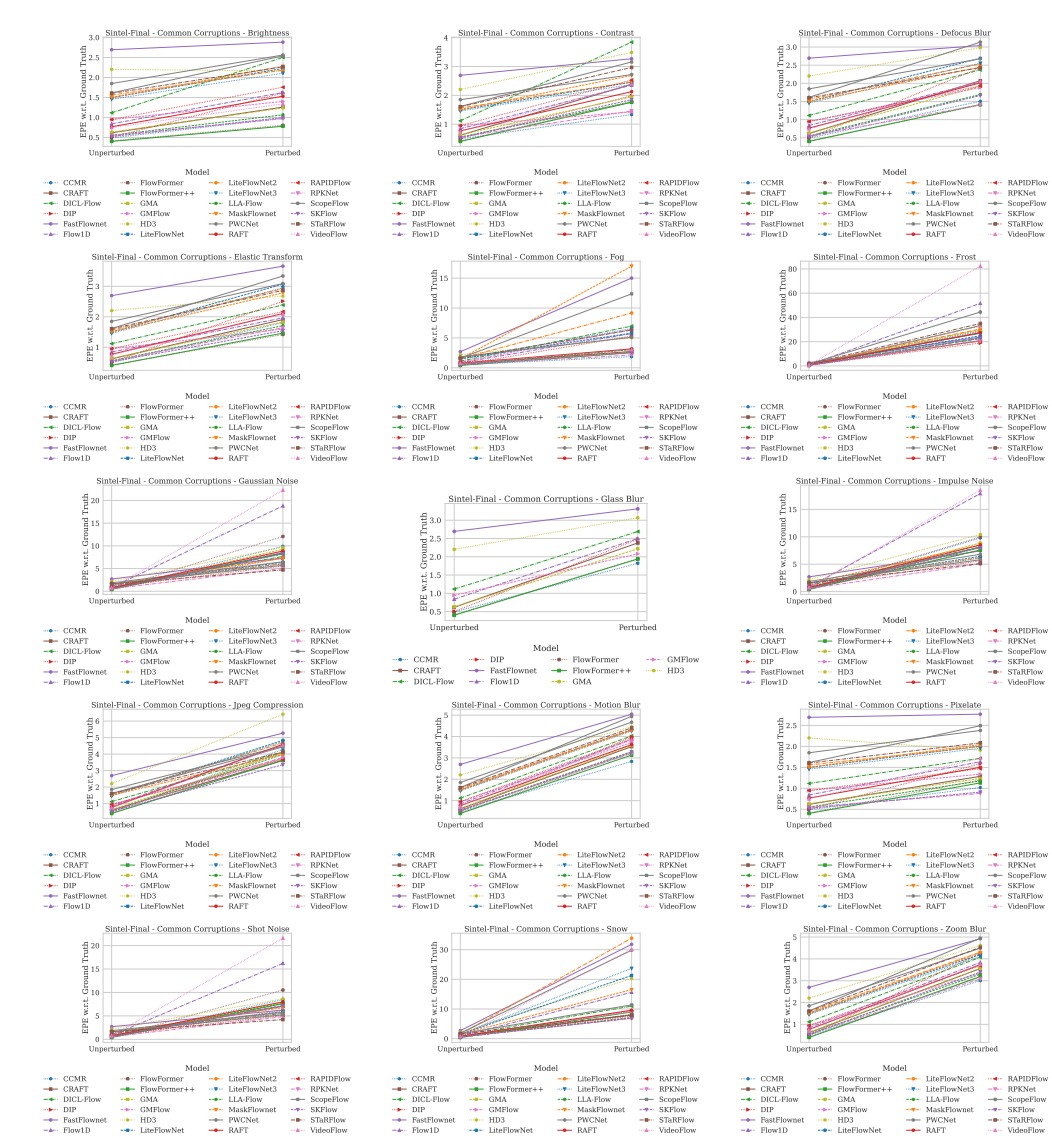

Figure 79: Evaluating optical flow estimation methods against all 2D Common Corruptions on the MPI Sintel (final) dataset.

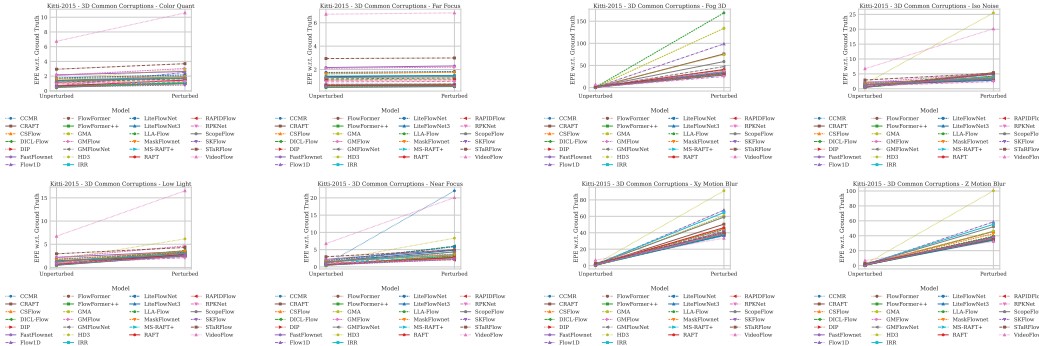

Figure 80: Evaluating optical flow estimation methods against the considered 3D Common Corruptions on the KITTI2015 dataset.

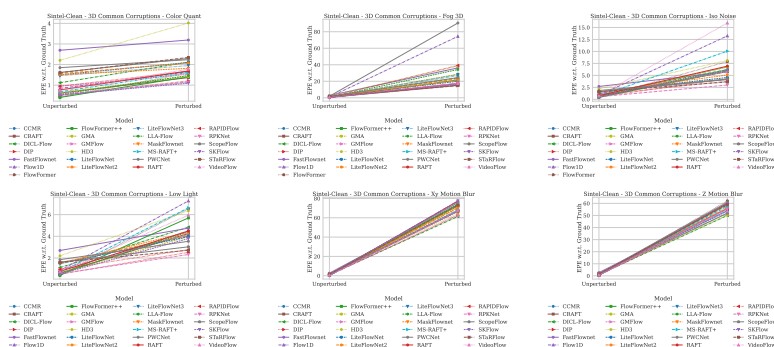

Figure 81: Evaluating optical flow estimation methods against the considered 3D Common Corruptions on the MPI Sintel (clean) dataset.

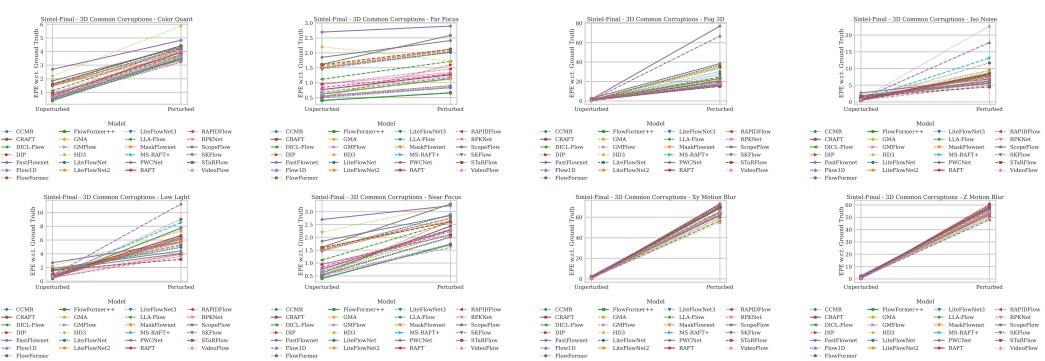

Figure 82: Evaluating optical flow estimation methods against the considered 3D Common Corruptions on the MPI Sintel (final) dataset.

## FlowBench

Optical flow estimation is a crucial computer vision task often applied to safety critical real-world scenarios like autonomous driving and medical imaging. While optical flow estimation accuracy has greatly benefited from the emergence of Deep learning, learning-based methods are also known for their lack of generalization and reliability. However, reliability is paramount when optical flow methods are employed in the real world, where safety is essential. Furthermore, a deeper understanding of the robustness and reliability of learning-based optical flow estimation methods is still lacking, hindering the research community from building methods safe for real-world deployment. Thus we propose FLOWBENCH, a robustness benchmark and evaluation tool for learning-based optical flow methods. FLOWBENCH facilitates streamlined research into the reliability of optical flow methods by benchmarking their robustness to adversarial attacks and out-of-distribution samples. With FLOWBENCH, we benchmark 91 methods across 3 different datasets under 7 diverse adversarial attacks and 23 established common corruptions, making it the most comprehensive robustness analysis of optical flow methods to date. Across this wide range of methods, we consistently find that methods with state-of-the-art performance on established standard benchmarks lack reliability and generalization ability. Moreover, we find interesting correlations between performance, reliability, and generalization ability of optical flow estimation methods, under various lenses such as design choices used, number of parameters, etc. After acceptance, FLOWBENCH will be open-source and publicly available, including the weights of all tested models.

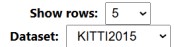

Show rows: 5

Dataset: KITTI2015

### Leaderboard: Optical Flow Estimation

| Rank | Architecture | CosPGD-EPE No... | PCFA-EPE target... | PGD-EPE Non-ta... | Checkpoint | Time_Proposed |
|------|--------------|------------------|---------------------|--------------------|------------|---------------|
| 1 | IRR_PWC | 2.4330115861 | 69.8680467474 | 3.8397940102 | KITTI | Mar 2020 |
| 2 | ScopeFlow | 2.6299911237 | 77.6024067444 | 4.5666427672 | KITTI | Nov 2023 |
| 3 | MS-RAFT+ | 2.695769617 | 40.9803659793 | 5.0118829945 | KITTI | Jul 2020 |
| 4 | StarFlow | 4.1756131017 | 38.06560606 | 5.4629693043 | KITTI | Mar 2024 |
| 5 | DICL | 10.5310789597 | 35.9340447974 | 21.0924557686 | KITTI | Jul 2021 |

Figure 83: A share a screenshot from our prototype website currently under works, that would help better understand the metrics. In this screenshot, the methods are ranked based on their EPE w.r.t. the ground truth flow under non-targeted CosPGD attack at 20 attack iterations under the $\ell_\infty$-norm bound (lower means the method is more robust) evaluated using the KITTI2015 dataset. We are currently designing it to make the numbers and column headings better visible to the users, and the users can dynamically rank these based on any of the columns.

Table 2: To ensure reproducibility of our adversarial attack evaluations we repeat experiments in two ways: first, three different runs with the same seed, and second, one run each for three different seeds. We observe very minute variations in results in both cases which can be attributed to calculation approximations made by different libraries such as pytorch (Paszke et al., 2019). Due to the compute-hungry nature of these evaluations, we limit them to using one method: RAFT on the KITTI2015 dataset, and the attack used is CosPGD. We evaluate multiple settings: different $\ell_p$-norm bounds, different attack optimization methods (optimizing w.r.t. ground truth flow and optimizing w.r.t. initial flow prediction.), and for targeted attacks, two different targets. The attack settings are consistent with the paper. Target 'None' means the attack was Non-targeted.

| $\ell_p$-norm bound | Target | Attack Optimized w.r.t. | EPE mean $\pm$ std | px3 error mean $\pm$ std |
|---|---|---|---|---|
| Three different runs on the same seed | | | | |
| $\ell_\infty$-norm | None | Ground Truth Flow | $119.504 \pm 2.95E+0$ | $0.078 \pm 6.76E\text{-}3$ |
| $\ell_\infty$-norm | $-\overrightarrow{f}$ | Ground Truth Flow | $45.357 \pm 4.26E\text{-}1$ | $0.200 \pm 7.84E\text{-}4$ |
| $\ell_\infty$-norm | $\overrightarrow{0}$ | Ground Truth Flow | $10.674 \pm 2.74E\text{-}1$ | $0.647 \pm 1.06E\text{-}2$ |
| $\ell_2$-norm | None | Ground Truth Flow | $0.644 \pm 2.72E\text{-}6$ | $0.968 \pm 3.38E\text{-}6$ |
| $\ell_2$-norm | $-\overrightarrow{f}$ | Ground Truth Flow | $73.454 \pm 3.12E\text{-}5$ | $0.129 \pm 2.86E\text{-}7$ |
| $\ell_2$-norm | $\overrightarrow{0}$ | Ground Truth Flow | $36.724 \pm 2.11E\text{-}5$ | $0.170 \pm 4.41E\text{-}7$ |
| $\ell_2$-norm | None | Initial Flow Pred | $0.643 \pm 7.74E\text{-}6$ | $0.968 \pm 1.49E\text{-}6$ |
| One run each using three different seeds | | | | |
| $\ell_\infty$-norm | None | Ground Truth Flow | $119.692 \pm 1.75E+0$ | $0.077 \pm 4.27E\text{-}3$ |
| $\ell_\infty$-norm | $-\overrightarrow{f}$ | Ground Truth Flow | $45.149 \pm 9.16E\text{-}1$ | $0.202 \pm 1.65E\text{-}3$ |
| $\ell_\infty$-norm | $\overrightarrow{0}$ | Ground Truth Flow | $11.016 \pm 4.91E\text{-}1$ | $0.625 \pm 9.90E\text{-}3$ |
| $\ell_2$-norm | None | Ground Truth Flow | $0.644 \pm 7.46E\text{-}6$ | $0.968 \pm 6.05E\text{-}6$ |
| $\ell_2$-norm | $-\overrightarrow{f}$ | Ground Truth Flow | $73.454 \pm 1.15E\text{-}4$ | $0.129 \pm 1.85E\text{-}7$ |
| $\ell_2$-norm | $\overrightarrow{0}$ | Ground Truth Flow | $36.724 \pm 1.10E\text{-}4$ | $0.170 \pm 7.92E\text{-}7$ |
| $\ell_2$-norm | None | Initial Flow Pred | $0.643 \pm 1.44E\text{-}4$ | $0.968 \pm 1.55E\text{-}5$ |

Table 3: To ensure reproducibility of our Common Corruptions evaluations we repeat experiments in two ways: first, three different runs with the same seed, and second, one run each for three different seeds. We observe extremely minute variations in results which can be attributed to differences in seeds and calculation approximations made by the Python libraries. Due to the compute-hungry nature of these evaluations, we limit them to using one method: RAFT on the KITTI2015 dataset, and all the fifteen 2D Common Corruptions.

| 2D Common Corruption Name | EPE mean ± std | px3 error mean ± std |
|---|---|---|
| Three different runs on the same seed | | |
| brightness | 1.235 ± 0.000 | 0.935 ± 0.000 |
| contrast | 1.187 ± 0.000 | 0.938 ± 0.000 |
| defocus blur | 2.026 ± 0.000 | 0.899 ± 0.000 |
| elastic transform | 1.043 ± 0.000 | 0.954 ± 0.000 |
| fog | 1.221 ± 0.000 | 0.936 ± 0.000 |
| frost | 29.640 ± 0.000 | 0.383 ± 0.000 |
| gaussian noise | 5.931 ± 0.000 | 0.732 ± 0.000 |
| glass blur | 2.409 ± 0.000 | 0.861 ± 0.000 |
| impulse noise | 6.098 ± 0.220 | 0.736 ± 0.002 |
| jpeg compression | 1.942 ± 0.000 | 0.892 ± 0.000 |
| motion blur | 5.515 ± 0.000 | 0.549 ± 0.000 |
| pixelate | 0.785 ± 0.000 | 0.960 ± 0.000 |
| shot noise | 4.435 ± 0.000 | 0.780 ± 0.000 |
| snow | 41.974 ± 0.000 | 0.354 ± 0.000 |
| zoom blur | 4.808 ± 0.000 | 0.746 ± 0.000 |
| One run each using three different seeds | | |
| brightness | 1.235 ± 0.000 | 0.935 ± 0.000 |
| contrast | 1.187 ± 0.000 | 0.938 ± 0.000 |
| defocus blur | 2.026 ± 0.000 | 0.899 ± 0.000 |
| elastic transform | 1.041 ± 0.009 | 0.953 ± 0.001 |
| fog | 1.282 ± 0.076 | 0.937 ± 0.001 |
| frost | 28.783 ± 0.827 | 0.391 ± 0.019 |
| gaussian noise | 5.877 ± 0.026 | 0.735 ± 0.001 |
| glass blur | 2.532 ± 0.105 | 0.859 ± 0.002 |
| impulse noise | 5.856 ± 0.067 | 0.737 ± 0.002 |
| jpeg compression | 1.942 ± 0.000 | 0.892 ± 0.000 |
| motion blur | 4.135 ± 0.141 | 0.565 ± 0.010 |
| pixelate | 0.785 ± 0.000 | 0.960 ± 0.000 |
| shot noise | 4.336 ± 0.143 | 0.781 ± 0.004 |
| snow | 42.984 ± 4.786 | 0.362 ± 0.007 |
| zoom blur | 4.808 ± 0.000 | 0.746 ± 0.000 |

