# OpenReview forum: "FlowBench: A Robustness Benchmark for Optical Flow Estimation"
_ICLR.cc/2025/Conference — Submitted to ICLR 2025_

### Official Review · Reviewer_GPBg · 2024-10-26

**Soundness:** 3
**Presentation:** 2
**Contribution:** 3
**Rating:** 5
**Confidence:** 2

**Summary:**

This paper proposes a benchmark, FlowBench, to evaluate the robustness of current optical flow methods. FlowBench is built on KITTI-2015 and MPI-Sintel. It mainly focuses on the EPE metric and provides an easy-to-use API for attacking optical flow models. Besides, it includes the performance analysis.

**Strengths:**

Main Strength:
- Engineering: FlowBench provides an easy-to-use API for further study and analyzes the reliability/generalization of current methods.


Minor Strength:
- Novelty (in Optical Flow): To my best knowledge, the robustness of optical flow has not been discussed for a long time. It is good to have a benchmark of current methods.

**Weaknesses:**

Main Weakness:
- Lack of technical contribution: the whole benchmark is built on existing datasets. Besides, the attack techniques are from existing papers, not domain-specific. It is good to see some analysis of the performance. However, there is no solution provided to mitigate the issue. That makes me feel the paper is mainly engineering.


Minor Weakness:
- EPE is not the only metric in optical flow. The paper should analyze more on 1px-outlier rate, WAUC, etc.
- The writing is not clear. I find it hard to understand the main points in the paper.

As an outsider of adversarial attacks, I may misunderstand the technical contributions in this paper. I am happy to raise the score if the authors can provide explanations in detail, and correct me if I am wrong.

**Questions:**

See weakness.

---

> ### Author Response · Authors · 2024-11-19
>
> Dear Reviewer GPBg,
>
> Thank you very much for your review, we greatly appreciate your input and suggestions.
>
> Indeed, FLOWBENCH does not provide a new method for optical flow estimation but is rather intended to facilitate research towards reliable and robust optical flow estimation methods and to provide insights and analysis to existing methods. While being “an engineering contribution”  - and an analysis of key findings -, it is timely and important.  This has also been acknowledged by the other reviewers.
>
>
> This work, as confirmed by Reviewer MVeY, is “An extensive study on 89 released models/checkpoints over four widely used datasets shows in-depth findings and insights” (now 91 models). As further highlighted by Reviewer MVeT, “This paper is among the first attempts to evaluate optical flow methods on reliability and generalization ability, which are both important but often overlooked in the research community” and “The results on all latest models are insightful and worth sharing.” Reviewer hv2r further extends the strengths of this work by mentioning that “This paper provides an interesting analysis that methods with the best accuracy are not necessarily the most robust methods. Given comprehensive analyses, readers can understand the pros and cons of each method. I believe the benchmark will certainly benefit the community and help design a better and more robust method for future work” and Reviewer F85d emphasizes the importance of our work by mentioning, “This work …  by focusing on robustness—vital for real-world applications like autonomous driving and medical imaging—it directly addresses the gap between research advancements and deployable technology, which is quite important”.
>
>
> **Metrics Beyond EPE:**
>
> Yes, we agree that apart from EPE, other metrics might also be interesting to different users. Thus, apart from EPE, FLOWBENCH also enables calculating a lot of other interesting metrics, such as outlier, 1-px error, 3-px error, 5-px error, and cosine distance between two vectors. These vectors are the same as that in the case of EPE calculations.
> We limited the analysis in this work to use EPE, since it is the most commonly used metric for evaluation, moreover, most works on optical flow estimation (such as CosPGD, PCFA, RAFT, MS-RAFT+, and many other works) show a very high correlation between performance evaluations using different metrics.
> Following your suggestion, we now better highlight this in Appendix C.
>
>
> **Writing:**
>
> Thank you very much for bringing this to our attention. we agree that the original submission might not have been easy to follow for researchers not familiar with adversarial and OOD robustness.
> Thus we now revise the text to better explain these (now available, revisions in blue color).
> For example, we now explain in Sec. 4.2 that an adversarial attack is a perturbation made on the input images to fool a method into changing its predictions while the input image looks semantically similar to a human observer.
> Most works that focus on the reliability of optical flow estimation methods perform adversarial attacks, however, these works either focus on targeted attacks or on non-targeted attacks, not both at the same time.
> The objective of targeted attacks is to optimally perturb the input image such that the method predictions are changed towards a specifically desired target, for example, a target can be a $\overrightarrow{0}$ flow i.e. attacking so that the flow prediction at all pixels should become zero.
> On the other hand, non-targeted adversarial attacks do not intend to shift the method's predictions to a specific target, they simply intend to fool the method into making any incorrect predictions.

---

> > ### Author Response · Authors · 2024-11-19
> >
> > Further, we now better highlight in the revised paper, the key takeaways from this work, which are as follows:
> >
> >
> >
> > First, we demonstrate the need for FLOWBENCH, the lack of a framework to study the reliability and generalization abilities of optical flow methods alongside their i.i.d. Performance has led to the unfortunate reality of methods proposed over time only improving the i.i.d. Performance and not their reliability and generalization abilities. As commonly used. reliability is the robustness against white-box adversarial attacks, as they serve as a proxy for the worst-case scenario, while, generalization ability is the robustness to image corruptions, specifically 2D and 3D Common Corruptions as they serve as a proxy for real-world domain shifts and signal corruptions. The lack of reliability and generalization ability is severely alarming, as the error of optical flow estimation methods in i.i.d. Evaluations is in the low single digits, whereas the error of methods in evaluations against adversarial attacks and common corruptions is in the high hundreds!
> >
> >
> > Second, in Sec. 5.1, we find that there is a high correlation in the performance of optical flow estimation methods against targeted attacks using different targets, thus saving compute for future works as they need to evaluate only against one target. Please note, that evaluations using adversarial attacks are extremely expensive both time-wise and computation-wise. Thus, the impact of this finding cannot be underscored.
> >
> >
> > Third, in Sec. 5.2, we observe the methods known to be SotA on i.i.d. samples are not reliable, and do not generalize well to image corruptions, demonstrating the gap in current research when considering real-world applications. Additionally, we observe here that there is no apparent correlation between generalization abilities and the reliability of optical flow estimation methods.
> >
> >
> > Fourth, in Sec. 5.3, we show that methods using attention-based pointing matching are marginally more reliable than methods using other matching techniques, while methods using CNN and Cost Volume-based matching have marginally better generalization abilities. Please note that observations third and fourth in conjecture help us conclude that based on current works, different approaches might be required to attain reliability under attacks and generalization ability to image corruptions.
> >
> >
> > Fifth, in Sec. 5.4, we show that, unlike image classification, increasing the number of learnable parameters does not help increase the robustness of optical flow estimation methods.
> >
> >
> > Sixth, in Sec 5.5, we show that white-box adversarial attacks on optical flow estimation methods can be independent of the availability of ground truth information, and can harness the information in the initial flow predictions to optimize attacks, thus overcoming a huge limitation in the field an allowing for reliability studies of optical flow methods even in the absence of ground truth flows.
> >
> >
> >
> > Such an in-depth understanding of reliability and generalization abilities to optical flow estimation methods can only be obtained using our proposed FLOWBENCH. We are certain that FLOWBENCH will be immensely helpful to gather more such interesting findings and its comprehensive and consolidated nature would make things easier for the research community.
> >
> >
> >
> > Please do let us know if you have any further questions or concerns, or if you feel that we missed addressing any of your questions or concerns, we would be happy to address them.
> >
> > Best Regards
> >
> > Authors of Paper #1055
> >
> >
> > References:
> >
> > [1] Zhang, Zhiyong, Huaizu Jiang, and Hanumant Singh. "NeuFlow: Real-time, High-accuracy Optical Flow Estimation on Robots Using Edge Devices." arXiv preprint arXiv:2403.10425 (2024).
> >
> > [2] Wang, Bo, Yifan Zhang, Jian Li, Yang Yu, Zhenping Sun, Li Liu, and Dewen Hu. "Splatflow: Learning multi-frame optical flow via splatting." International Journal of Computer Vision (2024): 1-23.

---

> ### Comment · Reviewer_GPBg · 2024-11-20
> **Reviewer Response**
>
> Thank you for your detailed explanation! The rebuttal provides more information to address my concerns. However, I agree with Reviewer F85d's concern on architecture analysis. For example, Flowformer combines transformers with correlation. But Table 1 classified it as *Attention*. This makes me suspect the reliability of the analysis in Sec 5.3/5.4. Also, different methods may use different training data, which can be a factor that severely affects the robustness. (e.g. Flowformer++ uses more data than Flowformer. RAFT is trained from scratch, but Flowformer loads pre-trained weights). I think that is why I feel the analyses are messy. A better classification of methods should be provided.

---

> ### Author Response · Authors · 2024-11-20
>
> Dear Reviewer GPBg,
>
> Thank you very much for your reply.
> Thank you for bringing to our attention that we misclassified Flowformer and Flowformer++ in our initial assessment.
> We have now correctly classified it as “Attention + Cost Volume”.
> Nonetheless, the findings from our observations still hold.
> The biggest takeaway from our work is that irrespective of the classification method i.e. irrespective of the point-matching method used and irrespective of the number of learnable parameters, all the DL-based optical flow estimation methods are not reliable under adversarial attacks and do not generalize to image corruptions.
> Yes, there are minor differences in robustness performance based on such factors, however, compared to the errors on i.i.d. Data, the errors under attack and errors under image corruptions are significantly higher, making these differences hardly significant and thus demonstrating an alarming threat to these methods if deployed in the real world.
> To the best of our understanding, this point comes across well from the analysis in Section 5.3 and Section 5.4.
>
> Yes, we agree that different methods use different training strategies, as also pointed out by Reviewer hv2r.
> We agree that the training setup can play a significant role in the robustness of a method. However, in the case of optical flow methods, they are hardly ever adapted to the target setting since the target application usually does not have ground truth leading one to rely on the available pre-trained models. Currently, the training strategies and architectural design choices of different methods are significantly intertwined. For example, MS-RAFT+ is trained significantly differently from FlowFormer++, FlowFormer++ can be evaluated on a dataset only if it is finetuned for that dataset (which we did), however, MS-RAFT+ is proposed with a single training strategy to work across datasets. To ensure fair comparison, we contacted the authors of MS-RAFT+ and they accepted that our evaluations would be fair according to them. In our current analysis, we attempt to make significant observations based on multiple models, keeping their training and evaluation settings intact. We agree that it would be difficult to disentangle if the performance difference is due to the architectural design choice or due to the training strategy, however, previous works also do not disentangle the two when comparing i.i.d. Performance. Given that, optical flow methods were not proposed to specifically overcome adversarial attacks or image corruptions, we hypothesize that training all optical flow with a different known setup would not help improve their reliability and generalization abilities and the current observations from this work would still hold. Thus, we believe that the robustness evaluations performed here are realistic. To improve the reliability and generalization abilities of optical flow methods, better approaches need to be proposed, that focus on these aspects, and studying these would be very interesting. Thank you for the suggestion, we now include a synopsis of this discussion in future work.
>
> Please do let us know if we successfully answered your questions and concerns, or if you have any further questions or concerns, we would be happy to address them.
>
>
>
> Best Regards
>
> Authors of Paper #1055

---

> > ### Comment · Reviewer_GPBg · 2024-11-21
> >
> > Thank the authors for their efforts during the rebuttal! It might be acceptable to regard a method as the combination of "training strategy" and "model architecture". I still strongly suggest authors rewrite the part about the analysis - you can pick some representative methods and their variants: RAFT/SEA-RAFT/MS-RAFT+/..., Flowformer/Flowformer++ ..., GMFlow/Unidepth/..., FlowNet/LiteFlowNet/... and analyze them in a group-wise way, which will make more sense to me.
> >
> > Also, as I am an outsider of adversarial attacks, I will keep a low confidence score. I will raise my review score.

---

> ### Author Response · Authors · 2024-11-23
>
> Dear Reviewer GPBg,
>
> Thank you, firstly for engaging in the discussion so far and secondly, for raising the score.
> We would additionally like to thank you for your suggestion, we have now incorporated this in our revised manuscript (now available).
>
> We believe your suggestion further improves the quality of the manuscript, helping us better understand the differences in different families (representative methods and their variants) of optical flow methods.
> In revised Appendix B we provide a detailed justification for why a particular method has been considered within the chosen family.
>
> In Section 5.3 we analyze the methods solely based on the family that is respectively analyzed, and the architectural design choices and training strategies that make different methods in a single family stand out.
>
> Please do let us know if we successfully addressed your concerns, or if you have any further questions or concerns, we would be happy to address them.
>
> Best Regards
>
> Authors of Paper #1055

---

> > ### Author Response · Authors · 2024-11-27
> >
> > Dear Reviewer GPBg,
> >
> > We highly appreciate your valuable suggestions.
> >
> > If you believe that we have addressed your remaining concerns by revisiting the analysis in Section 5.4 and Section 5.5 of the paper according to your latest suggestions, we would highly appreciate if you could re-evaluate your current recommendation.
> >
> > As acknowledged by Reviewer MVeY “The results …. are insightful and worth sharing.”, and Reviewer hv2r “I believe the benchmark will certainly benefit the community and help design a better and more robust method for future work.”, and Reviewer F85d “by focusing on robustness—vital for real-world applications like autonomous driving and medical imaging—it directly addresses the gap between research advancements and deployable technology, which is quite important”, we believe this work getting accepted would highly benefit the community and serve them by promoting DL-based optical flow methods that are safer for real-world deployment.
> >
> > If you have any other concerns or questions, please feel free to share them with us, we would certainly try to address them.
> >
> > Best Regards
> >
> > Authors of Paper #1055

---

### Official Review · Reviewer_F85d · 2024-10-27

**Soundness:** 2
**Presentation:** 2
**Contribution:** 2
**Rating:** 3
**Confidence:** 4

**Summary:**

This paper introduces FlowBench, a novel benchmarking tool designed to evaluate the robustness of optical flow estimation methods, specifically their resistance to adversarial attacks and performance under out-of-distribution (OOD) conditions. As deep learning methods continue to improve accuracy in optical flow tasks, their vulnerability to adversarial perturbations and limitations in generalizing to OOD data remain concerns. FlowBench addresses this by providing a systematic and scalable evaluation across 89 models on three datasets and under seven types of adversarial attacks and 23 corruptions, making it one of the most comprehensive benchmarks for optical flow to date.

**Strengths:**

The work provides a wide-ranging benchmark covering multiple models, datasets, adversarial attacks, and common corruptions. This expansive scope allows researchers to evaluate optical flow models comprehensively. In addition, by focusing on robustness—vital for real-world applications like autonomous driving and medical imaging—it directly addresses the gap between research advancements and deployable technology, which is quite important.

**Weaknesses:**

A limitation of FlowBench lies in its evaluation scope, which, while robust, could benefit from a broader range of real-world data to fully capture the variability that optical flow models encounter in practical applications. The three datasets used provide valuable insights but may not represent the complete spectrum of challenges seen in diverse environments, such as varying lighting and extreme motion scenarios.

Furthermore, the analysis correlating robustness with model parameter count, though insightful, stops short of examining deeper architectural factors, such as layer structure or feature extraction mechanisms, which may also significantly impact robustness. To make this analysis more actionable, the authors could explore specific architectural aspects, such as comparing various attention mechanisms (e.g., FlowFormer vs. FlowFormer++) or different convolutional architectures used in optical flow models.

Lastly, the emphasis on adversarial robustness, while essential, could be complemented by additional real-world stressors to offer a more holistic view of model performance, moving beyond adversarial scenarios to include conditions that challenge models in everyday applications. For example, evaluating performance under specific conditions such as varying lighting, motion blur, or partial occlusions—common in real-world scenarios but not fully captured by the current corruption set—could help round out the analysis.

**Questions:**

Given that state-of-the-art models on i.i.d. benchmarks often underperform under adversarial attacks and corruptions, how might FlowBench be extended to offer predictive insights into robustness? Could specific patterns in initial evaluations serve as indicators of a model's robustness, potentially reducing the need for exhaustive testing and saving computational resources? For example, the authors could explore correlations between performance on specific corruption types and overall robustness, or investigate whether particular architectural features consistently predict stronger robustness across models. Developing such predictive indicators could reduce the need for exhaustive testing and conserve computational resources.

Additionally, the claim of comprehensiveness is somewhat compromised, as several concurrent models are omitted, including some with available code for KITTI and Sintel datasets.

---

> ### Author Response · Authors · 2024-11-19
>
> Dear Reviewer F85d,
>
> Thank you very much for your review, we greatly appreciate your input and suggestions.
>
> As you rightly point out, we might have initially missed the opportunity to better communicate the evaluation metrics such as GAE (previously GAM) in the main paper, by only providing details in the appendix.
> We therefore revised our submission to now better highlight that the work does consider realistic corruptions that an optical flow method can face in the wild similar to those suggested by you (revisions are in blue in the revised paper now available).
> We would like to take this opportunity to highlight these evaluations and the contributions of said evaluations:
>
>
> Our analysis and all possible evaluations using FLOWBENCH, go beyond just theoretical worst-case scenarios of adversarial attacks.
> FLOWBENCH as a benchmarking tool, and our evaluated benchmark for analysis, both also consider synthetic image corruptions.
> We now discuss and empirically prove in Appendix A, that synthetic Common Corruptions serve as a reliable proxy for the real world.
> Unfortunately, there exists no dataset captured in the wild with corruptions and domain shifts that also contains ground truth for optical flow estimation, however, this effort has been made for semantic segmentation with the ACDC dataset.
> In Appendix A we discuss how for semantic segmentation, synthetic image corruptions (very similar to those used in FLOWBENCH) accurately represent the possible domain shifts and corruptions that can be captured in the wild.
>
>
> We specifically use 2D Common Corruptions and 3D Common Corruptions for our evaluations. The calculated Generalization Ability, GAE (previously GAM) is the maximum EPE of a method across all 15 2D Common Corruptions: `Gaussian Noise`, `Shot Noise`, `Impulse Noise`, `Defocus Blur`, `Frosted Glass Blur`, `Motion Blur`, `Zoom Blur`, `Snow`, `Frost`, `Fog`, `Brightness`, `Contrast`, `Elastic Transform`, `Pixelate`, `JPEG Compression`, and eight 3D Common Corruptions: `Color Quantization`, `Far Focus`, `Fog 3D`, `ISO Noise`, `Low Light`, `Near Focus`, `XY Motion Blur`, and `Z Motion Blur`. All the common corruptions are at severity 3. There exist more 3D Common Corruptions, however computing them is extremely resource intensive.
> Please note that attempting to capture these corruptions in the real world, and annotating the ground truth flow is significantly more expensive.
>
>
> Thus, we believe that our analysis is well-rounded and takes care of possible real-world scenarios suggested by you.
>
>
> Regarding the suggestions for the analysis in this work, thank you very much for bringing it to our attention. We now address this in the revised conclusion of the work. We discuss that methods using attention-based pointing matching are marginally more reliable than methods using other matching techniques, while methods using CNN and Cost Volume-based matching have marginally better generalization abilities.
>
> This in conjecture with a previous observation that there is no apparent correlation between generalization abilities and the reliability of optical flow estimation methods, helps us conclude that based on current works, different approaches might be required to attain reliability under attacks and generalization ability to image corruptions.
>
> Please note, here we can merely talk about “improvements” as “marginal” because as seen from the TARE, NARE, and GAE (previously TARM, NARM, and GAM) values, which are essentially EPE values calculated over multiple evaluation methods, there is an absolute lack of robustness in all the optical flow methods. These observations are merely highlighting “the least bad among the worst”. While the EPE values of these methods for i.i.d. Evaluations are in the low single digits, their EPE values for evaluations under adversarial attacks and image corruptions are in the hundreds!

---

> ### Author Response · Authors · 2024-11-19
>
> Answers to Questions:
>
>
> FLOWBENCH currently covers the following three predictive analyses that help significantly lower the computational expenditure of future works in this direction:
>
> 1. In Sec. 5.1, we find that there is a high correlation in the performance of optical flow estimation methods against targeted attacks using different targets, thus saving compute resources for future works as they need to evaluate only against one target. Please note, that evaluations using adversarial attacks are extremely expensive both time-wise and computation-wise. Thus, the impact of this finding cannot be underscored as one can be used as a prediction for the other.
>
> 2. In Sec. 5.5, we show that white-box adversarial attacks on optical flow estimation methods can be independent of the availability of ground truth information, and can harness the information in the initial flow predictions to optimize attacks, thus overcoming a huge limitation in the field. Here, the initial predicted flow acts as a proxy for the ground truth flow, reducing the huge amount of work required to capture ground truth flow.
>
> 3. In Appendix A, we show that synthetic common corruptions included in FLOWBENCH serve as a reliable proxy for possible real-world domain shifts and corruptions, thus significantly reducing the effort to gather such data in the wild, and helping users predict the performance of their methods in real-world scenarios.
>
>
>
> Lastly, please note that training optical flow methods is extremely expensive, and thus we use publicly available checkpoints. FLOWBENCH is built upon ptlflow, and any new method and checkpoint added to ptlflow in the future would be implemented using FLOWBENCH.
> Following your suggestion and given the time constraint, we extend our analysis to two recently proposed methods SplatFlow (February, 2024) [1] and NeuFlow (March, 20204) [2]. We urge the community to use FLOWBENCH upon acceptance and expand the analysis to future methods as they are proposed.
>
>
> Please note that our current analysis contains evaluations against many targeted and non-targeted adversarial attacks: FGSM ($\ell_{\infty}$ and $\ell_2$-norm bounded), BIM ($\ell_{\infty}$ and $\ell_2$-norm bounded), PGD ($\ell_{\infty}$ and $\ell_2$-norm bounded), CosPGD ($\ell_{\infty}$ and $\ell_2$-norm bounded), PCFA ($\ell_2$-norm bounded), and Adversarial Weather, and 23 Common Corruptions for 91 models. Barring limited analysis on Adversarial Weather (sparks, rain, snow, fog) due to their high computational requirements, all other evaluations exist for each of the 91 models, i.e. 91*23 (common corruptions) + 91*8 ( $\ell_{\infty}$ and $\ell_2$-norm bounded non-targeted adversarial attacks) + 91*18 ($\ell_{\infty}$ and $\ell_2$-norm bounded targeted adversarial attacks with zero target and negative flow) + 91 (i.i.d. evaluations), which is over 4500 evaluations total over the datasets: KITTI2015, Sintel (clean), and Sintel (final). Thus we believe that our claim of “comprehensive” is justified. To the best of our knowledge, this is the most comprehensive optical flow benchmark to date. Please refer to the appendix for all the results.
>
>
>
> Please do let us know if you have any further questions or concerns, or if you feel that we missed addressing any of your questions or concerns, we would be happy to address them.
>
> Best Regards
>
> Authors of Paper #1055
>
>
>
> References:
>
> [1] Wang, Bo, Yifan Zhang, Jian Li, Yang Yu, Zhenping Sun, Li Liu, and Dewen Hu. "Splatflow: Learning multi-frame optical flow via splatting." International Journal of Computer Vision (2024): 1-23.
>
> [2] Zhang, Zhiyong, Huaizu Jiang, and Hanumant Singh. "NeuFlow: Real-time, High-accuracy Optical Flow Estimation on Robots Using Edge Devices." arXiv preprint arXiv:2403.10425 (2024).

---

> > ### Comment · Reviewer_F85d · 2024-11-20
> > **Thanks for the responses**
> >
> > The authors have addressed some of my concerns regarding data diversity. However, several fundamental issues remain unresolved. For instance, regarding data accessibility, the authors could provide the proportion of data used during submission to facilitate a more thorough review of the work. Additionally, as Reviewer GPBg noted, "the evaluation is messy." The rebuttal does not provide convincing information or sufficient evidence of comparisons with more state-of-the-art (SOTA) methods.

---

> ### Author Response · Authors · 2024-11-20
>
> Dear Reviewer F85d,
>
> Thank you very much for your reply.
>
> Anticipating requests for data accessibility, we have included all individual experimental results in the appendix, additionally, we have included samples of perturbed and corrupted images in the appendix. We request the reviewer to please confirm if this is the data being referred to or if we have misunderstood the request?
>
> Regarding the reviewer's suggestion to include “more state-of-the-art (SOTA) methods”, to the best of our knowledge we have now included all peer-reviewed state-of-the-art (SOTA) optical flow estimation methods available publicly.
> Would the reviewer have suggestions of any particular optical flow method that we might have overlooked?
>
> As acknowledged by Reviewer MVeY, “The writing is overall in good quality.”, and Reviewer hv2r, “The paper is written clearly so that it's easy to follow and understand many graphs, metrics, and analyses in the paper.”, we have attempted to show that compared to the errors on i.i.d. Data, the errors under attack and errors under image corruptions are significantly higher, making the differences between methods hardly significant and thus demonstrating an alarming threat to these methods if deployed in the real world. Additionally, this work has more significant contributions, such as those highlighted in the “Summary to First Author Response”:
>
> First, we demonstrate the need for FLOWBENCH, the lack of a framework to study the reliability and generalization abilities of optical flow methods alongside their i.i.d. performance has led to the unfortunate reality of methods proposed over time only improving the i.i.d. performance and not their reliability and generalization abilities. As commonly used, reliability is the robustness against white-box adversarial attacks, as they serve as a proxy for the worst-case scenario. Generalization ability is the robustness to image corruptions, specifically 2D and 3D Common Corruptions as they serve as a proxy for real-world domain shifts and signal corruptions. The lack of reliability and generalization ability is severely alarming, as the error of optical flow estimation methods in i.i.d. evaluations is in the low single digits, whereas the error of methods in evaluations against adversarial attacks and common corruptions is in the high hundreds!
>
> Second, in Sec. 5.1, we find that there is a high correlation in the performance of optical flow estimation methods against targeted attacks using different targets, thus saving compute resources for future works as they need to evaluate only against one target. Please note that evaluations using adversarial attacks are extremely expensive both time-wise and computation-wise. Thus, the impact of this finding cannot be underscored.
>
> Third, in Sec. 5.2, we observe the methods known to be SotA on i.i.d. samples are not reliable, and do not generalize well to image corruptions, demonstrating the gap in current research when considering real-world applications. Additionally, we observe here that there is no apparent correlation between generalization abilities and the reliability of optical flow estimation methods.
>
> Fourth, in Sec. 5.3, we show that methods using attention-based pointing matching are marginally more reliable than methods using other matching techniques, while methods using CNN and Cost Volume-based matching have marginally better generalization abilities. Please note that observations three and four together help us conclude that based on current works, different approaches might be required to attain reliability under attacks and generalization ability to image corruptions.
>
> Fifth, in Sec. 5.4, we show that, unlike image classification, increasing the number of learnable parameters does not help increase the robustness of optical flow estimation methods.
>
> Sixth, in Sec 5.5, we show that white-box adversarial attacks on optical flow estimation methods can be independent of the availability of ground truth information, and can harness the information in the initial flow predictions to optimize attacks, thus overcoming a huge limitation in the field.
>
> Such an in-depth understanding of reliability and generalization abilities to optical flow estimation methods can only be obtained using our proposed FLOWBENCH. We are certain that FLOWBENCH will be immensely helpful to gather more such interesting findings and its comprehensive and consolidated nature would make things easier for the research community.
> We would be happy to include any specific suggestions the reviewer might have to improve the manuscript to better put across our takeaways.
>
> Please do let us know if we successfully answered your questions and concerns, or if you have any further questions or concerns, we would be happy to address them.
>
>
>
> Best Regards
>
> Authors of Paper #1055

---

> > ### Comment · Reviewer_F85d · 2024-11-23
> >
> > Thank you for your responses. One of my primary concerns is that the authors did not provide empirical results (training logs, weights, data) in the original submission. This omission prevents a thorough evaluation of the work and represents a significant shortcoming in what is claimed to be a “comprehensive evaluation.” Without access to the results, reviewers cannot properly assess the validity or significance of the findings, which undermines the paper’s claims.
> >
> > Additionally, while the paper reports correlations between performance, reliability, and generalization ability, it fails to establish whether these correlations imply causation. Without experimental evidence isolating causal factors, these findings remain less actionable for informing future advancements in optical flow estimation. This issue is compounded by generalized claims such as “methods that are SotA on i.i.d. are remarkably less reliable,” which lack sufficient qualifiers. It is possible that some SotA methods could achieve higher reliability under specific configurations that the authors did not explore. Such overgeneralizations weaken the conclusions, and a more cautious presentation with detailed caveats would lend credibility to the analysis.
> >
> > Another important omission is the lack of benchmarking for non-deep-learning methods. The authors themselves acknowledge that these methods might demonstrate greater robustness to adversarial attacks. Excluding traditional approaches limits the study’s scope and misses an opportunity to compare DL-based methods with their predecessors. Including this comparison would provide a more comprehensive perspective on the relative strengths and weaknesses of different approaches, addressing an important gap in the field.
> >
> > The proposed FLOWBENCH framework is ambitious and impressive in scale, but its resource-intensive nature—requiring over 4,500 experiments—may discourage adoption by researchers with limited computational resources. Furthermore, the reliance on synthetic datasets and predefined perturbations raises concerns about the framework’s ability to generalize to real-world scenarios with unseen corruptions. Incorporating real-world data with naturally occurring noise and disturbances would enhance the benchmark’s practical relevance. Similarly, the introduction of new metrics, such as Generalization Ability Error (GAE) and Reliability Error (NARE/TARE), is a promising contribution, but the paper lacks sufficient justification for their superiority over traditional metrics like mean End-Point Error (EPE). A comparative evaluation demonstrating the unique value of these metrics would strengthen their utility and encourage adoption within the research community.
> >
> > The reliance on ground truth data for optimizing adversarial attacks is also identified as a limitation. While the authors propose optimizing attacks against initial predictions as an alternative, this proxy may not fully capture robustness, potentially weakening the evaluation. A more thorough validation of this approach would be necessary to ensure its reliability. Finally, while the authors commit to open-sourcing FLOWBENCH, the framework’s complexity and significant resource demands may hinder its widespread adoption, especially if documentation is insufficient. Clear, well-structured documentation and streamlined instructions would be essential to support the research community in leveraging this tool effectively.

---

> > > ### Author Response · Authors · 2024-11-23
> > > **Clearing Misunderstandings by Reviewer F85d**
> > >
> > > Dear Reviewer F85d,
> > >
> > > We understand that reviewing is a difficult process, especially with other concurrent deadlines and we highly appreciate the effort by the reviewer to engage in discussion. We believe there are misunderstandings on the part of the reviewer regarding our proposed work. We humbly urge the reviewer to re-evaluate their stance and consider the amount of effort invested in a work of this magnitude and the huge benefit this work would be for the community.
> > > We take this opportunity to answer specific concerns raised by the reviewer.
> > >
> > >
> > > 1. All experimental results are available in Appendix H of the paper as plots. We have reported all evaluations used for the analysis. We request the reviewer to please look at these numbers. As far as sharing log files is concerned, hardly any double-blind submission ever does, as this helps maintain the integrity of the double-blind reviewing process. The validity of the findings can easily be judged with the evaluation results already included in Appendix H, and the paper's claims stand.
> > >
> > > 2. We show in our experiments that methods with high i.i.d. performance have low reliability and generalization ability. While doing so, we do not claim causation. We believe the reviewer has misunderstood this point. We discuss that reliability and generalization ability are important for safety in the real world. However, methods that are SotA on i.i.d. are not reliable. This is because optical flow works proposed over the years have not focused on these aspects, their primary focus has been improving i.i.d. performance, in doing so, we as a community have overlooked the reliability and generalization abilities of proposed optical flow methods and we attempt to draw focus to these with our work while providing an easy-to-use benchmarking tool. Additionally, we currently do not understand the reviewer’s comment “It is possible that some SotA methods could achieve higher reliability under specific configurations”, to the best of our knowledge we have evaluated methods against all considered configurations, could the reviewer please elaborate on potential configurations to which the reviewer is referring?
> > >
> > > 3. The last time any top-venue peer-reviewed optical flow estimation work required evaluating a non-DL-based optical flow method was when FlowNet [R1] and Flownet2 [R2] were proposed, back in 2015 and 2016, about a decade ago! Only in light of the findings and observations of our proposed work thus one hypothesizes that maybe one possible approach to decrease this gap between i.i.d. performance and reliability and generalization abilities would be to retrospectively analyze non-DL-based optical flow methods to ascertain their reliability and generalization abilities. Making such an analysis a very interesting **"extension"** of our proposed work. We believe the reviewer might be overlooking the complexity and time required to analyze non-DL-based approaches under a similar setting. Such a work, in our most humble opinion, would warrant a top-tier publication of its own.
> > >
> > > 4. We propose FLOWBENCH to precisely help the community, especially researchers with access to only limited computing resources. We believe there to be a misunderstanding on the part of the reviewer. Given that we have already performed over 4500 evaluations using FLOWBENCH, we provide these evaluations to everyone, to use as a simple tabular benchmark that can be queried using very easy commands that have already been well documented in Appendix G. We urge the reviewer to please have a look at the appendix!

---

> > > > ### Author Response · Authors · 2024-11-23
> > > > **Continuation of Clearing Misunderstandings by Reviewer F85d**
> > > >
> > > > 5. Thank you for acknowledging that the metrics are a promising contribution. GAE, NARE, and TARE are not meant to be superior to EPE, they are essentially the EPE values (thus a low value is good)! GAE and NARE are the maximum EPE values for a given method across evaluations against different common corruptions (for GAE), and different non-targeted adversarial attacks (for NARE) respectively. In the case of targeted attacks, for consistency (to maintain a low value is good), we use the maximum negative EPE with respect to the target, across all targeted adversarial attacks as NARE. Given that multiple evaluations are being performed, our objective behind using the maximum EPE is to ascertain "How bad or unreliable can a method become?", GAE, NARE, and TARE help us do exactly this. We urge the reviewer to please read Section 4 of the paper where we define the metrics and justify their existence very well.
> > > >
> > > > 6. Regarding, real-world domain shifts, as mentioned in our earlier response there exists no dataset captured in the wild with corruptions and domain shifts that also contain ground truth for optical flow estimation, however, this effort has been made for semantic segmentation with the ACDC dataset. In Appendix A we discuss how for semantic segmentation, synthetic image corruptions (very similar to those used in FLOWBENCH) accurately represent the possible domain shifts and corruptions that can be captured in the wild.
> > > >
> > > > 7. We show in Figure 6, that when using initial flow prediction as a proxy for ground truth for optimizing white-box adversarial attacks, the trends are identical to when using real ground truth.
> > > > Please note that top-tier venue publications for white-box adversarial attacks on pixel-wise prediction tasks like CosPGD, PCFA, SegPGD [R3], and Adversarial Weather used 10 or fewer models to evaluate their attack and claim to be SotA when proposed.
> > > > This is because white-box adversarial attacks, especially for pixel-wise prediction tasks are very expensive to compute, and thus following this norm we too evaluate using 10 different models, from different families and training strategies to show the generalized nature of our finding.
> > > > We would be glad to receive feedback if the reviewer now agrees with our claim.
> > > >
> > > > 8. We have included sufficient documentation in Appendix F and G to use FLOWBENCH, moreover, FLOWBENCH is built upon ptlflow, an open-source project. We urge the reviewer to visit their website: https://ptlflow.readthedocs.io/en/latest/index.html and please let us know if the reviewer believes there is any documentation lacking.
> > > >
> > > >
> > > > Please do let us know if we successfully answered your questions and concerns, or if you have any further questions or concerns, we would be happy to address them.
> > > >
> > > > Best Regards
> > > >
> > > > Authors of Paper #1055
> > > >
> > > >
> > > > References:
> > > >
> > > > [R1] Fischer, Philipp, Alexey Dosovitskiy, Eddy Ilg, Philip Häusser, Caner Hazırbaş, Vladimir Golkov, Patrick Van der Smagt, Daniel Cremers, and Thomas Brox. "Flownet: Learning optical flow with convolutional networks." arXiv preprint arXiv:1504.06852 (2015).
> > > >
> > > > [R2] Ilg, Eddy, Nikolaus Mayer, Tonmoy Saikia, Margret Keuper, Alexey Dosovitskiy, and Thomas Brox. "FlowNet 2.0: Evolution of Optical Flow Estimation with Deep Networks." arXiv preprint arXiv:1612.01925 (2016).
> > > >
> > > > [R3] Gu, Jindong, Hengshuang Zhao, Volker Tresp, and Philip HS Torr. "Segpgd: An effective and efficient adversarial attack for evaluating and boosting segmentation robustness." In European Conference on Computer Vision, pp. 308-325. Cham: Springer Nature Switzerland, 2022.

---

> > > > > ### Comment · Reviewer_F85d · 2024-11-26
> > > > >
> > > > > Please refrain from using the term "misunderstandings," as the authors' responses did not directly address my concerns. At this stage, I doubt the authors are capable of addressing them adequately. The core contribution is the FLOWBENCH benchmark, which is NOT included in the work. I cannot evaluate any of the claims by scrutinizing the FLOWBENCH repository. Without this information, the paper is nothing more than an empirical report without proof. To be fair, the authors have included much informative content in the paper, but they need to provide a more polished and complete product before resubmission. This concludes my evaluation.

---

> ### Comment · Reviewer_hv2r · 2024-11-27
>
> Thanks for the discussion, both Reviewer F85d and authors.
>
> I partially agree with Reviewer F85d. Here are some points that I agree with:
>
> * Data accessibility
>
>   Though Appendix H includes raw data, it's very hard to parse the information. Also, there are so many methods in Fig. 2 that it takes time to digest the information and find corresponding methods in the legend (it's good to be comprehensive, though). It will be much better to have a benchmark website where people can interactively sort methods by each criterion. Will it be possible to prepare an early version of an interactive benchmark website during the rebuttal period?
>
> * Adding more recent SOTA methods
>
>   When quickly looking at the Sintel benchmark, yes, some recent methods are missing from the paper, such as CroCo-Flow (ICCV 2023), FlowDiffuser (CVPR 2024), MemFlow (CVPR 2024), but there can be more. (~CVPR 2024 papers can be considered as unpublished as the ICLR deadline was in May.~) However, I believe that the new methods can be added easily by running the established library. Is that right?
>
> However, I disagree with these points:
>
> * Empirical report
>
>   Yes, the paper is a bit closer to an empirical report. It doesn't discuss the causation of each design factor but rather talks about correlation at a method/checkpoint level. However, I think this is also one of the things that the paper teaches us: after looking at the results in the paper, now we know that we also need analyses at an architectural component level under a controlled setup. It's a bit hard to realize that before reading this paper. To have a better robustness benchmark, we gotta start from somewhere I think. In-depth robustness analyses under controlled configuration sound an interesting direction, by the way as a whole new paper.
>
> * 4,500 experiments
>
>   If I understood correctly, these are already done, and people don't have to run by themselves again. If a new method comes, it can only run the new one and include it in the benchmark I think.
>
>
> * Adding non-DL methods
>
>   Yes, it's really interesting to see how the non-dl methods behave. However, in terms of EPE, I don't remember what the best-performing one was on Sintel or KITTI. Maybe the last one can be around 2017/2018. Could you name some methods if they shouldn't be missed in the comparison? If needed, this paper can frame itself as a robustness benchmark that compares DL methods exclusively. Also, one could also write a new paper for a comprehensive robustness comparison between DL vs non-DL methods.
>
>
> I hope Reviewer F85d has more time or bandwidth to discuss these points instead of concluding the evaluation early. I hope to hear your thoughts on those.

---

> ### Comment · Reviewer_F85d · 2024-11-27
>
> Reviewer hv2r,
>
> As reviewers, it is our responsibility to uphold the standards of the community and ensure the integrity of the reviewing process. Our role is to assess submissions critically and recommend disqualification for work that does not meet the rigorous standards of ICLR, a flagship conference in AI. While I welcome and respect differing opinions, it is not appropriate to instruct a fellow reviewer on how to proceed. Such decisions ultimately rest with the Area Chair (AC).
>
> Regarding this particular submission, I have consistently pointed out in my comments that the core value of the work is undermined by the absence of critical elements such as data accessibility (adversarial samples, empirical weights, and logs). Without this information, it is impossible to reproduce or rigorously examine the results. The authors’ arguments have not convinced me otherwise, as the empirical data forms the foundation of the work’s value. Understanding factors that impact algorithmic robustness is important, but without access to supporting data, the validity of the conclusions cannot be verified.
>
> I am trying to understand your motivation in encouraging me to change my judgment. Do you believe this is an exceptional work that must be shared with the community despite these shortcomings? If so, I would question your understanding of the task and the standards of ICLR. Alternatively, if there are other reasons, I am open to hearing them.
>
> Until then, I firmly believe that this submission requires further refinement before it is ready for consideration.

---

> ### Author Response · Authors · 2024-11-27
>
> Dear Reviewer hv2r,
>
> Thank you very much, we deeply appreciate your efforts for not just providing a sound and thorough review with a positive evaluation of our work, but also positively engaging in further discussions.
>
> **Data accessibility**
>
> We agree that making meaningful observations with plots when performing a study as large scale as the one provided by us can be difficult. In our experience, including tables for the numbers would have been even more difficult to read thus we opted for plots. We completely agree that an interactive website is the way to go, anticipating this we already have an initial version of the website ready, and we would have it hosted with the camera-ready submission. The domain we have for the website unfortunately reveals our identity, and thus to safeguard the sanctity of the double-blind review process, we chose to not share the URL.
> We share a screenshot from a very initial prototype of this website in Appendix I (latest revision).
> Additionally, we now created an identity leak-proof anonymous Google sheet with multiple sheets that have values from our evaluations, that we shared with the chairs. Sharing these results publicly will allow anyone to easily reproduce and steal all our evaluations which is why we refrain from publicly sharing this document before acceptance.
> We believe that if required, one can already verify our evaluations using Appendix H.
>
>
> **Adding more recent SOTA methods**
>
> Yes, your understanding is correct, “new methods can be added easily by running the established library”.
> We agree that including these could enhance the comprehensiveness of the work, but as acknowledged by you (Reviewer hv2r) and Reviewer MVeY, there will always be new methods coming up. The only reason we could not include these in our evaluations is because they are currently not reliably supported by PTLFLow as they are fairly new and integrating new methods is a time-consuming process, and, PTLFlow mentions “PTLFlow is still in early development, so there are only a few models available at the moment, but hopefully the list of models will grow soon.” Post acceptance, we hope to publicly aid in this endeavor.
>
> **Empirical report**
>
> Thank you very much for acknowledging the contributions of this work.
>
> **4,500 experiments**
>
> Yes, we attest that your understanding is correct.
>
>
> **Adding non-DL methods**
>
> Thank you very much for acknowledging that “one could also write a new paper for a comprehensive robustness comparison between DL vs non-DL methods”. Currently, we attempt to frame the paper towards DL-based optical flow methods, while in favor of saving space, we often refer to methods as just “optical flow estimation methods” or “optical flow methods”, in some cases like the abstract and introduction, we call them “learning-based optical flow estimation methods” and “DL-based optical flow estimation methods”. We begin the motivation of this work by mentioning the vulnerability of DL-based methods towards adversarial attacks and common corruptions, and then we extend the motivation to have safer DL-based optical flow estimation methods.
> Nonetheless, if the reviewers suggest, we can try to align the paper more towards DL-based optical flow methods and mention so more exclusively.
>
> Best Regards
>
> Authors of Paper # 1055

---

> > ### Comment · Reviewer_hv2r · 2024-11-27
> >
> > (nit) my apologies for being mistaken; CVPR conference date was before the ICLR deadline on September, so CVPR papers also considered as published.

---

> > > ### Author Response · Authors · 2024-11-27
> > >
> > > Dear Reviewer hv2r,
> > >
> > > Thank you, we have now revised our reply accordingly.
> > >
> > > Best Regards
> > >
> > > Authors of Paper #1055

---

> ### Comment · Reviewer_hv2r · 2024-11-28
>
> Reviewer F85d,
>
> My apology if my response sounds like instructing a fellow reviewer on how to proceed. My intention was more like trying to open a discussion with a fellow reviewer as "this concludes my evaluation" sounded like "the end of discussion" to me. I wanted to hear more details on the thoughts that are different from mine and doublecheck if there are some points that I missed. And I am glad that the discussion can be continued.
>
> What are the adversarial samples, empirical weights, and logs? If I understood correctly, the paper grabbed over 90 pretrained checkpoints, performed each adversarial attack, and evaluated the metric. Does adversarial samples mean that corrupted/attacked images of each method? Do empirical weights are pretrained checkpoints of all methods used in the benchmark? Are logs the ones from evaluation processes? If that's the case, I think authors can simply attach those data in the supplementary so that we all can have a look. This will ensure transparency of the evaluation. If not, this concern will still remain.
>
> Again, I am not encouraging you to change your judgment but just wanted to hear your thoughts and try to understand the shortcomings of the paper for a better judgment. I would just appreciate it if you are willing to discuss something even without changing the rating. I think this paper is not an exceptional work but rather a good seed work that people can have a look at the extensive benchmark, compare almost all (DL-based) existing optical flow methods, and hopefully design more robust architectures, and conduct more improved analyses or benchmarking experiments, based on this paper.
>
> If only exceptional works that must be shared with the community are the papers accepted at ICLR, probably you are right that I have different standards on ICLR.
>
> Best,
>
> Reviewer hv2r

---

> > ### Comment · Reviewer_F85d · 2024-12-01
> >
> > Reviewer hv2r,
> > Thank you for your clarification. I want to apologize for my choice of verbiage and the seemingly incorrect assumptions about your message. Reviewing misconduct, particularly in terms of nepotism, is a significant issue for conferences like ICLR and NIPS, and I am very frustrated by it.
> >
> > I agree that this work provides some insights into improving model robustness from an adversarial perspective. However, I must emphasize that this is the only value I see in this work. Despite presenting numerous results, the authors failed to prepare a demo repository to reproduce even some of these results, which is quite concerning. Furthermore, some adversarial samples (e.g., Fig. 8) offer limited insights for the audience to understand the broader patterns related to systemic robustness and adversarial effects.
> >
> > With that said, I do not believe this paper is ready for publication.

---

> ### Author Response · Authors · 2024-11-28
> **Sharing Some Model Weights as Requested by Reviewer F85d**
>
> Dear Reviewer F85d,
>
> As requested, following we share some model weights in FLOWBENCH and their respective sources.
> Please note, FLOWBENCH itself offers many more combinations.
> As we mentioned in the paper and mentioned by Reviewer hv2r, we have not trained the models ourselves, thus for the training implementation we share their respective GitHub repositories.
>
> Please do let us know if this answers your concerns.
>
>
> | Method | Dataset Finetuned On | Link to Model Weights | Link to GitHub |
> | --- | --- | --- | --- |
> | CCMR | KITTI 2015 | https://drive.google.com/file/d/1QbjV_W_6Kiog-JxuEzHTTyzw0C0G0-fx/view?usp=sharing | https://github.com/cv-stuttgart/CCMR |
> |  | MPI Sintel | https://drive.google.com/file/d/1qCZKXcyL11oPhj4BX5tkIDvtV3_EEJ6f/view?usp=drive_link |  |
> | CRAFT | KITTI 2015 | https://drive.google.com/file/d/1081nT4xfqHKLjUCk6DlDuN_P_DzCn-Wn/view?usp=drive_link | https://github.com/askerlee/craft |
> |  | MPI Sintel | https://drive.google.com/file/d/1seMmHwNeUpLmrH4ON7GcfXc5OomnjxMn/view?usp=drive_link |  |
> | CSFlow | KITTI 2015 | https://drive.google.com/file/d/1UPfJc92tvapTe9jw-i3cZ8lL3TYzaLRF/view?usp=drive_link | https://github.com/MasterHow/CSFlow |
> | DICL-Flow | KITTI 2015 | https://drive.google.com/file/d/1D0ZksIg6Nf7NtQwSM5KK0bk7tl7bIShQ/view?usp=drive_link | https://github.com/jytime/DICL-Flow |
> |  | MPI Sintel | https://drive.google.com/file/d/11Hp0CMpxh2AYYkiel1IlT6ZCXJtmDaeJ/view?usp=drive_link |  |
> | DIP | KITTI 2015 | https://drive.google.com/file/d/1uPGHm5e2gwgCxvFc8KEHRlr7tWhn0ckI/view?usp=drive_link | https://github.com/zihuazheng/DIP |
> |  | MPI Sintel | https://drive.google.com/file/d/1jQFd_Es2_KCXdMxGHb6SZ7HheH9aqeKJ/view?usp=drive_link |  |
> | FastFlownet | KITTI 2015 | https://drive.google.com/file/d/1uqdek7ToPcMK6QVkEspxX4x7R6fsQMue/view?usp=drive_link | https://github.com/ltkong218/FastFlowNet |
> |  | MPI Sintel | https://drive.google.com/file/d/1TLWFPBlJ8g5SwSLPz5Yiy1QxiR1z9fNM/view?usp=drive_link |  |
> | Flow1D | KITTI 2015 | https://drive.google.com/file/d/1I5bCSpwSN2JYtaILx72t6rMg_syFWK9T/view?usp=drive_link | https://github.com/haofeixu/flow1d |
> |  | MPI Sintel | https://drive.google.com/file/d/1GkgQrc-HKhM3hSYkrTBJAn8eGz_CSm_h/view?usp=drive_link |  |
> | FlowFormer | KITTI 2015 | https://drive.google.com/file/d/1z_dp7kLzv9zXUSAxHBtP8jMeQtV7RRmf/view?usp=drive_link | https://github.com/drinkingcoder/FlowFormer-Official |
> |  | MPI Sintel | https://drive.google.com/file/d/1zP4cmmwyCZzBTcoWsVVaaGR5txh6c9Ft/view?usp=drive_link |  |
> | FlowFormer++ | KITTI 2015 | https://drive.google.com/file/d/1sddn8AmbXgKls-zEQCKeMsZKfuB54ue_/view?usp=drive_link | https://github.com/XiaoyuShi97/FlowFormerPlusPlus |
> |  | MPI Sintel | https://drive.google.com/file/d/1U0gi2uGdcPC56NULuzB6Gmq9v9OSO_Pp/view?usp=drive_link |  |
> | GMA | KITTI 2015 | https://drive.google.com/file/d/1rk50s_kSMj6xq2b9uQmy0i5awbuxSGEZ/view?usp=drive_link | https://github.com/zacjiang/GMA |
> |  | MPI Sintel | https://drive.google.com/file/d/1GLWRlKGmBdXbuTDaSXXGM9sObcdflIDi/view?usp=drive_link |  |
> | GMFlow | KITTI 2015 | https://drive.google.com/file/d/1HQGMRoqskfcptRPx3hoPPd4esAp6_CiY/view?usp=drive_link | https://github.com/haofeixu/gmflow |
> |  | MPI Sintel | https://drive.google.com/file/d/1XBeb6gNO_pFZkg360hn6TOLAUGqnC5TL/view?usp=drive_link |  |
> | GMFlowNet | KITTI 2015 | https://drive.google.com/file/d/1LAjrQuAfDIVNv6lJLK3UwmI4shyw7iei/view?usp=sharing | https://github.com/xiaofeng94/GMFlowNet |
> | HD3 | KITTI 2015 | https://drive.google.com/file/d/1e39YJavE3eSRUzy1Ay_3VZP3BWnrrkoB/view?usp=drive_link | https://github.com/ucbdrive/hd3 |
> |  | MPI Sintel | https://drive.google.com/file/d/130a31H8Y_SM6lQOsK-Sv8Ksy52QpONIP/view?usp=drive_link |  |
> | IRR-PWCNet | KITTI 2015 | https://drive.google.com/file/d/1-Lc2CIYFZKFJxV-oVUyYfn8Wah8guUPB/view?usp=drive_link | https://github.com/visinf/irr |
> |  | MPI Sintel | https://drive.google.com/file/d/1cZ_4YRYSepEp908cIhJsQL0GQB3ikdi7/view?usp=drive_link |  |
> | LiteFlowNet | KITTI 2015 | https://drive.google.com/file/d/192URC5eKQmGWFB1gQheB4Y1N8JTldNng/view?usp=drive_link | https://github.com/twhui/LiteFlowNet |
> |  | MPI Sintel | https://drive.google.com/file/d/1Oxo2vlXL8ZhW65_wFgf0FS24wPK-pN8C/view?usp=drive_link |  |
> | LiteFlowNet2 | MPI Sintel | https://drive.google.com/file/d/120WZz3U_jnNwSqOneWyJigPS3fKsV2P8/view?usp=drive_link | https://github.com/twhui/LiteFlowNet2 |
> | LiteFlowNet3 | KITTI 2015 | https://drive.google.com/file/d/1-Jr2Kty7qp-4BqIbJ3ut0N1IOcMtxeS9/view?usp=drive_link | https://github.com/twhui/LiteFlowNet3 |
> |  | MPI Sintel | https://drive.google.com/file/d/1AafQpWIEnV0_5bEtna4qcQjzZBirrxX7/view?usp=drive_link |  |
> | LLA-Flow | KITTI 2015 | https://drive.google.com/file/d/1THChtnmxKZ1lOeW8clBGoMqu0ff6GQcX/view?usp=drive_link | https://github.com/mansang127/LLA-Flow |
> |  | MPI Sintel | https://drive.google.com/file/d/1BDULMcJr8GC1IySHR4feQW30OFLgbh4o/view?usp=drive_link |  |

---

> ### Author Response · Authors · 2024-11-28
> **[contd.] Sharing Some Model Weights as Requested by Reviewer F85d**
>
> | Method | Dataset Finetuned On | Link to Model Weights | Link to GitHub |
> | --- | --- | --- | --- |
> | MaskFlownet | KITTI 2015 | https://drive.google.com/file/d/1jU4O7TIygR_RVwO1QTZvLUoTvxzZu-3t/view?usp=drive_link | https://github.com/cattaneod/MaskFlownet-Pytorch |
> |  | MPI Sintel | https://drive.google.com/file/d/1HY-93iv14k7zJIknKRctSCN9Sbd5nhRA/view?usp=drive_link |  |
> | MaskFlowNetS | MPI Sintel | https://drive.google.com/file/d/1eaCU19554qh4Vyx3hjjwITfGjkrwPkpM/view?usp=drive_link | https://github.com/cattaneod/MaskFlownet-Pytorch |
> | MatchFlow | KITTI 2015 | https://drive.google.com/file/d/1XPCj88fXqLTgz9KLxuwnM088eJWP7z3i/view?usp=drive_link | https://github.com/DQiaole/MatchFlow |
> |  | MPI Sintel | https://drive.google.com/file/d/10oKBWTfzQl3j8TGoSKbibGKVCzIcrjba/view?usp=drive_link |  |
> | MS-RAFT+ | Kitti-2015 + MPI Sintel | https://drive.google.com/file/d/1xYkYWefufntC7rrwR9hOB_5k-5BWgNGI/view?usp=drive_link | https://github.com/cv-stuttgart/MS_RAFT_plus |
> | NeuFlow | MPI Sintel | https://drive.google.com/file/d/1LatcZ8U-U6hjxKTf9rVtmriqvvSUH4jX/view?usp=drive_link | https://github.com/neufieldrobotics/neuflow |
> | PWCNet | MPI Sintel | https://drive.google.com/file/d/1meJiGp0M2o05BIof4n10bssqB2oG80lE/view?usp=drive_link | https://github.com/NVlabs/PWC-Net |
> | RAFT | KITTI 2015 | https://drive.google.com/file/d/1w6FDNab3HnhzFLggJqWRF4PYAkUis75d/view?usp=drive_link | https://github.com/princeton-vl/RAFT |
> |  | MPI Sintel | https://drive.google.com/file/d/1ivfqQXciHRS2bt5VN9L7nEJDTY6SeHqR/view?usp=drive_link |  |
> | RAPIDFlow | KITTI 2015 | https://drive.google.com/file/d/1gwZOFF-VdsDIqYw29XBG7fJDB5gBWXf1/view?usp=drive_link | https://hmorimitsu.com/publication/2024-icra-rapidflow/ |
> |  | MPI Sintel | https://drive.google.com/file/d/15Sq1dH-tKPVmpIexbtSDlm2OBeFcaIny/view?usp=drive_link |  |
> | RPKNet | KITTI 2015 | https://drive.google.com/file/d/1sQZcv3inwIh98Z02pouLqtTWbafXpta2/view?usp=drive_link | https://hmorimitsu.com/publication/2024-aaai-rpknet |
> |  | MPI Sintel | https://drive.google.com/file/d/1cXXcT_8lbW-FeqkovF47z16CESDff95T/view?usp=drive_link |  |
> | ScopeFlow | KITTI 2015 | https://drive.google.com/file/d/192BYU-tV8IDqosD_4ckbaN2X1NfvGyIQ/view?usp=drive_link | https://github.com/avirambh/ScopeFlow |
> |  | MPI Sintel | https://drive.google.com/file/d/1sI9FPfLBCapEzA6SGaDDx1LdHwdeQdkV/view?usp=drive_link |  |
> | SKFlow | KITTI 2015 | https://drive.google.com/file/d/1oFdsnbmXVsluxrymaOp9jrwM2RQiBfYU/view?usp=drive_link | https://github.com/littlespray/SKFlow |
> |  | MPI Sintel | https://drive.google.com/file/d/1dv2856zwjLsFZHnUauVL8o8Et7sqR6QB/view?usp=drive_link |  |
> | SplatFlow | KITTI 2015 | https://drive.google.com/file/d/1SwfVNqQrrg1RwkaDEOorT1CO_bH_vRNc/view?usp=drive_link | https://github.com/wwsource/SplatFLow |
> | STaRFlow | KITTI 2015 | https://drive.google.com/file/d/1V48XDSeQQrFYV-Z6iGXZQREGLWXme3gk/view?usp=drive_link | https://github.com/pgodet/star_flow |
> |  | MPI Sintel | https://drive.google.com/file/d/1W8oywvlGpzgeeQzD_IOFkMEcLJJTqOqH/view?usp=drive_link |  |
> | VideoFlow | KITTI 2015 | https://drive.google.com/file/d/1tTjYZquuHWx_YovCOwEr5BXYT5DGr7It/view?usp=drive_link | https://github.com/XiaoyuShi97/VideoFlow |
> |  | MPI Sintel | https://drive.google.com/file/d/1aKxF19YgcrNviZtB6w5cV9LRV27sm4Me/view?usp=drive_link |
>
>
> Best Regards
>
> Authors of Paper #1055

---

> ### Author Response · Authors · 2024-11-30
> **More Data needed?**
>
> Dear Reviewer F85d,
>
> Do these model weights clarify the issue? We would appreciate your letting us know whether more data is needed.
> Regarding plain evaluation files and logs, we hope you understand that we can not post them openly on openreview prior to acceptance. It would allow anyone to copy our entire paper. We have, however, made sheets with all plain evaluation outcomes available to the AC to prove that our results are valid and reproducible. Further, we will, of course, make everything publicly available upon acceptance. Screenshots of the respective website can already be viewed in the current revision.
>
> Thank you for the discussion and for letting us know whether anything more is required from your side!
>
> Best Regards,
>
> Authors of Paper # 1055

---

> > ### Comment · Reviewer_F85d · 2024-12-01
> >
> > Thanks for your effort. But these are not what I am looking for.

---

> > > ### Author Response · Authors · 2024-12-02
> > > **Codebase used for FLOWBENCH**
> > >
> > > Dear Reviewer F85d,
> > >
> > > https://anonymous.4open.science/r/flowbench-1055/ contains the base code for the benchmarking tool FlowBench. For security purposes, and for protecting anonymity we only share some initial demo code to reproduce our results and not the benchmarking tool commands itself.
> > >
> > > We include commands on how to use the codebase in its current form.
> > >
> > > **Please note that the proposed FlowBench benchmarking tool is merely a wrapper on this codebase.**
> > >
> > > We appreciate the responses by Reviewer F85d, however, we admit that at this point, we are not entirely sure what the reviewer is asking for. In particular, mentioning “Thanks for your effort. But these are not what I am looking for.” did not help us better understand the needs.
> > >
> > > To the best of our understanding, you raised the following concerns that we have now addressed:
> > >
> > > 1. Robustness evaluation (original review): Addressed on 19th November.
> > >
> > > 2. Evaluation metrics (original review): Addressed on 19th November.
> > >
> > > 3. Questions regarding predictive insights (original review): Addressed on 19th November.
> > >
> > > 4. Requesting additional evaluations on very recently proposed methods (original review): Addressed in our revision by 23rd November
> > >
> > > 5. Presentation of evaluations (20th November): Addressed in our revised submission on 23rd November.
> > >
> > > 6. Significance of the large-scale evaluations (23rd November):  Addressed on 23rd November.
> > >
> > > 7. Request for empirical results (23rd November): Included in Appendix H and now shared with the AC.
> > >
> > > 8. Request for model weights (23rd November): Addressed on 28th November.
> > >
> > > 9. Request for sample logs (23rd November): Addressed now in this response.
> > >
> > > 10. Request for demo code (1st December): Addressed now in this response.
> > >
> > >
> > > Please let us know if we were successful in answering your concerns, if so, then we would highly appreciate if you would re-evaluate your assessment of our work.
> > >
> > >
> > > Best Regards
> > >
> > > Authors of Paper #1055

---

### Official Review · Reviewer_hv2r · 2024-11-03

**Soundness:** 3
**Presentation:** 3
**Contribution:** 3
**Rating:** 8
**Confidence:** 4

**Summary:**

The paper proposes a FlowBenmark that benchmarks the robustness of 89 learning-based optical flow methods under different types of adversarial attacks and their generalization ability. The method also analyzes what factors can affect the robustness (e.g., the number of parameters, architecture components, etc). The paper promises to make the benchmark open source upon its acceptance.

---
* Justification on the rating

   I support the acceptance of the paper unless any significant flaws are found during the reviewing period. It provides a comprehensive and insightful benchmark on the robustness of optical flow methods. The findings in the paper will benefit the community and help better design of optical flow method in the future. However, it will be great if the paper provides more in-depth technical analyses and answers the questions below for better clarity.

**Strengths:**

* Insightful benchmark

   The paper provides comprehensive analyses of existing learning-based optical flow methods on their robustness. In the optical flow literature, papers mostly have focused on their EPE accuracy only. This paper provides an interesting analysis that methods with the best accuracy are not necessarily the most robust methods. Given comprehensive analyses, readers can understand the pros and cons of each method. I believe the benchmark will certainly benefit the community and help design a better and more robust method for future work.

* Clarity

   The paper is written clearly so that it's easy to follow and understand many graphs, metrics, and analyses in the paper.

**Weaknesses:**

* Concern about fair comparison

   The checkpoints compared in the analyses might not be trained in the same setups, (e.g., data augmentation (photometric, random noise, etc), regularization, the usage of drop-out, etc), which can affect the robustness of the method. It would be ideal if all methods were trained in the same setup, but it's unrealistic and may not be possible. What does the paper think about the current comparison in the context that the methods are trained in different setups? Will it change any discussion that the paper made in the paper?

   Also, can the paper include or list brief details on training setups (e.g., data augmentation, type of regularization, etc.) of each method? Or discussing them as future work (e.g., each method is trained on different setups, but analyses didn't consider this axis yet) would be also fine.

* Less in-depth analyses

  Given the robustness measure, the paper discusses what method shows better generalization, accuracy, or robustness, but it's a bit lacking in what makes them better or more robust. There are analyses of the number of parameters or matching concepts (attention, CNN) at a high level, but it's a bit unclear what technical designs (e.g., architectural choices, domain knowledge, training strategies, etc.) really make these differences. Can the paper discuss more on the technical aspects?

**Questions:**

* Continuing on the question above, if we want to build an optical flow method that is both robust and accurate, what architectural design choices can we make by learning from this benchmark paper? Maybe this is the task of readers, but can the paper provide a brief discussion of any insights or findings while analyzing 89 methods? What can be the best architectural design choices for future models?

* nit
   * In Fig. 2, IRR shows the most reliable method, but the text says that RapidFlow is the most reliable one. There may be a mistake during making a graph or writing the text.
   * In Fig. 1 right, generalization ability seems to improve when discarding one outlier on the right middle (i.e., one point at mid-2022 and GAM 80), when drawing the graph, does the paper discard a few extreme outliers or include all data points?

---

> ### Author Response · Authors · 2024-11-19
>
> Dear Reviewer hv2r,
>
> Thank you very much for your score and your review, we greatly appreciate your input and suggestions.
>
> We have incorporated all your suggested improvements in our revised submission (available now, revisions are in blue), thank you very much for these suggestions.
>
> We would like to take this opportunity to further discuss the concerns you shared:
>
> **Comparability of Evaluated Models**: Yes, we agree that the training setup can play a significant role in the robustness of a method. However, in the case of optical flow methods, they are hardly ever adapted to the target setting since the target application usually does not have ground truth leading one to rely on the available pre-trained models.
> Currently, the training strategies and architectural design choices of different methods are significantly intertwined.
> For example, MS-RAFT+ is trained significantly differently from FlowFormer++, FlowFormer++ can be evaluated on a dataset only if it is finetuned for that dataset (which we did), however, MS-RAFT+ is proposed with a single training strategy to work across datasets. To ensure fair comparison, we contacted the authors of MS-RAFT+ and they accepted that our evaluations would be fair according to them.
> In our current analysis, we attempt to make significant observations based on multiple models, keeping their training and evaluation settings intact. We agree that it would be difficult to disentangle if the performance difference is due to the architectural design choice or due to the training strategy, however, previous works also do not disentangle the two when comparing i.i.d. Performance.
> Given that, optical flow methods were not proposed to specifically overcome adversarial attacks or image corruptions, we hypothesize that training all optical flow with a different known setup would not help improve their reliability and generalization abilities and the current observations from this work would still hold
> Thus, we believe that the robustness evaluations performed here are realistic. To improve the reliability and generalization abilities of optical flow methods, better approaches need to be proposed, that focus on these aspects, and studying these would be very interesting. Thank you for the suggestion, we now include a synopsis of this discussion in future work.
>
> **In-depth analyses and Question**: Thank you very much for raising this question, we now attempt to address this question in the revised conclusion of the work. We discuss that methods using attention-based point matching are marginally more reliable under attack than methods using other matching techniques, while methods using CNN and Cost Volume-based matching have marginally better generalization abilities to image corruptions.
> This in conjecture with another observation, in this work, that there is no apparent correlation between generalization abilities and the reliability of optical flow estimation methods, helps us conclude that (based on current works) different approaches might be required to attain reliability under attacks and generalization ability to image corruptions.
> Please note, here we can talk about “improvements” as merely “marginal” because as seen from the TARE, NARE, and GAE (previously TARM, NARM, and GAM) values, which are essentially EPE values calculated over multiple evaluation methods, there is an absolute lack of robustness in all the optical flow methods. These observations are merely highlighting “the least bad among the worst”. While the EPE values of these methods for i.i.d. Evaluations are in the low single digits, their EPE values for evaluations under adversarial attacks and image corruptions are in the hundreds!
>
> Regarding the comment on Fig. 1: We continue the discussion from the previous question. Indeed there are a few data points belonging to recently proposed methods that lie below the regression line, but we do not manually remove any data point. We observe in Fig. 1 (right), the EPE values across 2D and 3D common corruptions, denoted using GAE, are in the 100s, while the EPE values on i.i.d. datasets are in the low single digits, thus these optical flow methods are very far away from true generalization abilities.
> Ideally, the generalization ability of methods to corruptions would be such that they are completely robust of image corruptions, having the same EPE on i.i.d. and OOD evaluations, however, currently we are very far from this ideal scenario, and thus even with the regression line slightly improving, to the best of our understanding, the observations would still hold.
>
>
> Regarding the comment on Fig. 2: Thank you very much, we indeed made a mistake in the text, we have now corrected this in the revised version.
>
>
> Please do let us know if you have any further questions or concerns, or if you feel that we missed addressing any of your questions or concerns, we would be happy to address them.
>
> Best Regards
>
> Authors of Paper #1055

---

> > ### Comment · Reviewer_hv2r · 2024-11-23
> >
> > Thanks for the response and the paper update! It resolves all of my concerns.
> >
> > I think the paper is already valuable in the sense that it establishes a robustness benchmark for optical flow, casts important questions, and shares analyses by benchmarking nearly a hundred methods.
> > Of course, one can always come up with a better evaluation metric, more realistic evaluation scenarios, and the inclusion of more recent/non-DL methods. However, I think this paper is the first step toward that goal, and we all can improve it as follow-up works.
> >
> > I am still digesting other reviews and comments. I will keep posted after reading them.
> >
> > Best,
> >
> > R hv2r

---

> > > ### Author Response · Authors · 2024-11-27
> > >
> > > Dear Reviewer hv2r,
> > >
> > > Thank you very much for acknowledging that “the paper is already valuable” and we are glad that we could address all your concerns.
> > >
> > > Best Regards
> > >
> > > Authors of Paper #1055

---

### Official Review · Reviewer_MVeY · 2024-11-04

**Soundness:** 3
**Presentation:** 3
**Contribution:** 4
**Rating:** 8
**Confidence:** 5

**Summary:**

This paper proposes FLOWBENCH, a new benchmarking tool designed to evaluate optical flow prediction specifically in terms of reliability (robustness against adversarial attacks) and generalization ability (out-of-distribution performance). Three related metrics are proposed, namely TARM (Targeted Attack Reliability Measure), NARM (Non-targeted Attack Reliability Measure), and GAM (Generalization Ability Measure). An extensive study on 89 released models/checkpoints over four widely used datasets shows in-depth findings and insights.

**Strengths:**

1. This paper is among the first attempts to evaluate optical flow methods on reliability and generalization ability, which are both important but often overlooked in the research community.
2. The effort put into this research is huge. Building an open-source benchmark tool requires a large amount of coding and implementation. The results on all latest models are insightful and worth sharing.
3. The writing is overall in good quality.

**Weaknesses:**

1. The definition of TARM and NARM is not very clear in the text. Are they just defined as the EPE in different settings? The scales of these metrics shown in the figures are also a bit confusing, as EPEs up to the level of hundreds are weird. Maybe the authors should elaborate this more in the main body (not just in the appendix). Maybe it is also better to discuss the specific differences between targeted and non-targeted attacks in the context of optical flow estimation.
2. The benchmarked models are all DL-based models. Maybe it will be better to pick a few SotA traditional methods as a baseline? It is commonly believed that traditional methods are more interpretable and thus more robust, so it may be interesting to see how they compare on this benchmark.
3. One minor point that makes the proposed metrics less accessible and a bit unnatural is: for TARM, the higher the better, but for NARM and GAM, the lower the better. I would suggest aligning them into a uniform format by, for example, taking the inverse/reciprocal in the definition of TARM, so that it is also the lower the better. Also, the name "measure" does not imply that much as "score" (assuming the higher the better) or "error" (assuming the lower the better). Making these changes may make the metrics better understood and easy to use.

**Questions:**

1. Are there randomness in the evaluation of these metrics? As a benchmark metric, I would assume that it should be deterministic (I should get the same results no matter how many times I run the evaluation). It seems that there could be randomness, for example, when we add corruptions. If so, how large is the randomness and how would they affect the final evaluation results?

---
**Additional comments**:
1. Just a typo: Line 085 says the benchmark is based on 3 datasets, but Line 093 says 4 datasets.
2. Line 123: the citation "Russakovsky et al. (2015)" should use \citep instead of \citet.
3. Line 141: Maybe it will be better to find the authors of mmflow instead of citing "Contributors".
4. Line 166: the citation "Schmalfuss et al. (2022b)" should use \citep instead of \citet.

---

> ### Author Response · Authors · 2024-11-19
>
> Dear Reviewer MVeY,
>
> Thank you very much for your score and for your review, we greatly appreciate your input and suggestions.
> We greatly appreciate the effort invested in also reading the appendix.
>
> For the specific suggestions for improvement, we have now incorporated these in our revised version of the paper (available now, revisions are in blue), thank you very much for these suggestions.
>
> We would like to address your concerns as follows:
>
> 1. Yes, we used EPE in different settings for TARE, NARE, and GAE (previously: TARM, NARM, and GAM) values. However, FlowBench also captures a lot of other metrics, some of them being Cosine Distance, outlier error, 1px error, 3px error, and 5px error (We now describe this in Appendix C). Users are free to choose which metric they would like to use.
> Regarding very high EPE values, yes indeed these very high values are EPE, as shown by Agnihotri, et. al. [3] in their work CosPGD, Section C.2 when evaluated against non-targeted adversarial attacks, the EPE values can exceed 100. This further highlights the extent to which optical flow methods are vulnerable against adversarial attacks.
> We now discuss the specific difference between targeted and non-targeted adversarial attacks in Section 4.2.
>
>
> 2. Yes, we agree this would indeed be very interesting. Our current understanding is that traditional non-DL-based optical flow methods should be more robust to adversarial attacks and most common corruptions when compared to DL-based optical flow methods. However, a few 3D common corruptions considered in this work, like z-blur, and xy-blur might pose a challenge to these traditional methods as well. Given that the community’s motivation behind moving to DL-based optical flow methods was purely i.i.d. Performance-based, it would be interesting to evaluate the robustness of the traditional methods to draw inspiration for future, more robust methods.
> Currently, given the short deadline for the rebuttal, and given that ptlflow currently does not support any traditional methods, it would be very difficult to evaluate them, however, this is a very interesting suggestion that we would definitely consider for the extension of this work and now mention as Future Work in Section 6 of our revised paper.
>
>
> 3. Thank you very much for this suggestion, we have now renamed the metrics as “Errors” where lower is better. For “TARE” (previously “TARM”), we report -1*Error, and thus now the lower the “TARE” value the more reliable the method and vice-versa. We tried using the reciprocal, however, this exhibits a weird scaling that might make it less intuitive for the readers. Please let us know whether the suggested TARE works better from your perspective!
>
>
>
> Answer to Question:
>
> We consider the reproducibility of our results to be of utmost importance. To the best of our understanding, there is no randomness in results. In the case of adversarial attacks such as PGD, and CosPGD the random noise added before optimization is the only point where some stochasticity can be introduced.
> However, using the seeds we used, all evaluations are completely reproducible. We provide these seeds in the code.
> For common corruption, the entire pipeline is completely deterministic.
>
> Please do let us know if you have any further questions or concerns, or if you feel that we missed addressing any of your questions or concerns, we would be happy to address them.
>
> Best Regards
>
> Authors of Paper #1055
>
> References:
>
> [3] Agnihotri, S., Jung, S. &; Keuper, M.. (2024). CosPGD: an efficient white-box adversarial attack for pixel-wise prediction tasks. Proceedings of the 41st International Conference on Machine Learning, in Proceedings of Machine Learning Research 235:416–451 Available from proceedings.mlr.press/v235/agnihotri24b.html.

---

> ### Comment · Reviewer_MVeY · 2024-11-23
>
> Thank you for your response, which resolves most of my concerns. On the randomness of evaluation, I don't think fixing random seeds is a great way to avoid randomness because different versions of packages may behave differently even if we use the same random seed, so maybe it will be better if we could have an analysis on the variance of the metrics to help better understand these metrics.
>
> Overall, although there may be weaknesses of this benchmark, as pointed out by the other reviewers, I still see values of this work as the first benchmark on optical flow robustness and generalization. Also, the extensive experiment results are worth sharing and can help researchers better understand the status quo of optical flow estimation from new perspectives other than just accuracies. Therefore, I agree with Reviewer hv2r and recommend acceptance.

---

> > ### Author Response · Authors · 2024-11-27
> >
> > Dear Reviewer MVeY,
> >
> > Thank you very much for your positive evaluation of our work, we highly appreciate your input and recommendations.
> >
> > We agree with your suggestion that an analysis on the variance of the metrics would help better understand these metrics.
> > We now attempt to address this by revising our “Reproducility Statement” and Appendix K (in the most recent revision) to include experimental evaluations of two kinds: First, different runs on the same seed, Second, runs with different seeds for limited adversarial attacks and common corruptions.
> > As rightly hypothesized by Reviewer MVeY, we observe extremely minute variations in the results in both cases which can be attributed to calculation approximations made by different libraries such as pytorch.
> >
> > We observe that the variance is extremely low and the analysis performed in this work still comfortably holds.
> >
> > Best Regards
> >
> > Authors of Paper #1055

---

### Author Response · Authors · 2024-11-19
**Summary to First Author Response**

Dear All Reviewers,

We are glad to receive this positive evaluation of our work. We thank each reviewer for their suggestions and time spent in reviewing.
“This paper is among the first attempts to evaluate optical flow methods on reliability and generalization ability, which are both important but often overlooked in the research community. The effort put into this research is huge. Building an open-source benchmark tool requires a large amount of coding and implementation. The results on all latest models are insightful and worth sharing.” [Reviewer MVeY]. “This paper provides an interesting analysis that methods with the best accuracy are not necessarily the most robust methods. Given comprehensive analyses, readers can understand the pros and cons of each method. … the benchmark will certainly benefit the community and help design a better and more robust method for future work.” [Reviewer hv2r]. “The work provides a wide-ranging benchmark covering multiple models, datasets, adversarial attacks, and common corruptions. This expansive scope allows researchers to evaluate optical flow models comprehensively. In addition, by focusing on robustness—vital for real-world applications like autonomous driving and medical imaging—it directly addresses the gap between research advancements and deployable technology, which is quite important.” [Reviewer F85d]. “FlowBench provides an easy-to-use API for further study and analyzes the reliability/generalization of current methods.  It is good to have a benchmark of current methods.” [Reviewer GPBg].


All four reviews have made two common suggestions: first, to include more details regarding the proposed and used evaluation metrics for reliability and generalization ability in the main paper rather than just in the appendix; second, to better highlight and discuss the findings. In the revised version of the paper, we have incorporated these suggestions and all other improvements suggested by the individual reviewers. The revised version of the paper is now available. For ease of understanding, we have colored all new changes and fixes in blue color. To the best of our knowledge, we have incorporated all mentioned improvements, please do let us know if we missed any.

Following the suggestion of Reviewer F85d, we have also extended our analysis to two new and recently proposed optical flow estimation methods SplatFlow (February, 2024) [1] and NeuFlow (March, 20204) [2]. The reviewers would be pleased to know that our observations and findings from before still hold, further highlighting the need for this work.

---

> ### Author Response · Authors · 2024-11-19
>
> Here, we summarize the key takeaways of our paper, as highlighted in our revision.
>
> First, we demonstrate the need for FLOWBENCH, the lack of a framework to study the reliability and generalization abilities of optical flow methods alongside their i.i.d. performance has led to the unfortunate reality of methods proposed over time only improving the i.i.d. performance and not their reliability and generalization abilities. As commonly used, reliability is the robustness against white-box adversarial attacks, as they serve as a proxy for the worst-case scenario. Generalization ability is the robustness to image corruptions, specifically 2D and 3D Common Corruptions as they serve as a proxy for real-world domain shifts and signal corruptions. The lack of reliability and generalization ability is severely alarming, as the error of optical flow estimation methods in i.i.d. evaluations is in the low single digits, whereas the error of methods in evaluations against adversarial attacks and common corruptions is in the high hundreds!
>
>
> Second, in Sec. 5.1, we find that there is a high correlation in the performance of optical flow estimation methods against targeted attacks using different targets, thus saving compute resources for future works as they need to evaluate only against one target. Please note that evaluations using adversarial attacks are extremely expensive both time-wise and computation-wise. Thus, the impact of this finding cannot be underscored.
>
>
> Third, in Sec. 5.2, we observe the methods known to be SotA on i.i.d. samples are not reliable, and do not generalize well to image corruptions, demonstrating the gap in current research when considering real-world applications. Additionally, we observe here that there is no apparent correlation between generalization abilities and the reliability of optical flow estimation methods.
>
>
> Fourth, in Sec. 5.3, we show that methods using attention-based pointing matching are marginally more reliable than methods using other matching techniques, while methods using CNN and Cost Volume-based matching have marginally better generalization abilities. Please note that observations three and four together help us conclude that based on current works, different approaches might be required to attain reliability under attacks and generalization ability to image corruptions.
>
>
> Fifth, in Sec. 5.4, we show that, unlike image classification, increasing the number of learnable parameters does not help increase the robustness of optical flow estimation methods.
>
>
> Sixth, in Sec 5.5, we show that white-box adversarial attacks on optical flow estimation methods can be independent of the availability of ground truth information, and can harness the information in the initial flow predictions to optimize attacks, thus overcoming a huge limitation in the field.
>
>
>
> Such an in-depth understanding of reliability and generalization abilities to optical flow estimation methods can only be obtained using our proposed FLOWBENCH. We are certain that FLOWBENCH will be immensely helpful to gather more such interesting findings and its comprehensive and consolidated nature would make things easier for the research community.
>
>
> We would additionally address reviews from each reviewer for finer details and questions asked.
>
>
>
> Thank You,
>
> Regards
>
> Authors of Paper #1055
>
> References:
>
> [1] Wang, Bo, Yifan Zhang, Jian Li, Yang Yu, Zhenping Sun, Li Liu, and Dewen Hu. "Splatflow: Learning multi-frame optical flow via splatting." International Journal of Computer Vision (2024): 1-23.
>
> [2] Zhang, Zhiyong, Huaizu Jiang, and Hanumant Singh. "NeuFlow: Real-time, High-accuracy Optical Flow Estimation on Robots Using Edge Devices." arXiv preprint arXiv:2403.10425 (2024).

---

### Author Response · Authors · 2024-12-03
**Summary of the Author-Reviewer Discussion Phase**

Dear Area Chair and Reviewers,

We would like to sincerely thank all reviewers and the chair for engaging in discussions and for their suggestions.

We propose FlowBench, a robustness benchmark and evaluation tool for learning-based optical flow methods. FlowBench is intended to facilitate streamlined research into the reliability of optical flow methods by benchmarking their robustness to adversarial attacks and out-of-distribution samples. With FlowBench, we benchmark 91 methods across 3 different datasets under 7 diverse adversarial attacks and 23 established common corruptions, making it the most comprehensive robustness analysis of optical flow methods to date.

We have shared a list of sample models available in FLOWBENCH, additionally, we have shared a demo repository (https://anonymous.4open.science/r/flowbench-1055/) upon which we built FLOWBENCH user interface as merely a wrapper, the shared repository already has many attack and evaluation logs and links to model weights (from PTLFlow). Please note that we will make the proposed user interface public upon acceptance.
Lastly, we are working on making our website for FLOWBENCH public, for better user accessibility of evaluations.

We would like to thank Reviewer MVeY for acknowledging the “ values of this work as the first benchmark on optical flow robustness and generalization. Also, the extensive experiment results are worth sharing and can help researchers better understand the status quo of optical flow estimation from new perspectives other than just accuracies.” Reviewer hv2r for mentioning, “I think the paper is already valuable in the sense that it establishes a robustness benchmark for optical flow, casts important questions, and shares analyses by benchmarking nearly a hundred methods. [...]”, and Reviewer GPBg for acknowledging that “FlowBench provides an easy-to-use API for further study and analyzes the reliability/generalization of current methods.”, and Reviewer F85d for acknowledging that this work “directly addresses the gap between research advancements and deployable technology, which is quite important.”.


Following is an overview of the suggestions by the reviewers that we have now included:

1. Improving clarity on the proposed metrics (Reviewer MVeY): Now addressed in the revised submission.

2. Discussion on non-DL based method (Reviewer MVeY): Now addressed in the revised submission and discussion with the Reviewers.

3. Accessibility of TARM (now TARE) (Reviewer MVeY): Now addressed in the revised submission.

4. Reproducibility of evaluations (Reviewer MVeY): Now addressed in the revised submission.

5. Comparability of Evaluated Models (Reviewer hv2r): Now addressed in the revised submission.

6. Having a website for better user experience (Reviewer hv2r): We currently have an initial website, we have included a screenshot of a prototype of this website, free of style (to protect anonymity) in the revised submission.

7. Framing the paper more towards DL-based methods (Reviewer hv2r): Now addressed in the revised submission.

8. Better understanding into the existing optical flow method by grouping them more effectively (Reviewer hv2r and Reviewer GPBg): Now addressed in the revised submission with detailed explanations for the new grouping.

9. Request for demo code, model weights, attack logs, evaluations (Reviewer F85d): Addressed in the replies to the reviewer, and as mentioned by Reviewer hv2r, “**Of course, one can always come up with a better evaluation metric, more realistic evaluation scenarios, and the inclusion of more recent/non-DL methods. However, I think this paper is the first step toward that goal, and we all can improve it as follow-up works.**”

We thank all reviewers as their suggestions have significantly improved the quality of this submission.


To the best of our understanding, we were able to successfully address all concerns and questions of the reviewers.
We thank all reviewers for engaging in discussions and hope for a positive outcome.


Best Regards

Authors of Paper #1055

---

### Meta-Review · Area_Chair_SAX2 · 2024-12-21

**Metareview:**

This paper addresses the task of setting up a benchmark which evaluates the robustness of optical flow techniques. The authors define two sets of robustness (adversarial and out of distribution) and evaluate many optical flow algorithms on these two values: reliability (for adversarial) and generalization ability error (for synthetic corruptions such as Gaussian Noise, Shot Noise, etc.). The paper then presents a considerable amount of raw data noting which of the techniques was vulnerable to different types of adversarial attacks as well as an added section which performs analysis per "method families" such as RAFT or FlowFormer.

The main strength of this paper is that it identifies a missing evaluation and does a considerable amount of work to set up a benchmark and provide data on this. The main weakness is that as currently written the paper primarily consists of raw data with relatively little in-depth analysis or conclusions. The authors correctly state that conclusions and causations are particularly difficult to identify and that the amount of compute required to do architectural or data ablations seems prohibitive.

The main concern raised by the reviewers who advocate for rejection is the lack of "science" in this paper - there is very little actionable facts. While we learn the results of the benchmark, it is not clear what should be done as a result. The main strength of the paper as raised by the reviewers who advocate for acceptance is that the paper addresses an interesting topic, has done a considerable amount of work, and the type of "science" requested by the other reviewers seems quite difficult. It's very hard to separate causation from correlation. This has led to very divergent reviews (3, 5, 8, 8) which are essentially based on each reviewer's concept of a "technical report" versus a scientific paper. Reviewers who are negative about the paper note that it has no "insight" or contribution which leads to future action. Reviewers who are positive about the paper point out that trying to establish a benchmark at all is a useful contribution.

I am personally quite torn. I lean towards rejection and suggest that the authors try their best to draw a coherence "insight" or concrete analysis by 1) doing the ablation studies on a limited subset of models or data sources, 2) trying to make a model more robust on their benchmark. Dataset papers often show that training on their dataset leads to better performance. Evaluation datasets often point the way towards certain techniques and away from others. While adding the 'family method' section makes a step towards this, I think more can be done.

**Additional Comments On Reviewer Discussion:**

Reviewers MVeY and hv2r rate the paper an 8 and suggest that there has been considerable work in setting up a useful benchmark for the field that may allow future algorithms to evaluate their robustness in these two settings.

Reviewers F85d and GPBg rate the paper a 3 and 5 respectively. These reviewers emphasize that the paper has no clear result or insight and is lacking in analysis. Considerable discussion was spent regarding trying to get a concrete 'insight' without any clear response.

See above for additional summary / thoughts.

---

### Decision · Program_Chairs · 2025-01-22

Reject